# Standardization transformation of C-lignin to catechol and propylene

Xiaojun Shen [1,2,3,9] ✉, Zhitong Zhao[4,9], Jialong Wen [1,3], Jian Zhang[2], Yi Ji[5,6], Guangjin Hou [5], Yuhe Liao [7], Chaofeng Zhang [8] ✉, Tong-Qi Yuan[1,3] ✉ & Feng Wang [2] ✉

Standardization transformation of lignin to high-value-added chemicals requires precise control of the reaction process based on the elaborate catalytic strategy design and lignin structure optimization. Here we report the selective and efficient preparation of bio-catechol and bio-propylene from the ideal C-lignin via a one-pot hydrogenolysis-dealkylation cascade catalysis. The optimized catalyst $Ni/HY_{30}$ could orderly cleave the corresponding $C_{\alpha/\beta}-OAr$ bonds and $C_{aryl}-C_{alkyl}$ bonds in the uniform benzodioxane units of C-lignin, which could directionally and selectively provide a 49 mol% yield of catechol and a 45 mol% yield of propylene from C-lignin under 200°C. Further techno-economic analysis and the life-cycle assessment confirmed the potential of this strategy in the $CO_2$-neutral preparation of catechol and propylene. In addition, the control experiments, catalyst characterizations, spectra identification, and DFT calculations indicated that the 4-propenylcatechol primarily generated from the selective hydrogenolysis of C-lignin was the critical intermediate for the following dealkylation, and the side chain was delicately deconstructed via the Brönsted acid-mediated protonation, γ-methyl migration and $C_{aryl}-C_{alkyl}$ scission pathway. Finally, the corresponding strategy design based on the concept of standardization transformation and mechanism revelation focusing on the cleavage of critical linkage bonds could provide guidance for further lignin depolymerization utilization.

With "carbon peaking" and "carbon neutrality" gaining increasing attention in the field of sustainable development, the utilization of renewable biomass resources to maintain the sustainable development of human society, with the ever-increasing need for energy, chemicals, and functional materials, has received extensive attention. The natural polymer lignin, as one of the three main components of lignocellulose represented by agricultural residues and forestry wastes, is normally composed of *p*-coumaryl alcohol (H unit), coniferyl alcohol (G unit), and sinapyl alcohol (S unit), which can be regarded as a unique and sustainable source of aromatic chemicals (Fig. 1)[1,2]. Extensive efforts have been devoted in recent decades to depolymerize lignin into aromatic chemicals by efficiently cleaving stable

[1]State Key Laboratory of Efficient Production of Forest Resources, Beijing Forestry University, Beijing, China. [2]Dalian National Laboratory for Clean Energy, Dalian Institute of Chemical Physics, Chinese Academy of Sciences, Dalian, China. [3]Beijing Key Laboratory of Lignocellulosic Chemistry, Beijing Forestry University, Beijing, China. [4]College of Chemical Engineering and Technology, Taiyuan University of Technology, Taiyuan, China. [5]State Key Laboratory of Catalysis, Dalian Institute of Chemical Physics, Chinese Academy of Sciences, Dalian, China. [6]University of Chinese Academy of Sciences, Beijing, China. [7]Guangzhou Institute of Energy Conversion, Chinese Academy of Sciences, Guangzhou, Guangdong, China. [8]Jiangsu Co-Innovation Center of Efficient Processing and Utilization of Forest Resources, College of Light Industry and Food Engineering, Nanjing Forestry University, Nanjing, China. [9]These authors contributed equally: Xiaojun Shen, Zhitong Zhao. ✉e-mail: shenxiaojun@bjfu.ecu.cn; zhangchaofeng@njfu.edu.cn; ytq581234@bjfu.edu.cn; wagnfeng@dicp.ac.cn

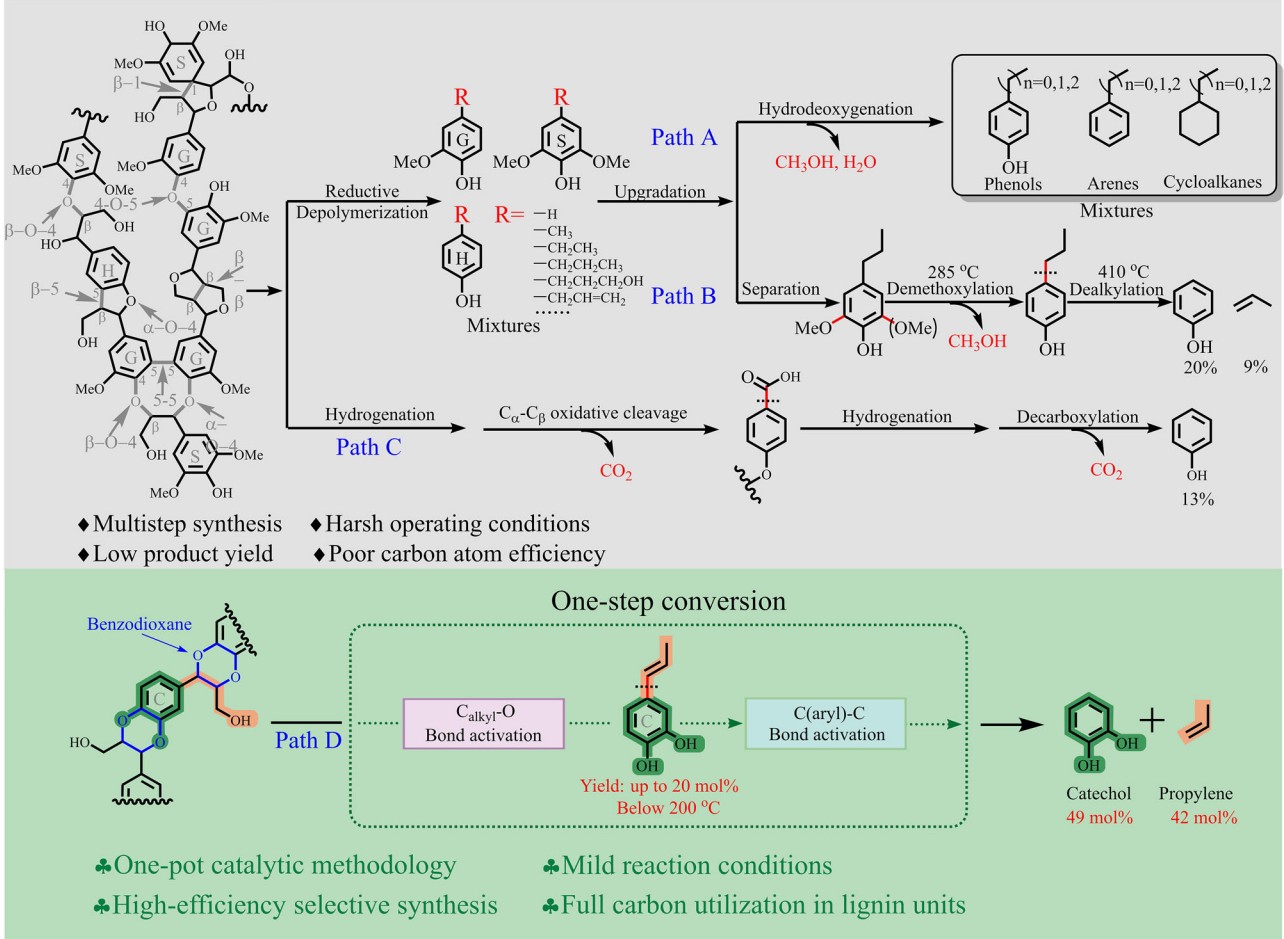

**Fig. 1 | Concepts of lignin standardization transformation based on high-value product normalization and substrate standardization.** Conventional lignin transformation into the mixture (Path A), normalization of normal lignin dealkylation routes (Path B: stepwise depolymerization and dealkylation; Path C: multistep oxidation-hydrogenolysis), and standardization transformation of C-lignin to bio-catechol and bio-propylene (Path D).

C−O and C−C linkages[3–8]. However, its practical application still faces significant challenges, primarily due to unfavorable cost comparisons with well-established petrochemical routes[9]. The efficient separation of various monomeric products from lignin depolymerization mixture is one of the most prominent issues[10]. The oxygen-containing groups make lignin-derived monomers have a high polarity and boiling point, and the physicochemical differences among the monomers with different aromatic nucleis (H-, G-, or S-unit) and side chains skeletons but similar functional groups are not obvious[11]. To commercialize lignin processing, it is necessary to put forward new ideas to integrate and optimize lignin depolymerization, which could efficiently provide high-value monomer products and simultaneously reduce the stress of the separation procedure.

To address the critical challenge of separating complex products from lignin catalytic depolymerization, multi-step tandem strategies[12,13] or sophisticated one-pot catalytic procedures[6,14,15] have been developed. These approaches aim to produce standardized products[9,16], which means a narrow or even single product distribution. For most native lignin depolymerization, especially for the lignin hydrogenolysis/self-hydrogenolysis with a relatively higher monomer yield[5,17], to realize the product standardization from the mixed H-, G-, and S-unit compounds with different structural groups, catalytic removal or transformation of the oxygen-containing hydroxyl-, methoxyl- structures and side $C_1$–$C_3$ alkyl tails on the para-position of phenol group could provide phenols[15,18], alkylphenols[19], arenes[20,21], and cycloalkanes[22], as the major products

(Fig. 1, Path A). Although the above strategies can efficiently achieve the standardization transformation of lignin, the excessive functional group removal will reduce the additional value of aromatic products, and the conversion of side chains to alkanes or $CO_2$ will also reduce the overall value and atomic economy of the reaction (Fig. 1, Path B and Path C)[13,19]. Therefore, the lignin standardization transformation requires precise control of the reaction process based on the elaborate catalytic strategy design and catalytic system establishment to conveniently and efficiently provide high-value-added chemicals (vide infra). Such a concept is also reflected in a recent one-pot catalytic cascade that converts lignin β-O-4 motifs into a single chemical product, 1,2-dimethoxybenzene, via sequential dehydrogenation, retro-aldol cleavage, and decarbonylation[23].

Furthermore, besides the reaction control or further refining, the standardization centering on the structure of the lignin substrate could also provide potential solutions for the separation challenge[9,24,25]. The standardized lignin substrate, with a single monomeric unit and even a homogeneous linkage structure among the units that can be easily transformed, will greatly reduce the difficulty and separation cost for lignin depolymerization to aromatic chemicals. As we all know, genetic engineering based on the corresponding plant gene modification or knockout can directly tune the ratio of monolignols and further intervene in the bond types among monolignols of lignin. For example, the genetically modified poplar with cinnamate-4-hydroxylase/5-hydroxylase overexpression can provide high-S lignin containing a very high proportion rich S units (the S/(G + H) ratio

~40)[25], which prefers to provide a linear lignin structure because of the 4-O, $C_\beta$, and limited $C_1$ radical species generated from the single S-monolignol[3,26]. Furthermore, the β-O-4 linkage remains predominant in this high-S lignin, and the monomer products from high-S poplar lignin depolymerization are mainly syringol-based compounds[27], but the β-β of resinol and spirodienone units was compensatorily higher (Fig. 1), which have relatively stable C−C linkage bonds[3] and may affect the monomer yields. In addition, besides the acquired genetically modified plants, some natural plants also contain unusual lignins due to their genetic defects. In some special cases, like vanilla (*Vanilla planifolia*), various members of the Cactaceae of the genus *Melocactus*, and *Ricinus communis* (castor)[6,24,28,29], the lack of O-methyltransferase (OMT) activity for the conversion of catechyl precursors to guaiacyl-(G) and, subsequently, syringyl- (S) monolignol or their precursors, results in 100% catechyl units or catechyl lignin (C-lignin) in the cell wall. Different from the typical lignin structure, C-lignin was essentially a homopolymer synthesized almost purely by β-O-4 coupling of caffeyl alcohol with the growing polymer chain, producing benzodioxanes as the dominant unit in the polymer (Fig. 1). In addition, because of the lack of an accessible and eliminable benzylic hydroxyl group in the benzodioxane units to benzyl cations that can trigger the condensation reactions, the C-lignin is highly acid-resistant and could be a more ideal lignin archetype for the production of aromatic chemicals[24], which were mainly catechols with alkyl groups[6–8,24,30,31]. While based on the above concept about the standardization transformation of lignin, the structure uniformization of the side chain is a crucial issue for the C-lignin utilization. Compared with the catechol derivatives containing alkyl side chains with a limited market demand, the catechol that traditionally requires multi-step synthesis via the fossil resource path is indeed a fine chemical with large demand, which is widely used in the synthesis of pharmaceutical compounds and functional materials. Therefore, it will be a landmark attempt to achieve the precision conversion of standardized C-lignin to produce high-value-added products, which requires designing a comprehensive strategy with an efficient cascade catalysis system to achieve the selective cleavage of stable benzodioxane ether bonds and inerter $C_{aryl}$−$C_{alkyl}$ bonds (Fig. 1, Path D).

For the comprehensive strategy design of C-lignin conversion, besides the phenolic nuclei, the high-value utilization of $C_3$-dealkylation fragment like the directed olefin synthesis is also noteworthy to focus on[18,32], which can provide two bulk chemicals from one lignin-derived monomer. While the dealkylation-olefination involves the critical $C_{aryl}$−$C_{alkyl}$ bond cleavage and following delicate reconstruction of $C_3$-fragment, the first $C_{aryl}$−$C_{alkyl}$ bond cleavage is usually carried out under a harsh condition with strong acid species and high temperature[32–34], especially for the phenolic substrate with a saturated alkane side chain via the direct generation of the (Ar−H$_{exogenous}$)$^+$ intermediate[3,18]. Referring to the adjacent functional group modification strategy[35,36], the first lignin substrate transformation to generate an active molecule or fragment intermediate with a weakened C−O/C −C bond, or generate new "substrate sites" for introducing a cleavaged reagent, can make lignin depolymerization more efficient at a mild condition. Based on our previous reports[14,15], the *n*-propenyl phenol was much more easily than the *n*-propyl phenol for further dealkylation by indirectly introducing the critical H$^+$ species into the aromatic ring to generate the (Ar−H$_{exogenous}$)$^+$ intermediate via a delicate catalytic process mediated by the zeolite catalyst with $H_2O$. Focusing on the generation of the optimized *n*-propenyl phenol intermediate, the slight hydrogenation dephenolization of the $C_\alpha$−OAr and $C_\beta$−OAr in benzodioxane unit to a C-lignin-derived monomer with the Ar−$C_\alpha$=$C_\beta$ structure should be more convenient and controllable than the normal lignin unit with a highly active $C_\alpha$−OH group and stable $C_\beta$−OAr bond[6] (Fig. 1, Path D), which allows the C-lignin has an inherent advantage in the dealkylation transformation to produce olefin (propylene). Furthermore, to make the propenyl intermediate dominate

the dealkylation without excessive hydrogenation, it is desirable to weaken the catalytic hydrogenation capacity and tune the concentration and generation rate of active hydrogen[5], which means the hydrogenation catalyst should be precisely optimized. At the same time, the tandem acid-dealkylation catalyst needs to have higher selectivity to mediate the dealkylation rather than the transalkylation of the aromatic ring with the released alkyl fragment[14,15,32], and the catalytic system can achieve the oriented reforming of alkyl fragments to olefins, during which the zeolite-based catalysts should be the potential candidate[14,18].

Furthermore, lignin hydrogenolysis over different metal catalysts yields markedly different products, which greatly influences their subsequent utilization and high-value transformation. A lot of investigations have consistently demonstrated that Ni exhibits lower hydrogenation activity compared to noble metals[37]. Even at high loadings (>30 wt%), Ni-based catalysts produce a higher proportion of propenyl-containing and unsaturated products compared to low-loading (5%) noble metal catalysts[38,39]. A low-loading Ni catalyst can effectively catalyze lignin to predominantly yield unsaturated phenolic products, even when using methanol as a hydrogen-donating solvent and under relatively mild conditions[40]. Under low catalyst loadings (5 wt%) and low hydrogen pressures (5–10 bar $H_2$), Ni-based catalysts can achieve high selectivity towards unsaturated phenolic products, such as 4-propenylphenols[41]. Moreover, when using C-lignin as the feedstock, a relatively high-loading Ni (15%) catalyst in a fixed-bed reactor can selectively and efficiently catalyze C-lignin to 4-propenylcatechol[7]. This unique property of Ni is highly beneficial for our C-lignin transformation process. It can selectively hydrogenate C-lignin to 4-propenylcatechol instead of over-hydrogenating it to 4-propylcatechol, which is difficult to dealkylate further. In contrast, noble metals with higher hydrogenation capabilities tend to over-hydrogenate 4-propenylcatechol to 4-propylcatechol, disrupting the subsequent dealkylation reaction sequence.

Based on the above analysis, the selective hydrogenation ability of Ni species[7,17,41], the shape-selective effect and hydrocarbon reforming activity of HY zeolite[14,15] enabled the first simultaneous production of catechol and propylene from the standardization transformation of C-lignin. This approach addresses key challenges in lignin valorization and contributes to sustainable chemical production. The primary objective of this work is to develop a one-pot hydrogenolysis-dealkylation cascade catalytic system capable of selectively cleaving both $C_{\alpha/\beta}$−OAr and $C_{aryl}$−$C_{alkyl}$ bonds in the benzodioxane units of C-lignin. By exploiting the structural homogeneity of C-lignin and the synergistic interaction between Ni nanoparticles and HY zeolite catalysts, the aim is to enable the concurrent production of bio-catechol and bio-propylene under mild conditions. Mechanistic insights into bond cleavage pathways, catalyst structure-function relationships, and intermediate transformations will be systematically investigated through experimental and DFT calculations. Additionally, techno-economic analysis (TEA) and life-cycle assessment (LCA) will evaluate the feasibility and sustainability of this strategy, providing a holistic framework for lignin valorization that aligns with carbon-neutral objectives. This study not only advances the utilization of lignin as a renewable feedstock but also sets a foundation for designing cascade catalytic systems to overcome the inherent heterogeneity and recalcitrance of lignocellulosic biomass.

## Results
The standardized C-lignin substrate was first isolated from the endocarp of castor seed coats via ball-milling and enzymatic hydrolysis. 2D-HSQC NMR analysis revealed that the benzodioxane β-O-4 bond was the only linkage in the isolated lignin (Fig. 2b, c), suggesting that C-lignin is the overwhelming lignin polymers in endocarp of castor seed coats. The purity of C-lignin and molar concentration of caffeyl alcohol was 81.42 wt% and 3.54 mmol/g, respectively (see

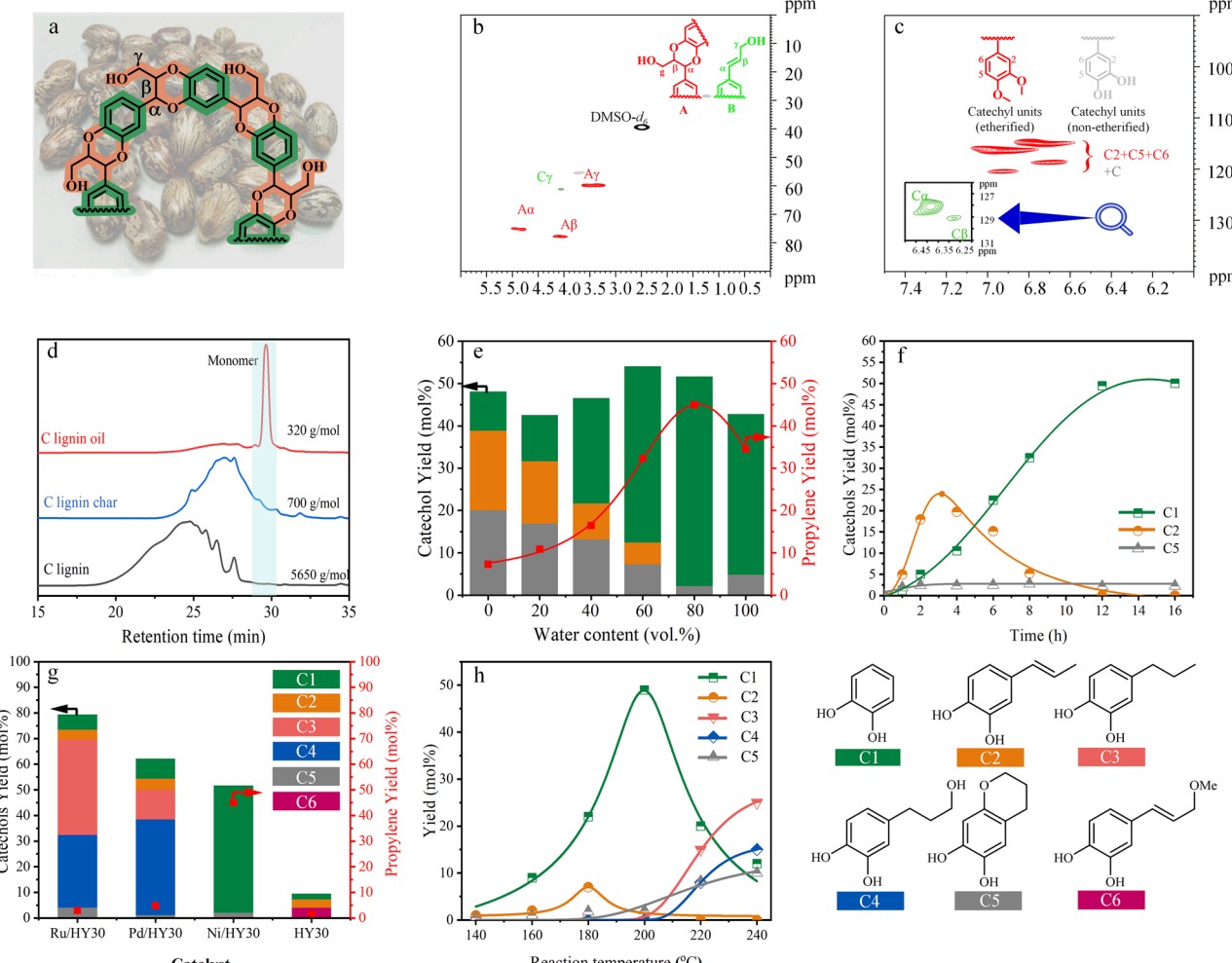

**Fig. 2 | Hydrogenolysis-dealkylation of C-lignin over Ni/HY₃₀ catalyst.**
**a** Overview of C-lignin structure. **b, c** 2D-HSQC NMR spectrum of C-lignin from the endocarp of castor seeds (DMSO-$d_6$); **d** The molecular weight distribution of C-lignin, oil, and char from catalytic reactions via gel-permeation chromatography analysis; The effects of **e** water content in methanol solution, **f** reaction time, **g** catalyst type, and **h** temperature on catechol and propylene yields, along with the corresponding product distributions. Reaction conditions: 50 mg C-lignin, 100 mg catalyst, 25 μL dodecane, 5 mL solvent, 3 MPa $H_2$, 200 °C, 12 h.

Supplementary Table 12 and Supplementary Fig. 5). Therefore, this C-lignin sample is the ideal substrate to produce catechol and propylene via hydrogenolysis-dealkylation cascade catalysis. For the construction of an effective cascade catalytic system, non-noble metal Ni is considered a promising hydrogenation catalyst due to its tunable catalytic behavior and demonstrated ability to cleave β-O-4 linkages. This enables the selective production of phenolic monomers bearing unsaturated propenyl side chains[7,17,41], which is much more easily than the *n*-propyl phenol for further dealkylation via decreasing the bond dissociation energy (BDE) of critical $C_{aryl}$–$C_{alkyl}$ bonds[14,15] by introducing the essential acid species of zeolite catalysts[18,32,42,43]. Therefore, the combination of Ni and zeolite was expected to achieve the hydrogenolysis-dealkylation reaction of C-lignin to generate catechol. The preliminary condition optimization based on the yield of dealkylation products showed that the reaction, performed at 200 °C, 3.0 MPa $H_2$ with 0.97 wt% Ni/HY₃₀ catalyst could provide a soluble fraction (58 wt%), a gaseous fraction (17 wt%), and a solid residue (22 wt%) in 12 h with an optimized solvent (methanol:water = 1:4) (Fig. 2d), which could keep a 97% mass balance based on the weight of the original C-lignin. The gel-permeation chromatography (GPC) analysis of the generated C-lignin oil and solid char after the reaction showed a significant decrease in the molecular weight (320 and 700 g/mol) as compared to the original C-lignin (5650 g/mol) (Fig. 2d). Further chromatography analysis showed that the reaction provided a 51 mol%

yields of well-defined catechols based on the molar concentration of caffeyl alcohol units in C-lignin (Fig. 2f), during which the yield of simple catechol (C1) without side chain could reach to 49 mol% and its selectivity could be up to 96% (Supplementary Fig. 6 and Supplementary Fig. 7). In addition, for the synergistic effects between the zeolite support and loading metal species, different control experiments were carried out (Fig. 2g). Firstly, the depolymerization efficiency of C-lignin was low when only the HY₃₀ catalyst was used, due to the high acid stability of its predominant benzodioxane structures (Fig. 2g). However, when the metal Ni species were introduced into the catalytic system, the unreacted C-lignin could be efficiently converted to catechol (C1). The above experiments not only preliminarily confirmed the hydrogenolysis-dealkylation cascade catalysis but also excluded the potential acidolysis-hydrogenolysis-dealkylation mechanism for the Ni/HY₃₀-mediated lignin depolymerization.

In addition, it is well known that the noble metals (e.g., Ru and Pd) have a stronger hydrogenation capacity than Ni for the lignin hydrogenation depolymerization[1,44,45]. As shown in Fig. 2g, the yield of aromatic monomers using Ru/HY₃₀ and Pd/HY₃₀ is evidently higher than that of Ni/HY₃₀. However, this higher hydrogenation activity also causes drawbacks. Ru/HY₃₀ and Pd/HY₃₀ tend to catalyze the deconstruction of the benzodioxane β-O-4 bond and hydrogenate the side chain to saturated alkyl catechols (C3 and C4). The saturated alkyl catechols are relatively stable, and their $C_{aryl}$-$C_{alkyl}$ bonds are difficult

to cleave, which is unfavorable for the production of catechol and propylene. In contrast, Ni/HY$_{30}$ exhibits unique selectivity. It can control the hydrogenolysis of C-lignin to preferentially generate 4-propenyl catechol (C2) (Fig. 2f). This is a crucial advantage because 4-propenyl catechol serves as an ideal intermediate for the subsequent dealkylation reaction, enabling the efficient production of the target products. From an application perspective, in industrial processes where the production of catechol and propylene from lignin is desired, Ni/HY$_{30}$-based catalysts can be better optimized to meet production requirements, while noble metal-based catalysts may need to be modified to avoid excessive hydrogenation. In addition, the reaction temperature also played a critical role in the selectivity of the first hydrogenolysis of the C-lignin, which can further affect the dealkylation by the tuning of critical intermediate. When the reaction temperature was above 200 °C, the catechols with saturated alkyl group began generating due to the hydrogenation of the C=C bond in the side chain of catechols at the high temperature (Fig. 2h)[41,46-48]. In addition, decreasing the reaction below 200 °C could restrain the activity of both hydrogenolysis and dealkylation reactions, which provided a low yield of catechol and a considerable amount of unconverted 4-propenyl catechol (C2).

Besides the effect of loading metal on the hydrogenolysis of C-lignin transformation, the zeolite support also played multiple roles in the hydrogenolysis-dealkylation of C-lignin. As shown in Supplementary Fig. 9, tuning the silica-alumina ratio of HY zeolite did not affect the yield of the degradation product but could change the distribution of the degradation product. Although lower Si/Al in zeolite (HY$_{5.2}$) would improve its acidity, most acid sites are inserted deeply in a microporous framework, which limits the accessibility to the reaction site[14]. Thus, compared with the catalytic performance of Ni/HY$_{30}$, the Ni/HY$_{5.2}$ could provide an approximate monomeric product yield but the catechol selectivity was decreased to 24.5% (Supplementary Fig. 9). On the contrary, the high-silica HY zeolites (HY$_{60}$ and HY$_{80}$) in which the framework aluminium was substituted by silicon have more abundant mesoporous pores, which enabled reactants to diffuse freely to the acid sites in the zeolite. However, compared to the high-Al zeolite (HY$_{5.2}$ and HY$_{30}$), HY$_{60}$ and HY$_{80}$ have a lower concentration of acid sites, especially strong acid sites, resulting in a lower catechol yield under the same conditions[14]. Therefore, HY$_{30}$ contained appropriate strong acid sites exposed in the mesoporous pore channels, which can catalyze the dealkylation of 4-propenylcatechol to form catechol and propylene with a high yield.

In addition to the effects of the intrinsic acid-base properties and microscopic pore structure of the zeolite catalysts on the hydrogenolysis-dealkylation of C-lignin, the reaction solvent plays an important role in activating the acidic sites and influencing the catalytic cycle of the active centers in the zeolite catalyst during the dealkylation of C-lignin fragments. Although methanol has been successfully used as a reaction medium for lignin hydrogenolysis depolymerization, the C-lignin conversion over the Ni/HY$_{30}$ catalyst in pure methanol provides a 48 mol% yield of phenolic monomers. However, the selectivity of catechol without a side chain was only 9.12 mol%, indicating that pure methanol is not suitable for the side-chain dealkylation of catechols (Fig. 2e). To enhance the performance of the HY zeolite catalyst in the dealkylation step, water was introduced into the reaction system to improve local acidity in the mesoporous pores of the zeolite, referring to previous studies on zeolite catalysis[49]. As expected, the selectivity of catechol (C2), the dealkylation product, increased gradually as the water proportion in the mixture was increased (Fig. 2e). Water helps form hydrated hydronium ions (H$_3$O$^+$) in the mesoporous pores of the HY zeolite. These hydrated hydronium ions serve as active acidic species, which protonate the 4-propenylcatechol intermediate, making the C$_{aryl}$–C$_{alkyl}$ bond more susceptible to cleavage. The positive charge on the hydronium ion polarizes the C–C bond in the side chain,

weakening it and lowering the activation energy required for the dealkylation reaction.

However, using pure water as the sole solvent limits catechol production due to the low solubility of C-lignin and intermediates in water[50]. The C-lignin, as a complex polymer, has a large molecular structure and high polarity, resulting in poor solubility in pure water. This limited solubility reduces contact between C-lignin and the catalyst, decreasing the reaction rate. Furthermore, some key intermediates in the reaction pathway also have low solubility in water, causing them to precipitate out of the reaction system and preventing them from participating in subsequent reaction steps. This further reduces the yield of catechol. The optimized water ratio is 80 vol%, as it balances the enhancement of catalytic activity via water's acidity-related effects while maintaining sufficient solubility of C-lignin and intermediates for efficient reaction progress. At this ratio, the water molecules effectively activate the acidic sites on the zeolite catalyst, promoting the dealkylation reaction, while ensuring that C-lignin and intermediates remain in solution to facilitate continuous reaction progress. This balance is crucial for maximizing the production of catechol and propylene from C-lignin. In addition, the C3 fragment after the dealkylation of the C-lignin fragment could be transformed to acetone, during which the O element from the H$_2$O that participates in the pre-activation of the acid centers could be transferred into the C3 fragment to acetone after the catalytic cycle (details in mechanism study discussion and DFT calculation). Furthermore, the acetone could be in situ converted into propylene via Ni/HY$_{30}$-catalyzed hydrodehydration (details in Supplementary Fig. 10).

To reveal the underlying reason for the observed excellent catalytic performance and provide support for further mechanism study, several techniques to characterize the catalysts were performed. The Ni content in the Ni/HY$_{30}$ catalyst was 0.97 wt% according to the analysis of inductively coupled plasma-optical emission spectroscopy (ICP-AES) (Supplementary Table 13). The weight percentage of Ni content in the spent Ni/HY$_{30}$ was 0.91 wt%, and no leaching of Ni element from the Ni/HY$_{30}$ catalyst was observed in the liquid fraction. The typical XRD pattern of the as-synthesized Ni/HY$_{30}$ catalyst showed well-defined diffraction peaks of the HY$_{30}$ zeolite structures (JCPSD card 38-0239) (Supplementary Fig. 11). However, no characteristic peaks of Ni or NiO species were observed in XRD, further TEM image showed that the Ni nanoparticles (Ni NPs) are uniformly distributed on HY$_{30}$ zeolite with a small average size of 1.3 nm (Fig. 3a and Supplementary Fig. 12). Meanwhile, after Ni loading, Ni/HY$_{30}$ still had rich mesoporous structures, which was conducive to the access of the degradation product of C-lignin to the acid sites for the rapid dealkylation of the side chain in the degradation product (Fig. 3b). In the X-ray photoelectron spectra of Ni/HY$_{30}$, the binding energy of Ni 2$p_{3/2}$ at 852.3 and 854.3 ev corresponded to the metallic and oxidized nickel on HY$_{30}$, respectively (Fig. 3c), and the metallic nickel was the dominant species in Ni/HY$_{30}$, which favors the hydrogenolysis of lignin[51]. These results are consistent with the CO-absorption Fourier transform–infrared (CO-FTIR) spectra analysis (Fig. 3d)[52]. No signal of ionic Ni was observed in the CO-FTIR spectra, indicating that the Ni species in the Ni/HY$_{30}$ catalyst exist predominantly in metallic and oxidized forms. Besides, no peaks were found in the CO-FTIR spectra of HY$_{30}$ after CO gas adsorption for 30 min (Supplementary Fig. 13). These results illustrate that Ni NPs are highly dispersed on HY$_{30}$ in metallic form; hence, C-lignin can be efficiently depolymerized into 4-propenyl catechol, which was a favor to further dealkylation for the selective production of catechol and propylene.

As compared to the C$_\alpha$–OAr/C$_\beta$–OAr bonds of the benzodioxane unit, scissoring the alkyl "tails of lignin" via the cleavage of the C$_{aryl}$–C$_{alkyl}$ bond is much more difficult but crucial[33,34,53]. Therefore, this work paid more attention to the mechanism study of the C$_{aryl}$–C$_{alkyl}$ bond cleavage, focusing on the reaction route, critical intermediates,

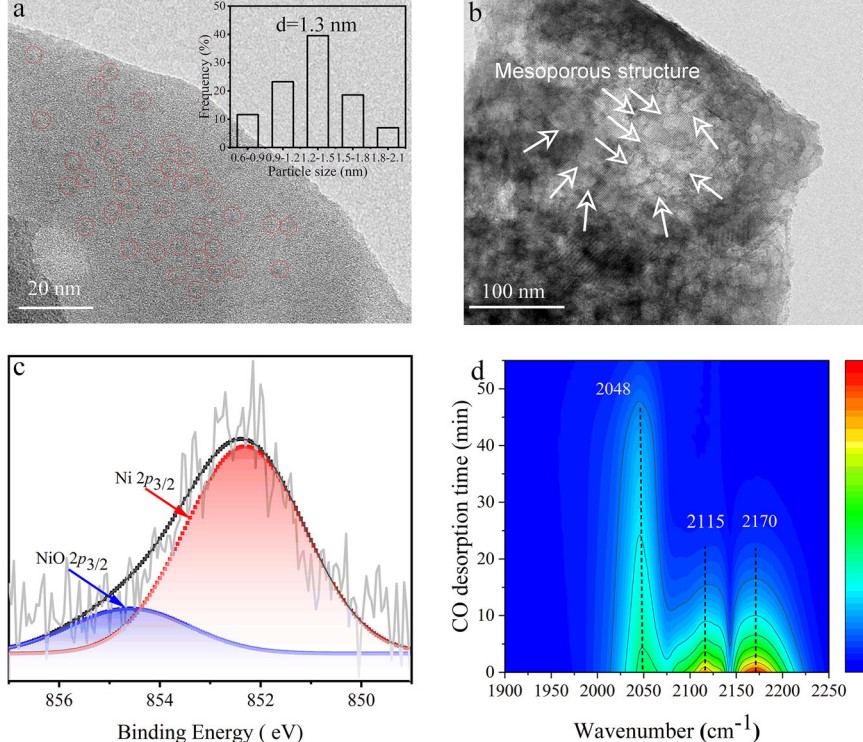

**Fig. 3 | Characterizations of Ni/HY$_{30}$ catalyst. a,b** TEM images (0.97 wt% Ni, inset: Ni particle size distribution of Ni/HY$_{30}$ catalyst), **c** nXPS spectra of Ni $2p_{3/2}$, and **d** CO-FTIR spectra at different desorption times.

and the structure-function relationship of the critical active centers of the catalyst.

As shown in the reaction curve of the C-lignin conversion (Fig. 2f), the first generation and then conversion of the 4-propenyl catechol, accompanied by the generation of the final product catechol, indicated that 4-propenyl catechol could be the critical intermediate in the C$_{aryl}$−C$_{alkyl}$ cleavage of C-lignin. Therefore, 4-propenyl catechol was used as the feedstock to investigate the dealkylation mechanism of C-lignin (Supplementary Table 14). As expected, HY$_{30}$ can efficiently catalyze the dealkylation of 4-propenyl catechol into catechol with a >99% yield, even if the reaction temperature decreases to 180 °C in water (Fig. 4 and Supplementary Table 14, Entry 1). Meanwhile, the alkyl side chain of 4-propenylcatechol was converted into acetone. To get more information about the dealkylation mechanism of the 4-propenyl catechol and other key intermediates during the following transformation of the 4-propenyl catechol, more control experiments were performed. The 4-propenyl catechol conversion in water was carried out at 180 °C using 50 mg HY$_{30}$ (half dosage). As expected, due to lower acidity, the yield of catechol was decreased to 43% but provided some intermediates beyond the 4-propenylcatechol. The reactivity of these possible reaction intermediates (Fig. 4) was checked by the control experiments. Firstly, since the BDE of the C$_{aryl}$−C$_{alkyl}$ bond of 4-propyl catechol (Fig. 4, Entry 2) was relatively high, the efficient cleavage of the C$_{aryl}$−C$_{alkyl}$ bond in the alkylphenols required high temperatures above 300 °C[18,32,42,43]. Herein, only 9% of 4-propyl catechol was converted at 180 °C with an 8% yield of catechol, which suggested that the further hydrogenation product 4-propyl catechol was not the critical intermediate of the process. Furthermore, the C=C double bond in the side chain of 4-propenyl catechol could be hydrated to 1,2-dihydroxy-4-(1-hydroxypropyl)benzene with the catalytic assistance of the acid. Given the fact that the 1,2-dihydroxy-4-(1-hydroxypropyl)benzene is quite unstable, the equivalent molecule 1-(3,4-ethylenedioxyphenyl) propan-1-ol (Fig. 4, Entry 3) was chosen to identify the key intermediates, which could provide an 89% yield of

benzo-1,4-dioxane with the cleavage of C$_{aryl}$−C$_{alkyl}$ bond (Fig. 4). In contrast, 1-phenyl-1,2-ethanediol (Fig. 4, Entry 4), which contains two hydroxyl groups on the C$_{α}$ and C$_{β}$ positions of the side chain, exhibited higher reactivity under strong acidic conditions. This promoted undesired C−C coupling and carbon deposition, resulting in >99% conversion but no detectable benzene product. Furthermore, theoretically, when the methyl group in the γ position of the 4-propenylcatechol was isomerized at the aliphatic C$_{α}$ position with the assistance of the acid species, 4-(prop-1-en-2-yl)benzene-1,2-diol should be generated. However, due to the instability, its dimethoxy-substituted analogue (Fig. 4, Entry 5) was used and exhibited a reactivity pattern similar to that of 4-propenylcatechol and provide a 92% yield of the dealkylation product catechol. In addition, the 3,4-dimethoxycumyl alcohol (Fig. 4, Entry 6), hydrate product of the above 1,2-dimethoxy-4-(prop-1-en-2-yl)benzene (Fig. 4, Entry 5), could also provide a 94% of the dealkylation product catechol.

Furthermore, besides the conversion difference of various substrates, the solvent also played an important role in the performance of the 4-propenylcatechol dealkylation. Water was found to be an excellent reaction medium for the dealkylation of 4-propenylcatechol, outperforming pure methanol and ethanol (Supplementary Table 14, Entries 10 and 11). As reported previously[49], the transition state of hydrated hydronium ions as the critical acid species can be stabilized when the zeolite-mediated reaction is carried out in water, which could also promote the hydration reactions. This stabilization of the transition state is a key factor in the enhanced dealkylation performance in water. In a water-rich environment, the hydrogen-bonding network formed by water molecules can interact with both the reactant and the transition state, facilitating their solvation. This solvation effect is more pronounced in water than in methanol or ethanol. In addition, water can interact with the acidic sites on the zeolite surface to form a more acidic micro-environment. Hydrated hydronium ions enhance the acidity of the zeolite's active sites, facilitating the protonation of the reactant and the subsequent cleavage of the C$_{aryl}$-C$_{alkyl}$ bond. In

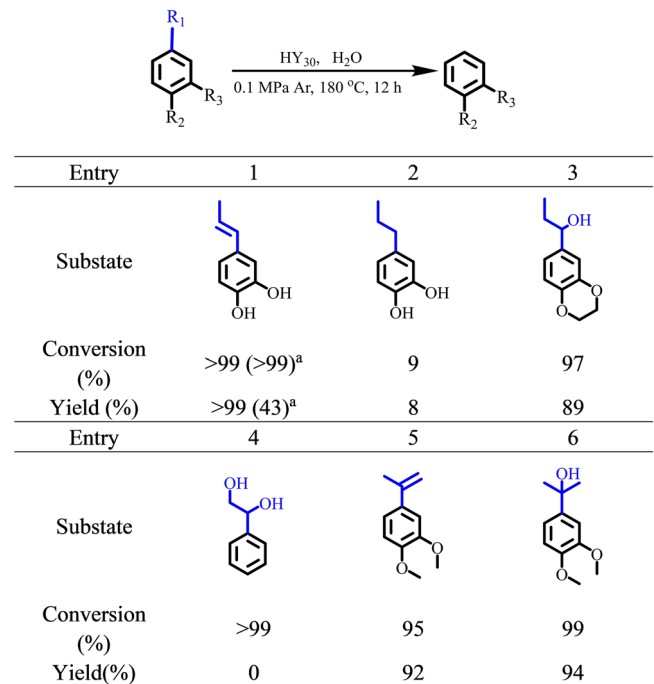

| Entry | 1 | 2 | 3 |
|---|---|---|---|
| Substate | | | |
| Conversion (%) | >99 (>99)ᵃ | 9 | 97 |
| Yield (%) | >99 (43)ᵃ | 8 | 89 |
| Entry | 4 | 5 | 6 |
| Substate | | | |
| Conversion (%) | >99 | 95 | 99 |
| Yield(%) | 0 | 92 | 94 |

**Fig. 4 | Control experiments on reaction mechanism.** Reaction condition: 50 mg substrate, 100 mg $HY_{30}$, 5 mL $H_2O$, 180 °C, 0.1 MPa Ar, 12 h. Note: ᵃ50 mg $HY_{30}$ was used.

contrast, methanol and ethanol are less effective at generating such a strong acidic micro-environment. Therefore, water is a superior solvent for the dealkylation of 4-propenylcatechol compared to pure methanol and ethanol. Its unique ability to stabilize the transition state, solvate charged species, interact with the zeolite catalyst's acidic sites, and create a favorable reaction micro-environment makes it the ideal solvent for optimizing the catalytic process and improving the overall efficiency of lignin conversion to high-value products.

Therefore, the above control experiments confirmed that the 4-propenyl catechol was the critical intermediate of the dealkylation mechanism, and the following transformation might also involve other intermediates, such as the hydrated intermediates, which call for a further structure-function relationship study of the catalyst.

To further reveal the interaction between the key intermediate C2 and heterogeneous HY zeolite catalyst, and explore the further molecular dealkylation mechanism, it is necessary to first identify the location and the more important acid type of the active centers in the zeolite catalysts. Firstly, a series of $HY_{30}$ catalysts ($HY_{30}$-X) with different acidity but similar textural properties were synthesized via sodium exchange[54], where the X represented the concentration of sodium acetate during sodium exchange. Then, the in-situ pyridine-IR characterization was carried out to investigate the acid types (Brönsted and Lewis acid sites, denoted as BAS and LAS) and amounts (Fig. 4a, b) of various $HY_{30}$ zeolites[55]. The concentration of BAS (c.a. 1545 cm$^{-1}$ IR peak) and LAS (c.a. 1445 cm$^{-1}$ IR peak) of the original $HY_{30}$ was 98 and 19 μmol/g, respectively, which implied that the cavity of the $HY_{30}$ surface was dominated by the strong BAS species. In addition, the introduction of Na into the $HY_{30}$ catalyst could remarkably decrease the BAS concentration but show a limited effect on the concentration of the LAS species (Fig. 5a), The BAS/LAS concentration ratio of various $HY_{30}$ zeolites decreased from 5.16 to 0 by increasing the concentration of sodium acetate to 2.4 mol/L (Fig. 5b). The 2.4 mol/L sodium acetate treatment could provide a complete transformation of the $HY_{30}$ to its sodium form $NaY_{30}$ ($HY_{30}$-2.4), which only presented the LAS species with a concentration of 16 μmol/g (Fig. 5a, b). Furthermore, it was found that the specific surface area and pore diameter

of the $HY_{30}$ and Na-modified $HY_{30}$-X zeolite catalysts kept almost unchanged (Fig. 5c, Supplementary Fig. 14, and Supplementary Table 15), which provides ideal catalysts to study the catalytic mechanism of BAS species in the 4-propenylcatechol (C2) dealkylation reaction by revealing the correlation between reactivity and acid concentration. As shown in Fig. 5d, the density of the reactive site was found chiefly proportional to the BAS concentration, translating into an almost constant normalized catechol yield (red points of Fig. 5d). On the contrary, the LAS species showed no activity on the dealkylation, which could be supported by the $HY_{30}$-2.4 with a sufficient sodium exchange. The above discussions indicated that the uniform nature of the BAS throughout the $HY_{30}$ zeolite was the reactive site of dealkylation.

In addition, it has been reported that turning the Si/Al ratio would change the acidity and pore diameter of zeolite, which leads to different reactivity[14]. Although the acidity would be increased via a decrease in Si/Al ratio, these acidity sites, especially the strong acid sites, were distributed in the micropores framework, limiting substrates to contact with the reactive sites for the dealkylation (Supplementary Table 14, Entry 3). On the contrary, increasing the Si/Al ratio can improve the accessibility of substrate to the acid sites that are located in the mesopores framework, but the accompanying acid strength decrease could inhibit the dealkylation of 4-propenylcatechol. The balance between the two factors made $HY_{30}$ zeolite show the optimized performance. Besides the HY zeolites, other types of zeolites, such as ZSM-5, MOR, Beta, MCM, and SAPO-34 were used in the test of 4-propenylcatechol dealkylation (Supplementary Table 14, Entries 4–8), but all showed low activities on the cleavage of the $C_{aryl}$–$C_{alkyl}$ bond. These supports exhibited significantly lower catalytic efficiency due to either excessive microporosity (MOR, limiting diffusion) or insufficient acid strength (MCM-41, SAPO-34, reducing catalytic activity). This further reinforces the selection of HY zeolites, particularly $HY_{30}$, as the optimal support. Furthermore, besides the heterogeneous Brönsted acid, the mineral acid HCl could catalyze the reaction but only afforded a 25.2% yield of catechol because of the instability of diphenols under strongly acidic conditions (Supplementary Table 14, Entry 9), which further highlighted the special catalysis of the BAS species in the dealkylation of 4-propenylcatechol.

According to the above experimental results, a new cascade catalytic strategy that could efficiently convert the benzodioxane units in C-lignin to catechol and propylene was achieved. The high efficiency of the strategy in cleaving C–OAr and $C_{aryl}$–$C_{alkyl}$ bonds could be attributed to the well-coordinated synergistic effect between the Ni NPs and $HY_{30}$ zeolite support. In this process, the Ni NPs firstly catalyzed the selective hydrogenolysis of the benzodioxane structure into the 4-propenylcatechol without the overhydrogenation of the side chain, while the Brönsted acid centers of the $HY_{30}$ zeolite subsequently catalyzed the dealkylation of 4-propenylcatechol into catechol and propylene. Besides the well-studied lignin hydrogenolysis to phenolic monomers with ether bonds, selective cleavage and following acetone selective hydrodehydration to propylene, the mechanism of the side chain dealkylation, especially for the efficient C-lignin transformation via a different original intermediate, should be investigated from the molecular level to reveal the structure-function relationship of the $HY_{30}$ catalyst.

The density functional theory (DFT) calculations for the 4-propenylcatechol dealkylation over the Brönsted acid centers of the $HY_{30}$ zeolite were carried out. As shown in Fig. 6a, the C=C bond in the side chain of 4-propenylcatechol was adsorbed on $HY_{30}$ (I), and then the $C_\beta$ in 4-propenylcatechol was bonded with the proton (H$^+$) in $HY_{30}$, leading to the generation of the carbenium ion (II). DFT calculations show that the energy barrier for the transition from intermediate I to II (i.e., direct protonation of the C=C bond) is only 0.05 eV (Fig. 6a), indicating that this process occurs readily without requiring prior

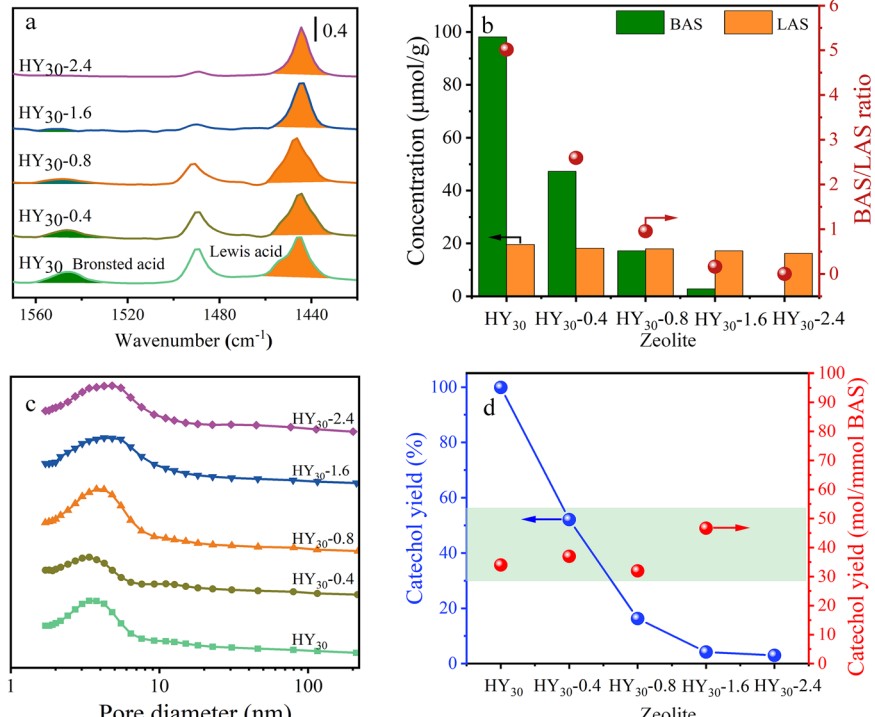

**Fig. 5 | Characterization and catalytic performance of HY$_{30}$ catalysts. a** Pyridine-IR spectra, **b** acid concentration and the BAS/LAS ratio, and **c** pore size distribution of HY$_{30}$, HY$_{30}$-0.4, HY$_{30}$-0.8, HY$_{30}$-1.6 and HY$_{30}$-2.4; **d** The catechol yield using HY$_{30}$, HY$_{30}$-0.4, HY$_{30}$-0.8, HY$_{30}$-1.6 and HY$_{30}$-2.4 with the similar textural property. Reaction condition: 100 mg substrate, 100 mg HY$_{30}$ catalyst, 5 ml H$_2$O, 0.1 MPa Ar, 180 °C, 12 h.

hydration and further dehydroxylation. Direct protonation is thermodynamically favored, simplifying the reaction pathway. Then, the γ-methyl on the side chain in structure II was thermodynamically migrated to the $C_\alpha$ position under the catalysis of the Brönsted acid center, involved in the chemically adsorbed structure III with an endothermic step of 0.10 eV. Subsequent proton abstraction by HY$_{30}$ zeolite delivers structure IV. Meanwhile, the $C_\alpha$ position in structure IV was protonated again to deliver carbonium ion structure V, which further captured H$_2$O to give the tert-propanol type geometry (VI) and simultaneously regenerate the Brönsted acid center of HY$_{30}$ zeolite. After that, the benzene ring skeleton of tert-propanol type geometry (VI) absorbed the Brönsted acid center of HY$_{30}$ zeolite to facilitate the protonation of the $C_{sp2}$ position in the benzene ring to deliver the carbonium ion VII, which further undergoes the original $C_{aryl}$−$C_{alkyl}$ bond scission and endothermically desorb from zeolite surface to generate catechol and acetone[56].

The above calculations and the potential mechanism were further supported by the spectral characterizations. The interaction between the reactant and active center in HY$_{30}$ results in the chemical bonding state, which was observed by the solid-state NMR analysis of the reaction system after 5 min quenching (Fig. 6b and Supplementary Table 16). Although the $^{13}$C NMR signal for the $C_\gamma$ atom in the C2 reactant could still be observed, new signals of $C_{\alpha1}$, $C_{\beta1}$, and $C_{\gamma2}$ in the protonated intermediate C7 structure at 46.9, 29.1, and 12.4 ppm were observed, indicating that C2 was rapidly protonated in the HY$_{30}$ framework structure to form the intermediate structure II (C7). It can be demonstrated that the unstable chemical properties of the carbon-carbon double bonds in the side chains of 4-propenylcatechol are the inducing factors for the efficient cleavage of $C_{aryl}$−$C_{alkyl}$ bonds under mild conditions and finally realize the direct extraction of catechol and propylene. Furthermore, the $^{13}$C NMR spectrum of the C−C bond evolution in the reaction of 4-propenylcatechol intermediate exhibited related signals ($C_{\alpha2}$, $C_{\beta2}$, and $C_{\beta3}$) of the γ-methyl migration intermediate structure III (C8) at 42.0, 21.9, and 36.4 ppm were detected[57].

As compared to the side chain in the intermediate structure II, the peaks of intermediate structure III (C8) shift to the lower field. This is because the $C_{\beta2}$ atom in the γ-methyl migration intermediate structure III will chemically bond with the oxygen atom at the center of the Bronsted acidic site [≡Al−O−Si≡] in the HY$_{30}$ zeolite framework, which shifted its resonance downfield from that of the original $C_{\beta1}$ and $C_{\beta2}$ atoms[58,59]. The shift changes further support the Bronsted acid-catalyzed mechanism for γ-methyl transfer. Echoing the $^{13}$C spectrum results, the chemical bond in the [≡Al−O($C_{\beta2}$)−Si≡] unit is also reflected to the tetrahedrally coordinated Al [Al(VI)] of the Bronsted acid center, i.e., through the solid $^{27}$Al NMR nuclear magnetic analysis further confirmed. As shown in Fig. 6c, only the common framework Al(VI) signal of the fresh HY$_{30}$ was clearly evident at 60.8 ppm[60]. Besides the above regular Al(VI) signal, a shoulder NMR signal at 56.7 ppm is distinctly identified in the HY$_{30}$ after the reaction (Fig. 6d)[60]. This new aluminium state is assigned to the Al(VI) atom in the distorted [≡Al−O($C_{\beta2}$)−Si≡] unit, where the charge compensation of the $C_{\beta2}$ atom weakens the inductive effect of the oxygen atom on the neighboring Al(VI) atom, reducing the angle of the Al−O−Si bond, prolonging the Al−O bond and Si−O bond, and finally the NMR signal of the Al(VI) atom in the regular [≡Al−O−Si≡] environment shift to the high field[61]. This γ-methyl migration is key to the formation of the 4-tert-propanol group, favoring eventual side chain scission. In addition, the protonated structure VII (C9) of the benzene ring was observed in the carbon spectrum. It shows that most of the 4-propenylcatechol hydrate is protonated quickly, the γ-methyl group is further migrated, and finally, the side chain is broken through the aromatic ring carbocation intermediate. Among them, the content of the γ-methyl transfer intermediate is the lowest, thus, it is speculated that the generation of γ-methyl transfer structure III is the rate-determining step of the whole dealkylation reaction.

As mentioned above, the mild and efficient refining of 4-propenylcatechol, proceeded by a step-by-step protonation, γ-methyl migration, and original $C_{aryl}$−$C_{alkyl}$ β scission, should be a potential

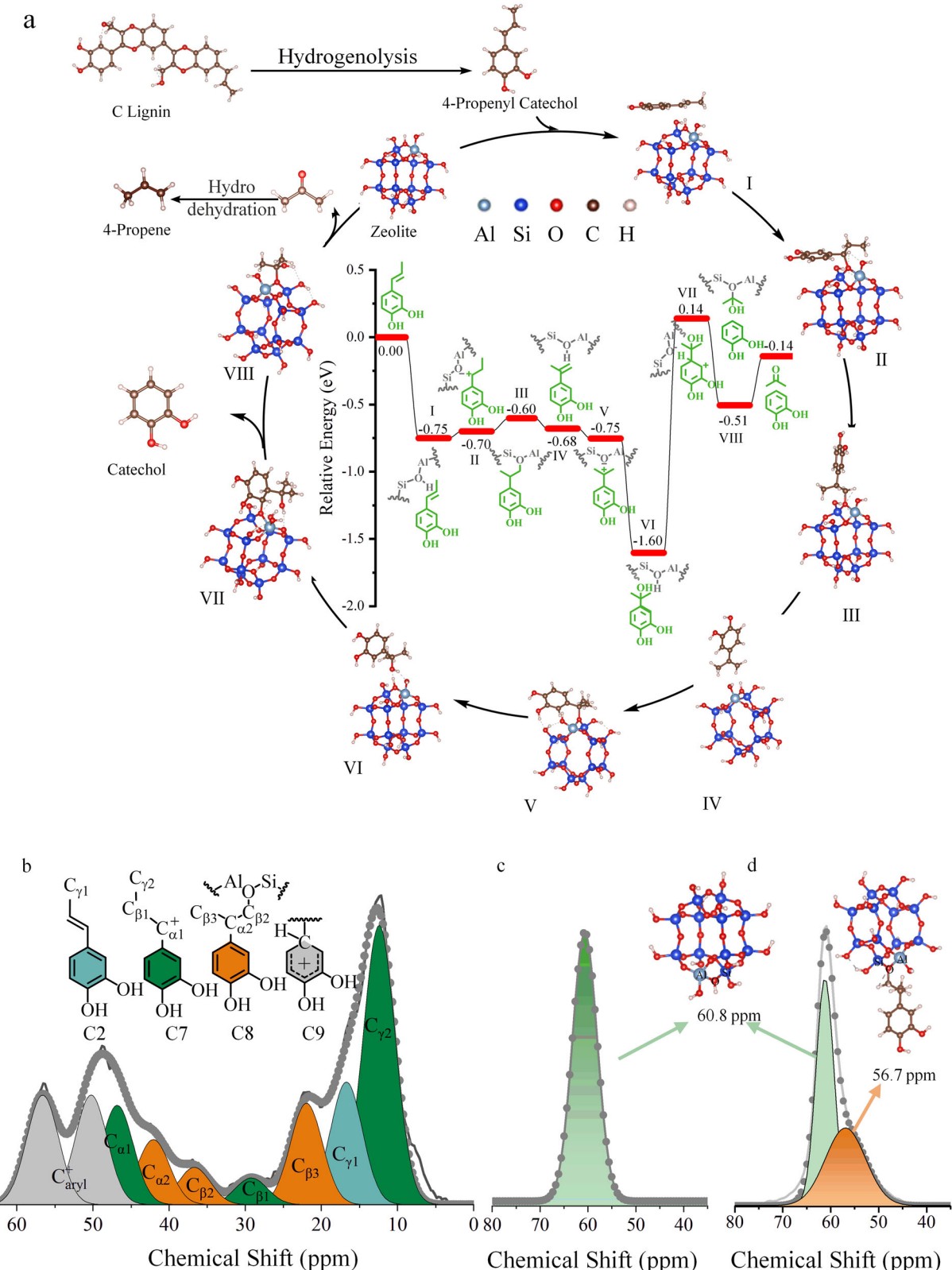

**Fig. 6 | Mechanistic studies. a** Energy profile for the transformation of the $C_{aryl}$–$C_\alpha$ bond in C2 on the $HY_{30}$ catalyst; **b** $^{13}C$ NMR spectrum of the intermediates in the reaction of C2 compound; a thin gray line represents the observed intensity, a solid gray ball is the total fitting curve, dark green, light green, yellow and grew areas represent the C2, C7, C8, and C9, respectively. **c** $^{27}Al$ NMR analysis of the initial $HY_{30}$ zeolite; the gray line represents the observed intensity, and the light orange area is the fitting curve of framework Al(IV). **d** $^{27}Al$ NMR analysis of the $HY_{30}$ zeolite in the reaction; the gray line represents the observed intensity, and the light orange area is the fitting curve of framework Al(IV), purple area is the fitting curve of the framework Al(IV) in the [Al–O($C_{\beta3}$)–Si] center. Reaction conditions: 50 mg C2 compound, 100 mg $HY_{30}$, 5 mL $H_2O$, 180 °C, 0.1 MPa Ar, 5 min.

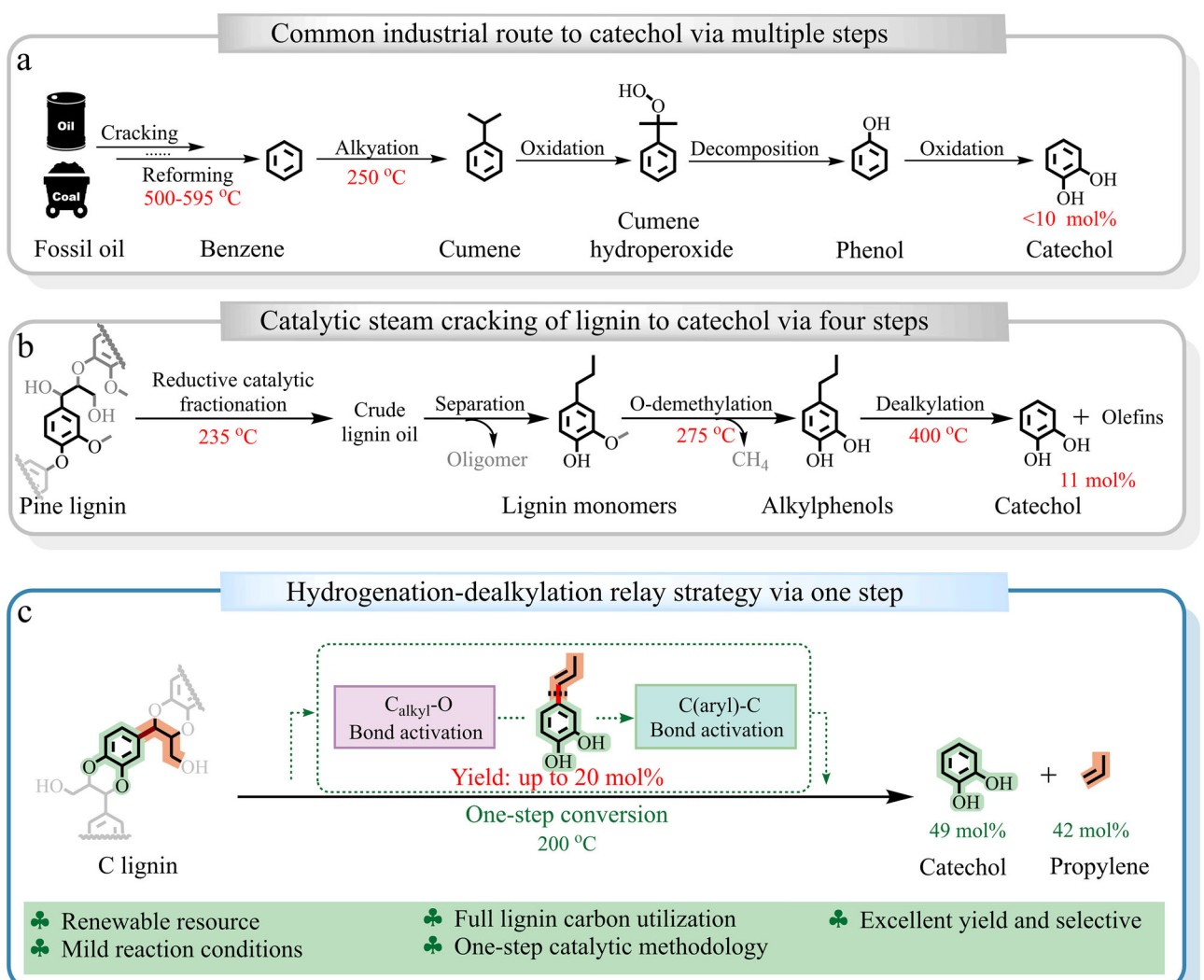

**Fig. 7 | Strategies for the production of catechol and propylene. a** Production of catechol from fossil resources via the common industrial route; **b** Catalytic steam cracking of lignin monomers to catechol via four steps; **c** The sustainable production of catechol and propylene via the one-pot hydrogenolysis-dealkylation cascade catalysis of C-lignin (This work).

pathway involved in this efficient dealkylation transformation of the C-lignin. The successful conversion is dependent on the nature of catalysis, the high chemical reactivity of the Brönsted acid sites, and the accessibility of the HY$_{30}$ zeolite framework, which can reduce the reaction activation energy barrier of the stable C$_{aryl}$–C$_{alkyl}$ after the delicate protonation and migration-transformation of the side chain in various 4-propenylcatechol-derivated intermediates[62,63].

Nowadays, catechol and propylene are two essential and basic components in the modern chemical industry and organic synthesis for producing a wide range of chemicals and materials[64,65], which have an ever-growing global demand. Besides the current gap between the annual demand and production, the traditional production of catechol and propylene relies heavily on fossil resources and has inherent disadvantages, such as severe conditions, low conversion range, environmental pollution, and high CO$_2$ emission (Fig. 7). Therefore, the more sustainable and efficient synthetic route of catechol and propylene with renewable and nonedible C-lignin as the feedstock has excellent potential.

Based on our above experimental results, an industrial-scale process model has been designed and simulated to evaluate the practical potential of this strategy for green catechol production from C-lignin biomass. The model involves lignocellulose pretreatment, dealkylation reaction, and separation units (Fig. 8a). Based on reaction conditions and inventory, a rigorous material and energy balance was obtained through a steady-state simulation using Aspen Plus (Supplementary Figs. 1–3, Supplementary Tables 1–7). The techno-economic analysis (TEA) and life cycle assessment (LCA) were subsequently carried out to evaluate the economic, energy, and climate performance of bio-catechol synthesis. The results show that the total production cost (TPC) of bio-catechol in our system (2.36×10$^4$ CNY/t) is currently unattractive, which was 133% higher than that of mature fossil-based catechol production (1.01×10$^4$ CNY/t) (Fig. 8b and Supplementary Table 8) due to the high energy consumption in biomass pretreatment. Hence, to diminish energy consumption, solvent extraction (60.1%, Supplementary Table 8) was employed to extract C-lignin, and the ensuing hydrogenolysis-dealkylation process for catechol and propylene production showed comparable yields to those obtained using BME process (Supplementary Fig. 16). The TPC of bio-catechol via solvent extraction was about 1.13×10$^4$ CNY/t, which was slightly higher than that of mature fossil-based catechol production. Thus, further research should concentrate on cost-effective pretreatment for lignin extraction, which can substantially reduce the energy usage of green catechol production.

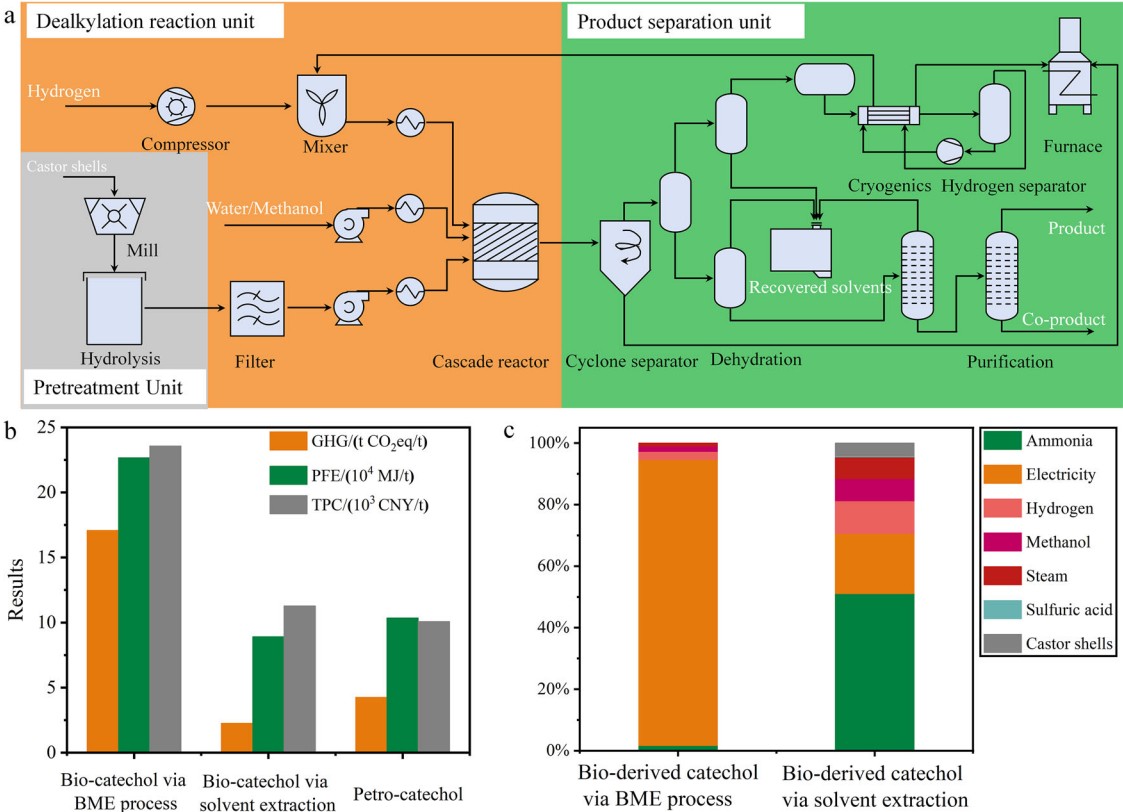

**Fig. 8 | TEA and LCA of catechol production. (a)** Conceptual model of catechol production from raw biomass, including key stages such as biomass pretreatment, catalytic dealkylation, and product separation. **(b)** Comparison of the economic (TPC, in $10^3$ CNY/t) and environmental (GHG emissions, in t $CO_2$ eq/t, and PFE depletion, in $10^4$ MJ/t) performance of bio-based and fossil-based catechol

production **(c)** Detailed breakdown of life cycle GHG emissions for bio-derived catechol using the BME process (left) and solvent extraction method (right), including contributions from biomass pretreatment, reaction, and product separation. Note: BME = ball-milling and enzymatic hydrolysis.

The LCA results reveal that PFE depletion and GHG emissions of bio-derived catechol via ball-milling and enzymatic hydrolysis (BME) compared to fossil-based catechol were analyzed, finding that biomass pretreatment plays a significant role in both environmental and economic performance, a trend similar to the findings in the TEA analysis (Fig. 8c and Supplementary Table 9-11). In the LCA, biomass pretreatment is the main contributor to PFE depletion and GHG emissions, while in the TEA, it is the primary factor leading to the high production cost of bio-derived catechol. In contrast, the life cycle PFE depletion and GHG emissions show a fascinating result of bio-catechol via solvent extraction. The life cycle PFE depletion and GHG emissions of bio-derived catechol via solvent extraction are 14% and 46% less than those of mature fossil-derived catechol. Similarly, for the bio-catechol via solvent extraction, contributions of 51.11% and 19.48% to the life cycle GHG emissions are responsible for ammonia and electricity, which are mainly derived from biomass pretreatment. Therefore, biomass pretreatment is the main barrier to reducing GHG emissions, while the catechol/propylene production process shows excellent potential in reducing GHG emissions, contributing to a low-carbon society.

Globally, castor beans are recognized as an important industrial resource, with India, Mozambique, Brazil, and China leading in production, achieving a total yield of $1.86×10^6$ tons in 2021 based on the data in the net of Global trading platform Tridge (https://www.tridge.com/intelligences/castor-bean-castor-seed/production). The oil content within castor beans, which ranges between 46% to 55%, renders them an ideal source for high-quality bio-oil production, widely applied in diverse industrial domains. Beyond the seeds, the endocarp of the castor coats, a by-product from the castor processing industry,

based on compositional analysis, is estimated to yield about $2.50×10^5$ tons of C-lignin. This work introduces a hydrogenolysis-dealkylation cascade catalysis strategy for converting C-lignin into high-value chemicals with remarkable efficiency. Under optimized conditions, 1 ton of C-lignin is capable of yielding 190 kg of catechol and 65 kg of propylene. Using the endocarp of castor seeds, a by-product of the global castor processing industry, there's a potential to produce around $4.76×10^4$ tons of bio-based catechol and $1.68×10^4$ tons of bio-based propylene. Notably, with the global production of catechol in 2021 standing at about $4.4×10^4$ tons, the output from this innovative process not only completely satisfies but has the potential to exceed global catechol demands (https://www.gep.com/blog/mind/catechol-faces-supply-crunch-while-prices-rise). This signifies a groundbreaking achievement in meeting global chemical demands through a sustainable, low-carbon approach by utilizing agricultural by-products. Furthermore, the process yields a substantial quantity of propylene, enhancing the overall value and economic viability of this technological application. This research not only showcases an efficient method for transforming agricultural by-products into valuable chemicals but also emphasizes the capability of this technology to fully meet global catechol demands in an environmentally friendly manner, offering new perspectives and solutions for the cyclical utilization of global resources and advancing sustainable development.

## Discussion
In summary, with the standardization transformation of lignin in mind, we report the selective and efficient production of bio-catechol and bio-propylene from the ideal C-lignin via a one-pot hydrogenolysis-dealkylation cascade catalysis. The Ni NPs and $HY_{30}$ zeolite in the

optimized catalyst Ni/HY$_{30}$ acted in synergy to orderly cleave the corresponding C−OAr bonds and C$_{aryl}$−C$_{alkyl}$ bonds in the uniform benzodioxane units of C-lignin, which could selectively provide a 49 mol% yield of catechol and a 45 mol% yield of propylene from C-lignin under 200 °C. Besides the remarkable monomer yield, the 96% selectivity to single catechol and the close to 100% selectivity to propylene in the current biomass route could significantly surpass the traditional preparation methods from fossil resources, which indicates a promising potential for industrial-scale production. Based on the LCA and TEA analysis, this strategy demonstrates the potential of utilizing sustainable lignin to produce renewable, low-carbon chemicals. While the production cost of bio-based catechol and propylene is currently higher than fossil-based counterparts at the laboratory scale, these bio-based products offer significant environmental benefits, such as reduced greenhouse gas (GHG) emissions and fossil energy consumption. With further optimization and scale-up, this approach is expected to become profitable and contribute to a more atom-economic and sustainable society. Furthermore, the control experiments, spectra characterizations, and DFT calculations indicated that the side chain modification in the intermediate 4-propenylcatechol, obtained from the selective hydrogenolysis of C-lignin, was exceedingly convenient for HY$_{30}$-triggered dealkylation, and the side chain was delicately deconstructed via the BAS-mediated protonation, γ-methyl migration, and C$_{aryl}$−C$_{alkyl}$ scission pathway. Finally, the corresponding strategy design based on the concept of standardization transformation and mechanism revelatio,n focusing on the cleavage of critical linkage bonds, could provide guidance for further lignin depolymerization utilization.

## Methods

### C-lignin Characterization

2D-HSQC NMR spectra of C-lignin were collected on a Bruker AVIII 400 MHZ spectrometer, and 60 mg C-lignin was dissolved in the 0.5 mL DMSO-d$_6$. The ether linkage in C-lignin was quantified via quantitative $^{13}$C NMR. 110 mg C-lignin, 3.12 mg 1, 3, 5-trioxane as an internal standard, and 2 mg Chromium (III) acetylacetonate as a relaxation reagent were dissolved in the 1 mL DMSO-d$_6$. The quantitative $^{13}$C NMR spectra were collected in the FT mode at 100.6 MHZ, and the inverse-gated decoupling sequence (C13IG) was used. The average molecular weight of C-lignin was determined by gel permeation chromatography (Shimadzu LC-2030plus) equipped with a UV detector at 254 nm on a phenogel 300 × 7.8 mm, which was calibrated with polystyrene standards.

### Synthesis of M/zeolite

The support Ni/zeolite, Pd/zeolite, and Ru/zeolite catalysts were prepared by the wet impregnation method. Typically, 500 mg HY$_{30}$ zeolite powders were dispersed in 20 mL of deionized water. After stirring for 4 h, 10 mL NiCl$_2$·6H$_2$O solution (8.5 mmol/L) was added drop-wise to HY$_{30}$ dispersion under stirring. Then, the NaBH$_4$ solution was added slowly to the above mixture in the ice water bath under an argon atmosphere. The reaction mixture kept on going at room temperature for 2 h to complete the reduction of the metal ion. The resulting gray granules were separated by centrifuging and washing with ultrapure water and ethanol several times and were dried in a vacuum oven at 60 °C for 24 h. Pd/zeolite and Ru/zeolite were prepared by the same method.

### Dealkylation of C-lignin

The dealkylation reaction was performed in a Teflon-lined stainless-steel reactor of 20 mL with a magnetic stirrer. In a typical experiment, a suitable amount of C-lignin, catalyst, and solvent were loaded into the reactor. The reactor was sealed, purged with N$_2$, and pressured to H$_2$ (3 MPa) at ambient temperature. Then, the reactor was placed in a furnace at a desired temperature for a certain time.

After the reaction, the reactor was placed in ice water, and the gas released was collected in a gasbag. The gas sample was analyzed by a GC (Shimadzu−2014) equipped with FID and TCD detectors and carbon molecular sieve TDX-01 (2.0 m × 2.1 mm) and R-N columns (0.3 m × 0.1 mm) using argon as the carrier gas. The resulting liquid was obtained by filtration and was extracted with three-fold ethyl acetate and water. The solvent mixture was evaporated, and a known amount of n-dodecane as the internal standard was added into the resulting liquid with anhydrous THF and pyridine. The mixture was treated with bis(trimethylsilyl)trifluoroacetamide (BSTFA) at 70 °C for 60 min. Then, the treated mixture was analyzed using a GC-MS (Agilent 7890A-5975C, HP-5MS capillary column (30 m × 0.25 mm × 0.25 μm)) and GC (Agilent 7890 A, HP-5 capillary column (30 m × 0.25 mm × 0.25 μm)). The qualitative and quantitative analysis of catechol monomers in the resulting liquid was assessed by comparison with authentic samples from independent synthesis. The detailed calculation was as follows

$$Y_{monomer} = \frac{n_{monomer}}{n_{CA}} \times 100\% \tag{1}$$

$$n_{CA} = Y_{CA} \times m_{C-lignin} \tag{2}$$

$$S_{monomer} = \frac{n_{monomer}}{\sum n_{monomer}} \times 100\% \tag{3}$$

In the equations, $Y_{monomer}$ (%) is the yield of monomer based on the molar amount of caffeyl alcohol in the C-lignin; $n_{monomer}$ (mmol) is the molar amount of monomer in each analyzed sample, $n_{CA}$ (mmol) is the molar amount of caffeyl alcohol in the C-lignin; $Y_{CA}$ (mol/mg) is the mole amount of caffeyl alcohol per milligram of C-lignin form the quantitative $^{13}$C NMR analysis (Supplementary Note 2); $S_{monomer}$ (%) is the selectivity of monomer; $\sum_{monomer}$ (mmol) is the sum of the molar amount of all detectable products.

### Dealkylation of catechols

The dealkylation reaction was performed in a Teflon-lined stainless-steel reactor of 20 mL with a magnetic stirrer. In a typical experiment, a suitable amount of catechols, catalyst and solvent were loaded into the reactor. The reactor was sealed, purged with N$_2$ and subsequently charged with 0.1 MPa of Ar at ambient temperature.

After the reaction, the reactor was placed in ice water, and the gas released was collected in a gasbag. The gas sample was analyzed by a GC (Shimadzu−2014) equipped with FID and TCD detectors and carbon molecular sieve TDX-01 (2.0 m × 2.1 mm) and R-N columns (0.3 m × 0.1 mm) using argon as the carrier gas. Meanwhile, the resulting liquid was obtained by filtration and was extracted with three-fold ethyl acetate and water. The solvent mixture was evaporated, and a known amount of n-dodecane as the internal standard was added to the resulting liquid with anhydrous THF and pyridine. The mixture was treated with bis(trimethylsilyl)trifluoroacetamide (BSTFA) at 70 °C for 60 min. Then, the treated mixture was analyzed using a GC-MS (Agilent 7890A-5975C, HP-5MS capillary column (30 m × 0.25 mm × 0.25 μm)) and GC (Agilent 7890 A, HP-5 capillary column (30 m × 0.25 mm × 0.25 μm)). The qualitative and quantitative analysis of catechol monomers in the resulting liquid was assessed by comparison with authentic samples from independent synthesis.

### Transformation of acetone

The transformation of acetone was performed in a Teflon-lined stainless-steel reactor of 20 mL with a magnetic stirrer. In a typical experiment, 100 μL acetone, 50 mg 0.97 wt% Ni/HY$_{30}$, and 5 mL methanol/water mixture (1:4) were loaded into the reactor. The reactor was sealed, purged with N$_2$ and pressured to H$_2$ (3 MPa) at ambient

temperature. After the reaction, the reactor was placed in ice water, and the gas released was collected in a gasbag. The gas sample was analyzed by a GC (Shimadzu−2014) equipped with FID and TCD detectors and carbon molecular sieve TDX-01 (2.0 m × 2.1 mm) and R-N columns (0.3 m × 0.1 mm) using argon as the carrier gas. Meanwhile, the resulting liquid was analyzed by HPLC (Agilent 1260 Infinity, equipped with HPX-87H column (300 × 7.8 mm, Bio-Rad, USA) and reflective index detector) with diluted $H_2SO_4$ solution as mobile phase.

### Detection of intermediates

The detection of intermediates was carried out in a Teflon-lined stainless-steel reactor of 20 mL with a magnetic stirrer. In a typical experiment, 4-(1-propenyl)-catechol (50 mg), Ni/HY$_{30}$ (100 mg) and $H_2O$ were loaded into the reactor. The reactor was sealed, purged with $N_2$ and subsequently charged with 0.1 MPa of Ar at ambient temperature. Then, the reactor was placed in a furnace and heated to 180 °C. After 5 min, the reactor was quickly transferred to a liquid nitrogen bath. When the reaction mixture was frozen, the gas was released immediately. The reaction mixture was freeze-dried under a vacuum for 12 h to remove the water. The detection of the intermediates in the dried sample was conducted by $^{13}C$ NMR, solid-state 2D $^{13}C$-$^1H$ dipolar-mediated HETCOR, $^{27}Al$ MQMAS, and $^{27}Al$ NMR analyses on Bruker Avance III 400.

### Life cycle assessment (LCA)

LCA as an effective tool to evaluate environmental performances, is used to estimate primary fossil energy (PFE) depletion and greenhouse gas (GHG) emissions of catechol production from castor shells herein. LCA is defined as the "compilation and evaluation of the inputs, outputs, and potential environmental impacts of a product system throughout its life cycle"[66]. Hence, the system boundary covers PFE consumption and GHG emissions to produce catechol and all upstream of feedstock, required materials, and utilities. Herein, the functional unit is defined as 1 metric ton of catechol. GHGs, including $CO_2$, $CH_4$, and $N_2O$, are estimated over a 100-year time horizon and normalized in units of kg $CO_2$ equivalent[67]. Note that the carbon element in biomass should be reduced because it is actually sourced from $CO_2$ in the atmosphere. The PFE consists of primary energy coal, petroleum, and natural gas, which is measured based on the corresponding lower heating value (LHV). GHG and PFE are calculated as follows:

$$GHG = \sum(E_{CO_2} + 25 E_{CH_4} + 298 E_{N_2O}) - C_{bio} \quad (4)$$

$$PFE = \Sigma(LHV_{coal} C_{coal} + LHV_{petro} C_{petro} + LHV_{gas} C_{gas}) \quad (5)$$

where $E_{CO2}$, $E_{CH4}$, and $E_{N2O}$ represent emissions of $CO_2$, $CH_4$, and $N_2O$ in each step of the life cycle model, respectively, kg; $LHV_{coal}$, $LHV_{petro}$, and $LHV_{gas}$ are the lower heating value of coal, petroleum, and natural gas, MJ/kg; $C_{coal}$, $C_{petro}$, and $C_{gas}$ are consumption amount of coal, petroleum, and natural gas, kg.

Castor shells are converted into catechol with the proper atmosphere, solvent, catalyst, and utilities. The life cycle of the catechol includes castor shell collection and catechol extraction from castor shells. Herein, castor shells are generally seen as a waste of castor. Thus, castor farming and harvesting are excluded in this study. Atmosphere ($H_2$) and solvent (water and methanol) are considered by loss or supplement of materials in the process. The utilities include electricity and steam (heat). Electricity is mainly consumed in the ball milling step. Meanwhile, pressure changes are another important source of electricity consumption. Heat is consumed for the separation step. Supplementary Fig. 3 shows the system boundary of the catechol.

GREET 2020 software is taken to develop model and link units[68]. The inventory of transformation from castor shells to catechol was simulated with Aspen Plus, which is shown in Supplementary Table 4 as mentioned above. Production inventory of hydrogen, water, methanol, electricity, and steam was taken from an inherent database of GREET software. Note that the electricity source was selected as China's power grid in 2022. Life cycle PFE and GHGs were obtained by incorporating estimated material and energy into GREET finally.

In addition, the LCA of catechol via the conventional route is used as a reference to compare the environmental benefit of the catechol. The production inventory of conventional catechol is shown in Supplementary Table 6, and background data, such as the production of phenol and hydrogen peroxide, were taken from the publication[69]. Other data was paralleled with the castor shells-derived catechol process.

### Techno-economic analysis (TEA)

The techno-economic analysis of this work has been conducted to evaluate total capital investment (TCI) and TPC. The TCI was calculated based on the installation costs of all equipment. Herein, installation costs of all pieces of equipment were estimated with the Aspen Process Economic Analyzer based on the calculated flow rates and experimental information such as acidity and the employed residence times. All capital investments of the equipment were adjusted to 2022 CNY with a location factor of 0.61[70] and the Chemical Engineering Plant Cost Index (CEPCI)[71].

The TPC includes raw materials, utilities, operating and maintenance, depreciation, plant overhead, administrative, distribution, and selling costs[71]. The detailed parameters and assumptions for the estimation of TCI and total product cost are depicted in Supplementary Table 7.

## Data availability

All data needed to evaluate the conclusions in the paper are present in the paper and/or the Supplementary Information. Additional data available from authors upon request.

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

## Acknowledgements

This work was supported by the National Natural Science Foundation of China (22308029 (X.S.), 22025206 (F.W.), 21721004 (F.W.), 32471809 (C.Z.) and 22308240 (Z.Z.)), the Research Fund for High-level Talents Introduction of Nanjing Forestry University (163105164) (C.Z.), the Liaoning Revitalization Talents Program (XLYC2002012) (F.W.) and the Excellent Research Assistant Funding Project, Chinese Academy of Sciences (CAS)(X.S.), the Fundamental Research Funds for the Central Universities (200-BLX202234) (X.S.) and the 5·5 Engineering Research & Innovation Team Project of Beijing Forestry University (No. BLRC 2023B05) (T.Y.). Additionally, the authors appreciate the assistance of the Innovation Platform for High-Value Utilization of Forest Resources at Beijing Forestry University.

## Author contributions

X.S., C.Z., T.Q.i.Y. and F.W. conceived and designed the project; Z.Z. carried out the techno-economic analysis and life-cycle assessment and wrote the manuscript; J. W., J. Z., Y.L., T.Q.i.Y., and F.W. advised on experiments and participated in data analysis and discussions; Yi Ji and G.H. completed solid NMR analyses in this work. X.S., C.Z. wrote the manuscript. All authors have approved the final version of the manuscript.

## Competing interests

The authors declare no competing interests.
