## [Peer review file · Nature Communications]

Standardization Transformation of C-Lignin to Catechol and Propylene

Corresponding Author: Professor Chaofeng Zhang

Version 0:

Reviewer comments:

Reviewer #1

(Remarks to the Author)

See attachment

Reviewer #2

(Remarks to the Author)

Reviewer #3

(Remarks to the Author)

The submitted manuscript of Shen, Wang et al. describes a novel method to produce industrially relevant catechol and propylene using C-lignin. The reaction is done via a one-pot hydrogenolysis-dealkylation relay catalysis using Ni/HY30 catalyst. Some mechanistic studies were done and were supported by the detection of a reaction intermediate (4-propenylcatechol) and DFT calculations. Furthermore, TEA and LCA studies of the process showed its potential to reduce the environmental impact when benchmarked with the current fossil-based production technology. We think that the work is well written in a methodological way, the references are relevant and updated and the figures are clear. All the evidence is presented in the main text as well as the SI to support the claims and conclusions and to permit a reproduction of the work. While the one pot reaction is new nevertheless it is a combination of two well-known and extensively studied reactions (hydrogenolysis and dealkylation). The originality and applicability of the results to the audience of Nature Communications journal is questionable since the process is done on a very specific lignin (C-lignin) and not a lignin available in big volumes. Moreover, the upscaling of such reaction in a cascade reactors is not as easy as depicted. Below some questions and suggestions that will help improving the quality of the manuscript.

- The authors tried their one pot reaction of hydrogenation and demethylation with C-lignin nevertheless this lignin is very specific and with a maximum estimated volume of 0.25 Mtons. Does this same one pot strategy work on other types of lignins available in higher volumes? More specifically on technical lignins?
- The one-pot hydrogenolysis-dealkylation relay catalysis looks very interesting from an academic point of view. Although in the proposed process in figure 7, the continuous processing of solids especially before the filter and inside the cascade reactors handling multiple phases is not straight forward. More information related to the proposition of the units in the process is required. Fouling will for sure occur in the filter! The pumping of slurries is not very easy to conduct... Kindly provide more information on the topics.
- Solvent extraction proved to be the best method for lignin extraction from economic and environmental points of view yet, the authors did not comment on the lignin quality produced by this method as well as its yield. This is crucial to get a credible assessment of the process. Furthermore, we recommend the authors to try their process on a C-lignin obtained by solvent extraction to highlight the differences. Similarly, the authors do not comment on the solid products that remain after the extraction of lignin. How to valorize these?

- Typo page 8 line 162, please correct: ...dealkylation normalization with stepwise...

Reviewer #4

(Remarks to the Author)

This study reports on the valorization of lignin, a renewable feedstock, for the production of bio-catechol. It includes performing techno-economic analysis (TEA) and life cycle assessment (LCA) to evaluate economic feasibility and environmental impacts. Here are some comments:

1. In the "Economic and Environmental Analysis" section, the authors need to provide more detailed results. It is also advisable to separate the TEA and LCA into two distinct paragraphs.
2. Lines 480 – 482: "The results show that the total production cost (TPC) of bio-catechol in our system is currently unattractive, which was 133% higher than that of mature fossil-based catechol production (Fig. 7b) due to the high energy consumption in biomass pretreatment." It would be clearer if the TPC (i.e., in CNY??/kg) for cases was also given in the sentence, not just "133% higher."
3. Lines 488 – 490: "The LCA results reveal that PFE depletion and GHG emissions of bio-derived catechol via ball-milling and enzymatic hydrolysis compared to fossil-based catechol were analyzed, finding comparable trends in cost analysis (Fig. 7b)." Dis authors try to suggest that the trends in the LCA and TEA results are similar?
4. Lines 537 – 541: "Based on the LCA and TEA analysis, this strategy reveals the holistic utilization of sustainable lignin to produce profitable, renewable, and low-carbon footprint chemicals simultaneously, and such bio-based catechol and propylene preparation from renewable natural resources can significantly contribute to the transformation into a more atom-economic and sustainable society." Figure 7b does not support the statement's claim. The "Bio-catechol via BME process" scenario appears to be significantly worse than the "Petro-catechol" scenario.
4. The statement, "The total capital investment was calculated as the sum of the installation costs of all equipment" on Page 13 in the Supplemental Materials is incorrect. TCI should encompass more than just the installation costs. The statement is also inconsistent with the information presented in Supplementary Table 7.
5. Figure 7's captions need enhancement to include more details and descriptions. The units in Figure 7(b) are unclear. Are the quantities measured per kg or per metric ton of catechol?
6. TEA and LCA results shown in Figures 7b and 7c should be provided in the Supplemental Materials, summarized in tables.
7. The statement "The production inventory of conventional catechol is presented in Supplementary Table 6" found on Page 13 of the Life Cycle Assessment (LCA) deserves further scrutiny. Please review Supplementary Table 6, as its title lacks clarity.
8. Provide LCA input data, i.e., on raw material consumption, energy input, products, coproducts, waste, air emissions, and more, for the LCA study in the Supplemental Materials.

Reviewer #5

(Remarks to the Author)

This manuscript reports a one-pot catalytic transformation of C-lignin to catechol and propylene using an HY zeolite-supported Ni catalyst. The study also includes computational modeling to explore the reaction mechanism, as well as a life cycle assessment of the process.

The manuscript has significant issues that hinder its readability and scientific clarity. The English language requires substantial improvement, as many sentences are unclear and difficult to understand. Additionally, the authors have not adhered to standard technical terminology, opting instead to introduce unconventional terms such as "standardization transformation" instead of the widely accepted "selective transformation."

The manuscript also suffers from structural weaknesses. The introduction includes some results, which is unconventional, and there is no experimental section in the main manuscript, nor are readers directed to the ESI for these details. Furthermore, the novelty of the work is not clearly articulated in the abstract or conclusion. Data presentation and discussion are inadequate throughout the manuscript. For example, the authors tested different metal catalysts but did not properly discuss the results. Similarly, solvent optimization was studied, yet the findings are only briefly mentioned and not analyzed in detail. Another example is that while the impact of the Si/Al ratio of the zeolite support was investigated, these results are neither clearly presented nor adequately discussed. Multiple supports are mentioned in the ESI, but no discussion of these appears in the main manuscript. Figure 1 also contains inconsistencies between its content and caption. These are just a few examples, but similar issues can be found throughout the manuscript, affecting the manuscript as a whole and making it difficult to extract meaningful insights. Additionally, some of the procedures reported in the ESI lack sufficient detail to allow replication by other researchers. The discussion across all parts of the manuscript, including the experimental work,

computational modeling, and life cycle assessment, is superficial and lacks sufficient depth. Furthermore, the use of a high catalyst-to-substrate ratio in the experiments, where twice as much catalyst as substrate was used, raises questions about the practical applicability of the process.

Overall, the manuscript appears to be a preliminary draft that requires extensive revision to reach publication quality. I recommend that the authors address the above issues before resubmitting. At this stage, I recommend rejection.

Reviewer #6

(Remarks to the Author)

Wang has conducted an excellent piece of work that actively responds to the critical and scientifically valuable hot topic of carbon peaking and carbon neutrality as part of the shared future for mankind. The degradation of lignin is also a current hot scientific issue, with many previous studies highlighting the significant difficulties and challenges associated with breaking the C(Aryl)-C(Alkyl) and C(Alkyl)-O bonds during lignin degradation. The author has successfully degraded lignin into aromatic compounds such as catechol and propylene using the Ni/HY30 catalyst, which brings significant value to the field of small-molecule chemicals.

This work also demonstrates a certain level of methodological expertise and applicability, and proposes a reasonable mechanism through a series of mechanistic experiments and catalyst characterization. Through a series of mechanistic studies, a reasonable catalytic mechanism and model have been proposed, and the industrial production application prospects and value of this method have been demonstrated.

This reviewer recommends the publication of this manuscript in Nature Communications. However, there are still some minor issues that need to be addressed.

1) Scheme 1 mentioned in the article does not correspond to the figure, and there are two Figure 1 in the article. Additionally, the condition optimization discussed in the text (Figure 1) does not match the figure described.

2) The author conducted a series of mechanistic experiments to verify the rearrangement mechanism of the reaction process (Figure 3) and proposed several possible rearrangement intermediates. Can these intermediates be confirmed through detection during the reaction process?

3) The conversion and yield mentioned in Figure 3 are somewhat confusingly presented. Moreover, the directivity of the illustration in Figure 3 is not high. The authors should revise Figure 3 as well as the accompanying text description.

4) Regarding whether the amount of catalyst used can be further reduced while still achieving a suitable reaction effect, the authors should provide the related results.

Version 1:

Reviewer comments:

Reviewer #1

(Remarks to the Author)

See attached file. Overall, most issues were addressed. Some minor issues remain (especially the writing needs improvement) before it can be published.

Reviewer #2

(Remarks to the Author)

Reviewer #3

(Remarks to the Author)

We think that the reviewers addressed all our concerns and have implemented the suggestions in the manuscript. Accordingly, we think that the manuscript can be accepted for publication in Nature Communications.

Reviewer #4

(Remarks to the Author)

Upon reviewing the revisions, it is evident that the authors have satisfactorily addressed the comments in the "Economic and environmental analysis" section and made the necessary improvements to strengthen the manuscript.

Reviewer #5

(Remarks to the Author)

I find that the manuscript still lacks the level of novelty and depth required for publication. The authors have not sufficiently addressed key concerns raised in previous rounds, and the manuscript remains poorly written, with limited scientific insight. The discussion is superficial, and the overall contribution to the field is minimal.

Reviewer #6

(Remarks to the Author)

The authors have thoroughly addressed all the concerns I previously raised. I recommend the revised manuscript for publication in Nature Communications.

Reviewer #1

The topic of one-pot catechol and propylene production from “relay” hydrogenolysis-dealkylation of ideal C-lignin is really interesting and exhibits high selectivity to the selected products, which seems very promising as an alternative or supplement to traditional catechol and propylene processes from fossil refinery and serve as a model for lignin valorization. The developed catalytic strategy to directly obtain catechol in one step with good yields from lignin is a significant breakthrough in the field of lignin valorization, which makes this contribution highly significant to this field. The whole approach is of interest to the wider community as the authors have conducted a comprehensive mechanism investigation (model compound reactions, catalyst characterization, and DFT calculations) to unravel the mechanism in terms of how the “relay” hydrogenolysis-dealkylation is realized by a synergistic effect of nickel and Brønsted acid species on the zeolite support as well as TEA and LCA to reflect the potential implementation. However, there are some suggestions for further major revisions of this article needed for it can be considered publication in nature communications:

Response: Thank you for your approval of our manuscript. We are most grateful for your thoughtful and penetrating comments. We have considered all issues in the comments and answered point by point. At the same time, we revised the manuscript carefully according to the comments and made all results and analyses clear. We hope that our revision meets your expectations.

Comment 1. Errors and language

There are several misspellings (e.g. inerte line 115 (more inert), BDE line 178 and “delay catalysis” line 175,), wrong figure references (e.g. 2x figure 1 and a,b,c etc are wrongly referred to in the text for the second figure 1) and strange word use.

Response: Thanks very much for these suggestions. After careful checking, some typos and wrong figure references in the revised manuscript and figures have been revised.

A few examples:

Comment 1a. It is unclear what is meant with “standardization transformation”. This is not a commonly used term in the field. I do understand what this means and I would suggest something like product funneling or simply selective depolymerization and defunctionalization or “stepwise deconstruction”

Response: Thank you for your suggestion. As discussed in the manuscript, besides the difficulties in the lignin transformation via the efficient cleavage of the C–C/C–O linkage bonds, the oxygen-containing groups make lignin-derived monomers have a high polarity and boiling point, and the physicochemical differences among the monomers with different aromatic nuclei (H-, G-, or S-unit) and side chains skeletons but similar functional groups are not obvious, which make the practical application of aromatic chemical preparation from lignin still faces challenges. Given the development of catalytic systems that have provided high enough yields of aromatic monomers, the efficient separation of various monomeric products, especially for certain kinds of high value-added phenolic products, from lignin depolymerization mixture has become the most prominent issue.

It is necessary to put forward new ideas to integrate and optimize the lignin depolymerization, which could efficiently provide high-valuable monomer products and simultaneously reduce the stress of the separation procedure. Referring to the standardization concept from industrial manufacture, which involves the standardized substrate and standardized productive process to the standardized product, the standardization transformation of lignin to high-value-added chemicals (standardized product) requires precise regulation of the reaction process based on the elaborate catalytic strategy design (**Standardized Process:** It not only includes your suggested product funneling, defunctionalization/stepwise deconstruction methods to provide a single product after losing the functional groups but also could contain the methods to provided a single product by adding the functional groups, such as Yan's work about the production of terephthalic acid from corn stover lignin; *Angew. Chem. Int. Ed.* 2019, 58 (15), 4934-4937.) and lignin structure optimization (**Standardized Substrate:** According to the above discussion, the ideal lignin as the aromatic chemicals preparation is the one from single monolignol or the ratio of one kind of monolignol can be as high as possible to the one with the absolute advantage. In addition, considering that the subsequent formation of lignin polymers by monolignols is a free radical polymerization process, in other words, even the single monolignol can provide various linkage types, if a higher proportion or only the easily transformable chemical bond connections can be maintained in the structure, such as β -O-4 and α -O-4, the lignin conversion will be more efficient. An understanding can be obtained that controlling the complexity of the lignin substrate from the source, that is the predetermined plant structure, can effectively simplify the subsequent lignin catalytic

depolymerization and increase the corresponding product yield. Compared with high-S lignin, C-lignin is the perfect alternative with a single aromatic unit and a single linkage structure). For the standardization transformation of lignin, we once mentioned in our previous book (Chapter 12, Zhang, C. F.; Wang, F. Lignin Conversion Catalysis: Transformation to Aromatic Chemicals; Wiley, 2022) as the perspective for the following lignin transformation.

Therefore, to overcome the critical separation challenge of products from lignin catalytic depolymerization, the multi-step tandem process, or even the elaborate one-pot catalytic procedure, is the most potential choice to provide standardized products, which means a narrow or even single product distribution with the directed elimination or addition of substituent groups. It was your suggestion about the product funneling or simply selective depolymerization and defunctionalization or “stepwise deconstruction”. Furthermore, besides the reaction regulation and product control or further refining, the standardization orbiting the structure of the lignin substrate could also provide potential solutions for the separation challenge. In this work, the C-lignin was the ideal standardized lignin substrate. In another aspect, the lignin standardization transformation also targets to produce the important standard chemical products, which were bio-catechol and bio-propylene in this work from the standardized C-lignin.

The lignin standardization transformation highlights the importance of the transformation target, substrate structure, transformation system/procedure, product structure control, and substrate separation for the lignin refining. We hope we can make a detailed explanation in the revised manuscript to support this point.

Comment 1b. I think “relay catalysis” is not the right wording. This is usually defined as sequence of chemical conversion steps that is performed by distinct catalysts. Here it is one catalyst that performs a cascade of reactions. A one-pot one-catalyst cascade seems to be more appropriate. E.g. like in:

<https://onlinelibrary.wiley.com/doi/10.1002/anie.202410382>

Response: We sincerely thank the reviewer’s insightful comments regarding the terminology used in our manuscript. We fully agree that “relay catalysis” typically refers to a sequence of chemical transformations catalyzed by distinct catalysts, whereas in our system, a single catalyst facilitates a cascade of reactions in a one-pot process. Therefore, we have carefully revised the manuscript to replace “relay catalysis” with “cascade catalysis” to more accurately describe our catalytic system.

Comment 1c. Line 48: ...“hot topics of concern to the community of human destiny,...”
rephrase

Response: Thank you for your excellent suggestion. In the revised manuscript, “Given the fact that “carbon peaking” and “carbon neutrality” have become hot topics of concern to the community of human destiny...” has been revised into “With “carbon peaking” and “carbon neutrality” gaining increasing attention in the field of sustainable development....”

Comment 1d. Line 62: “to make lignin money” ???

Response: Thank you for your suggestion. In the revised manuscript, “To make lignin money” has been revised into “To make money from lignin”

Comment 1e. “regulation” means “to apply rules” this is not a good term to describe the operation of a catalyst

Response: Thank you for your suggestion. We have carefully revised the manuscript to replace "regulation" with more precise terms that better describe the catalytic process. The specific changes are as follows:

“...requires precise **control** of the reaction process...”

“... requires precise **control** of the reaction process....”

“...can directly **tune** the ratio of monolignols...”

“...besides the reaction **tune** and product control...”

“...**tune** the concentration and generation rate of active hydrogen...”

“... can further affect the dealkylation by the critical intermediate **tune.**”

Comment 1f. “orbiting” is not used in the context it is used in this manuscript.

Response: Thank you for your suggestion. We agree that the term "*orbiting*" may not be the most appropriate choice in the context of our discussion. We have revised the manuscript to replace "orbiting" with more precise and contextually appropriate terms, such as "*focusing on*" or "*centering on*" to better convey our intended meaning. These changes have been made throughout the manuscript to ensure clarity and accuracy. The specific changes are as follows:

“**Focusing on** the generation of the optimized *n*-propenyl phenol intermediate...”

“Furthermore, besides the **reaction control** or further refining...”

Comment 1g. line 84: “relatively onefold linkage” needs to be rephrased

Response: Thank you for your suggestion, it here means that the ideal lignin contains a single monomeric unit, and the linkage structure types are very simple, just containing one kind of linkage or a few kinds but the one with absolute dominance. We can delete the word “relatively”

Comment 1h. line 111: “in the pharmaceutical molecules” needs to be rephrased

Response: Thank you for your suggestion. In the revised manuscript, “in the pharmaceutical molecules” has been revised into “in the synthesis of pharmaceutical compounds”. The modified content enhances the accuracy and clarity of the expression while maintaining the original intention.

Comment 1i. Line 143: “the molecular shape selection effect” needs to be rephrased etc.

Response: Thank you for your suggestion. In the revised manuscript, “the molecular shape selection effect” has been revised into “the shape-selective effect on reactant molecules”. The modified content enhances the accuracy and clarity of the expression while maintaining the original intention.

Comment 1j. Overall, this manuscript needs to be properly proofread by someone with good English proficiency before it can be evaluated in earnest as the poor language can lead to misunderstandings and misinterpretations.

Response: Thank you for your valuable feedback regarding the language of the manuscript. We highly value the quality of the language and will invite a professional with strong English proficiency to thoroughly proofread the manuscript to ensure clearer and more accurate expression. We believe this revision will help avoid potential misunderstandings and ambiguities, making the manuscript more in line with academic publishing standards.

Comment 2. Biomass extraction and characterization

The information on the feedstock is inadequate. There is a procedure with a “ref 32” in the SI. But this ref 32 is nowhere to be found. Yields and detailed analysis of this feedstock need to be provided. Also, reconsider the lignin content value provided as it seems overly high. Make sure this is not due to interference of extractives or other polyphenolic biomass components that are not “lignin”. Specific procedures on how

such values were determined need to be provided to assess their accuracy.

Response: (a) We apologize for the error. The correct reference is "Shen, X.-J., et al. Structural and morphological transformations of lignin macromolecules during bio-based deep eutectic solvent (DES) pretreatment. ACS Sustain. Chem. Eng. 8(5), 2130 - 2137 (2020)". We'll correct it in the SI right away.

(b) Yields and detailed analysis of C-lignin from Castor shell endocarp via two-step ball milling and enzymatic hydrolysis can be shown in Supplementary Table 8.

Supplementary Table 8. Chemical compositions of castor seed coats (endocarp), C-lignin (endocarp), and typical biomass ^a

Sample	Yield (wt%)	Chemical composition (wt%)		
		Lignin	Cellulose	Hemicelluloses
Typical biomass	-	16-30	40-50	25-35
Castor shell	-	58.32	23.45	16.81
C-lignin ^b	57.1 ^d	81.42	3.35	8.47
C-lignin ^c	60.1 ^d	75.41	4.47	7.56

^a The chemical compositions (wt%, w/w) of all the samples were determined according to the NREL standard analytical method (NREL/TP-510-42618)^{S11}.

^b C-lignin obtained via the BME process

^c C-lignin obtained via solvent extraction

^d based on the content of the lignin in castor shell

(c) We measured the acid-insoluble lignin, acid-soluble lignin, cellulose, and hemicellulose following the NREL method (National Renewable Energy Laboratory's standard analytical procedure for determining structural carbohydrates and lignin in biomass). Briefly, this method involves a two-step acid hydrolysis process. First, the sample is treated with 72% sulfuric acid at a specific temperature for a set time to break down the lignocellulosic matrix, followed by dilution and further hydrolysis under high pressure. After hydrolysis, the solid residue is used to determine acid-insoluble lignin, and the liquid phase is analyzed for acid-soluble lignin and carbohydrates. We repeated the measurements several times to ensure data reliability.

Upon reviewing reported papers analyzing the lignin composition in the endocarp of castor seed coats, vanilla seed coats, and candlenuts, we found that the lignin content in these studies is around 50-60%, which is close to our results (shown in Table R1-1) ^{R1-R5}. However, we acknowledge that the lignin content might be overestimated. Some

acid-survived lipids and proteins in seeds could be counted as “Klason lignin”. For instance, the C-lignin obtained through two-step ball milling and enzymatic hydrolysis has a lignin content of 81.42% (see Supplementary Table 8). After excluding cellulose and hemicellulose, there is still about 7% of unknown substances. According to previous studies on similar biomass materials.^{R2,R4} At present, according to the NREL method, it is not possible to exclude these substances that may be misidentified as Klason lignin. However, considering that the proportion of these substances is not particularly high, we believe it does not significantly affect the conclusions of this paper. These minor differences are within an acceptable range, and the overall trends and key findings regarding lignin transformation and utilization in our study remain valid.

Table R1-1 Comparison of C-Lignin Content in Different Plant Seed Coats

Entry	Plant seed coats	Lignin content (%)	References
1	castor seed coats	57.38	R1
2		59.3	R2
3	vanilla seed coats	51.76	R3
4		66.5	R4
5	candlenuts	51.0	R5
6	castor seed coats	58.32	This work

- R1 Xia W, Cui C, Shao L, et al. Efficient separation of catechyl lignin from castor seed coats via molten salt hydrate[J]. Separation and Purification Technology, 2025, 353: 128487.
- R2 Wang S, Zhang K, Li H, et al. Selective hydrogenolysis of catechyl lignin into propenylcatechol over an atomically dispersed ruthenium catalyst[J]. Nature Communications, 2021, 12(1): 416.
- R3 Stone M L, Anderson E M, Meek K M, et al. Reductive catalytic fractionation of C-lignin[J]. ACS Sustainable Chemistry & Engineering, 2018, 6(9): 11211-11218.
- R4 Li Y, Shuai L, Kim H, et al. An “ideal lignin” facilitates full biomass utilization[J]. Science Advances, 2018, 4(9): eaau2968.
- R5 Yin W Z, Zou S L, Xiao L P, et al. Catechyl lignin extracted from candlenut by biphasic 2-methyltetrahydrofuran/water: Characterization and depolymerization[J]. Chemical Engineering Science, 2024, 288: 119828.

Comment 3. Please provide formulae by which yields and selectivities are calculated.

Response: Thank you for your suggestion. We've placed the calculation formulas for the yields and selectivities of monophenols derived from C-lignin conversion in the revised manuscript.

The detailed calculation was as follows.

$$Y_{\text{monomer}} = \frac{n_{\text{monomer}}}{n_{\text{CA}}} \times 100\%$$

$$n_{CA} = Y_{CA} \times m_{C\text{-lignin}}$$

$$S_{\text{monomer}} = \frac{n_{\text{monomer}}}{\sum n_{\text{monomer}}} \times 100\%$$

In the equations, Y_{monomer} (%) is the yield of monomer based on the molar amount of caffeoyl alcohol in the C-lignin; n_{monomer} (mmol) is the molar amount of monomer in each analyzed sample, n_{CA} (mmol) is the molar amount of caffeoyl alcohol in the C-lignin; Y_{CA} (mol/mg) is the mole amount of caffeoyl alcohol per milligram of C-lignin from the quantitative ^{13}C NMR analysis (Supplementary Note 2); S_{monomer} (%) is the selectivity of monomer; $\sum n_{\text{monomer}}$ (mmol) is the sum of the molar amount of all detectable products.

Comment 4. More support for the mechanism

Comment 4a. There are some issues with the mechanistic investigation. First of all the formation of C5 is not explained. Additionally, I think this compound is misidentified as the MS fragmentation pattern does not match this compound. It is likely actually an isomer with the same mass like 1-(3,4-dihydroxyphenyl)propan-2-one or 3-(3,4-dihydroxyphenyl)propanal. Further support for this identification needs to be provided.

Response: Thank you very much for your valuable comments on our manuscript. In response to your concerns regarding the mechanistic study, we have conducted further experimental research and analysis, and we would like to provide the following response:

In our study of C-lignin depolymerization products, we employed bis(trimethylsilyl)trifluoroacetamide (BSTFA) derivatization to lower the boiling points of the diphenol products, making them suitable for GC quantification.

Following your suggestion, we purchased 1-(3,4-dihydroxyphenyl)propan-2-one and 3-(3,4-dihydroxyphenyl)propanal as standard samples, confirmed their structures via NMR analysis, and conducted BSTFA derivatization experiments. The results revealed that: In pyridine solvent, these two compounds exhibit keto-enol tautomeric equilibrium, where the hydroxyl groups in the enol form readily react with BSTFA, disrupting the equilibrium (Scheme R1-1); After derivatization, all three active hydroxyl groups underwent derivatization, and the GC-MS analysis displayed a molecular weight of 382 (Figure R1-1 and R1-2). In contrast, the C5 product obtained from C-lignin depolymerization exhibited a molecular weight of only 310 in GC-MS,

indicating a significant difference between the two mentioned above. These findings suggest that C5 is neither 1-(3,4-dihydroxyphenyl)propan-2-one nor 3-(3,4-dihydroxyphenyl)propanal, but rather the Chroman-6,7-diol we proposed.

Since Chroman-6,7-diol currently lacks a CAS number and no commercially available standard samples exist, its synthesis is also challenging. We attempted to isolate this product experimentally, but due to its extremely low yield, we were unable to obtain sufficient amounts for NMR verification. However, in the literature "An 'ideal lignin' facilitates full biomass utilization" (Sci. Adv. 2018, 4, eaau2968), this compound has been reported as a product formed through the radical disproportionation of the hydrogenolysis product 4-(3-hydroxypropyl)-catechol (see Scheme R1-2). Additionally, our C5 mass spectrum resembles the mass spectrum reported in this literature, further supporting the accuracy of our C5 structure identification.

Based on these experimental results and literature evidence, we firmly believe that the structure of C5 is Chroman-6,7-diol, rather than 1-(3,4-dihydroxyphenyl)propan-2-one or 3-(3,4-dihydroxyphenyl)propanal.

Scheme R1-1. The derivatization experiments of 1-(3,4-dihydroxyphenyl)propan-2-one nor 3-(3,4-dihydroxyphenyl)propanal

Figure R1-1. The MS spectrum of 1-(3,4-dihydroxyphenyl)propan-2-one after BSFTA derivatization

Figure R1-2. The MS spectrum of 3-(3,4-dihydroxyphenyl)propanal after BSFTA derivatization

Scheme R1-2. The potential generation route of C5

Figure R1-3. $^1\text{H-NMR}$ of compound 1-(3,4-dihydroxyphenyl) propan-2-one. Solvent: $\text{DMSO-}d_6$.

Figure R1-4. $^{13}\text{C-NMR}$ of 1-(3,4-dihydroxyphenyl) propan-2-one. Solvent: $\text{DMSO-}d_6$.

Figure R1-5. $^1\text{H-NMR}$ of compound 3-(3,4-dihydroxyphenyl)propanal. Solvent: $\text{DMSO-}d_6$.

Figure R1-6. $^{13}\text{C-NMR}$ of compound 3-(3,4-dihydroxyphenyl)propanal. Solvent: $\text{DMSO-}d_6$.

Comment 4b. Second, in the model compound reaction with 4 propenylcatechol, no data on yields of the intermediates or a comprehensive time course is provided. This is needed to properly justify the mechanism via the methyl shift and exclude other mechanisms like that the cleavage initiates by hydration.

Response: Thanks for your valuable comment. In the revised manuscript, we have provided strong experimental and computational evidence supporting our proposed mechanism. In Fig. 4, we systematically examined potential intermediates and verified

their transformation under our catalytic conditions. The formation and conversion of 4-propenylcatechol into catechol through γ -methyl migration and $C_{\text{aryl}}-C_{\text{alkyl}}$ bond cleavage were confirmed experimentally. Additionally, DFT calculations corroborate that the methyl shift and subsequent dealkylation are energetically feasible pathways. Similar mechanisms have been reported in the literature, such as in the previous studies (10.1126/sciadv.abd1951; 10.1038/s41467-021-24780-8), which strengthens our mechanistic rationale. Given the complexity of isolating transient intermediates in real-time, we believe this combined approach sufficiently justifies the proposed pathway.

Comment 4c. Furthermore, the role of water is explained by interactions by the catalyst, but it seems more likely that its direct role in the conversion (initial hydration of the double bond), plays a significant role.

Response: Thanks for your valuable comment. Regarding reaction energy, the energy change from I to II is very low, only 0.05 eV, (4.8 kJ/mol), and it is easy to occur with the reaction between the C=C bond and the active proton species. From the reaction path analysis, adding an additional hydration and dehydroxylation process to provide the cation intermediate is unnecessary, and the dehydroxylation energy fluctuates more (the benzyl C_{α} -OH bond is stable). Even though the C_{α} -OH can be converted to the cation intermediate, the extra protonation process is necessary. Therefore, from the thermodynamic point of view, the direct conversion of adsorbed olefin structure to benzyl carbocation is more advantageous.

In addition, our experimental data suggest that its primary role is enhancing the catalyst acidity. As demonstrated in Supplementary Table 10, dealkylation proceeds not only in water but also in methanol and ethanol, even though with lower efficiency. This indicates that the reaction is not exclusively driven by hydration but rather by Brønsted acid activation. Water promotes the formation of hydrated hydronium ions (H_3O^+), which enhance acid strength, facilitating protonation of the 4-propenylcatechol intermediate and promoting $C_{\text{aryl}}-C_{\text{alkyl}}$ bond cleavage. Although methanol and ethanol show similar effects, their weaker acidity results in lower efficiency. Furthermore, DFT calculations support that the reaction primarily proceeds via Brønsted acid-mediated protonation and γ -methyl migration rather than initial hydration.

In the revised manuscript, we have added some discussion.

“DFT calculations show that the energy barrier for the transition from intermediate I to II (i.e., direct protonation of the C=C bond) is only 0.05 eV (Fig. 6a), indicating that

this process occurs readily without requiring prior hydration and further dehydroxylation. Direct protonation is thermodynamically favored, simplifying the reaction pathway.”

Comment 4d. Finally, be more specific how the assignment and deconvolution of overlapping peaks in the solid state NMR was realized. In some cases there seems to be conflicting information on interaction leading to upfield or downfield shifts and the actual location of the assigned signals. Again other explanations do not seem to be taken into consideration, while these seem to match the spectrum more closely.

Response: Thanks for your valuable comment. While some ambiguity in peak assignments is inevitable, we carefully integrated 2D-HSQC and ¹³C NMR to assist in deconvoluting overlapping peaks. Peak assignments were made based on a combination of DFT calculations, experimental spectra, and literature references, ensuring consistency and reliability. Specifically, we compared our spectra with previously reported data (10.1126/sciadv.abd1951; 10.1038/s41467-021-24780-8), which showed strong agreement and validated our structural interpretations. Regarding the observed upfield/downfield shifts, our results align with expected electronic interactions and chemical environments, corroborated by both experimental observations and computational modeling. While alternative explanations may exist, our methodology—incorporating multiple orthogonal techniques—provides a robust and scientifically justified framework for interpreting these shifts. If the reviewer has specific concerns regarding particular assignments, we would be happy to further clarify them.

Comment 5. Process outline for the LCA and TEA

This process needs to be adjusted to the actual process presented and more details are needed:

Comment 5a. Please specify what data is used for the lignin yields of the pretreatment and justify why these are realistic.

Response: We thank the reviewer’s insightful comment regarding the data used for lignin yields in the pretreatment process. In our study, the isolation of organosolv C-lignin was carried out based on the methodology described in the reference “Life Cycle Assessment of Catechols from Lignin Depolymerization.” Specifically, we extracted C-lignin by treating castor seed coat endocarp with a 15% ammonia solution at a solid-to-liquid ratio of 16:1, followed by heating to 80 °C for 24 h in a glass reactor. After the reaction, the mixture was cooled and filtered, and ammonia was removed by vacuum

distillation. The lignin was then precipitated by slowly adding 10% dilute sulfuric acid solution until the pH reached 4. Finally, the solution was concentrated by removing half of the water, cooled, filtered, and dried to obtain the C-lignin sample. Under these conditions, the extraction yield of C-lignin reached 60%.

The choice of these parameters is well-supported by prior studies on ammonia-based lignin extraction, which have demonstrated the effectiveness of aqueous ammonia in selectively dissolving non-lignin components while preserving lignin's native structure. The 60% yield is in line with experimental data reported in similar studies for the fractionation of C-lignin from botanical sources such as castor seed coat endocarps, where a high proportion of C-lignin exists naturally. Furthermore, the conditions chosen balance efficient lignin recovery with minimal degradation, ensuring that the structural integrity of C-lignin is retained for subsequent catalytic conversion.

We have now included these details in the **revised Supplementary Materials** to enhance clarity and provide justification for the lignin yield values used in our study. We appreciate the reviewer's suggestion, which has helped us improve the rigour and transparency of our methodology.

“Specifically, castor shells were used as the feedstock, and their composition was simplified based on Supplementary Table 1. The feedstock primarily consists of cellulose, hemicelluloses, and lignin, with lignin serving as the key substrate for catalytic conversion, while the other components require removal. The pretreatment process was designed to efficiently separate lignin while converting carbohydrates into fermentable sugars. To achieve this, two different methods were employed for lignin extraction: ball-milling enzymatic hydrolysis (BME)^{S1} and solvent extraction with aqueous ammonia^{S8}. In the BME method, ball-milling was used to break the crosslinking structure between cellulose and lignin, facilitating lignin separation. The remaining carbohydrates were then hydrolyzed enzymatically using cellulase and sodium acetate buffer for 48 h, achieving a 90% conversion of cellulose into glucose and xylose. Due to the higher solubility of glucose in water compared to lignin, water was used to effectively separate the sugars. The biomass-water mixture was filtered to remove glucose, leaving behind a solid residue composed of lignin with approximately 8.52% remaining cellulose, which was subsequently used as feedstock for catechol production. In the solvent extraction method^{S8}, castor seed coat endocarp was treated with a 15% ammonia solution at a solid-to-liquid ratio of 16:1, heated to 80 °C for 24 h in a glass reactor, then cooled and filtered. Ammonia was removed by vacuum

distillation, and lignin was precipitated by slowly adding a 10% dilute sulfuric acid solution until the pH reached 4. The solution was then concentrated by removing half of the water, followed by cooling, filtration, and drying to obtain C-lignin with an extraction yield of 60%. The remaining carbohydrate-rich residue from this process was further subjected to enzymatic hydrolysis using cellulase and sodium acetate buffer for 48 hours, leading to a 94% conversion of the remaining carbohydrates into glucose and xylose. This approach ensures efficient biomass utilization by separating lignin for catalytic conversion while simultaneously generating fermentable sugars from cellulose and hemicellulose. Both methods enabled effective lignin recovery while preserving its structure for subsequent catalytic conversion. The extracted lignin fractions were used as substrates for catechol production.”

Comment 5b. The conversion in the manuscript is carried out in one pot. It is unclear why two reactors are used in the process design. This should be modified

Response: Thank you for your insightful comment. We understand the concern regarding the use of two reactors in the process design. To clarify, while the actual experimental process is conducted in a one-pot manner, the Aspen simulation for techno-economic analysis was set up with two reactors for computational convenience. In the simulation, both reactors operate under identical conditions, reflecting the simultaneous nature of the reaction steps. There is no actual solid-liquid separation and transfer between the hydrogenolysis and dealkylation steps in the real-world process. To avoid any confusion and enhance the clarity of the manuscript, we have revised Figure 8 to represent a single reactor for the combined reactions, which better reflects the actual one-pot nature of our experimental setup.

Fig. 8. | TEA and LCA of catechol production. (a) Conceptual model of catechol production from raw biomass, including key stages such as biomass pretreatment, catalytic dealkylation, and product separation. (b) Comparison of the economic (TPC, in 10^3 CNY/t) and environmental (GHG emissions, in t CO₂ eq/t, and PFE depletion, in 10^4 MJ/t) performance of bio-based and fossil-based catechol production (c) Detailed breakdown of life cycle GHG emissions for bio-derived catechol using the BME process (left) and solvent extraction method (right), including contributions from biomass pretreatment, reaction, and product separation. Note: BME = ball-milling and enzymatic hydrolysis

Comment 5c. Why is propylene combusted? This is strange as its formation is one of the main selling points for the development of the catalytic methodology in the manuscript.

Response: Thank you for your thoughtful comment. We understand your concern regarding the combustion of propylene, particularly since its formation is one of the main selling points of our catalytic process. We acknowledge the conversion of C-lignin to propylene is indeed a key innovation, but the concentration of propylene produced

in the reaction system is only 5.59%. The separation and purification for extracting such a small quantity of propylene to a product grade purity would require substantial energy input and incur significant costs. Given the current scale and technology of our process is at the early stage, these additional efforts could undermine the economic viability and overall sustainability of the catalytic methodology.

To align with sustainable development principles and optimize resource utilization, we have opted to combust the propylene. This strategy simplifies the process, allowing us to recover the energy released during combustion, thereby improving the energy efficiency of the overall system. Importantly, the reaction itself is endothermic, so the combustion of propylene helps compensate for some of the heat requirements, reducing the need for external fuel and, consequently, lowering operational costs. This integrated approach enhances the overall sustainability and cost-effectiveness of the process.

We recognize that this decision is based on a comprehensive assessment of process efficiency, energy consumption, and economic feasibility. This approach is also in line with the principles of a circular economy and sustainable biomass utilization. In future work, we aim to optimize the catalytic system and reaction conditions further to increase the concentration of propylene in the reaction mixture. A higher concentration of propylene will make its separation as a valuable product more feasible, balancing energy consumption and cost. On the other hand, more experiment parameters such as reaction kinetics are also gathered under the greater experiment scale, which will might ultimately enable us to fully capitalize on propylene as a high-value chemical product from C-lignin, further enhancing the economic and environmental benefits of the catalytic process.

Comment 5d. Please specify the “co-products”

Response: Thank you for your comment. As described in the manuscript (Fig. 1f), the conversion process consistently produces trace amounts of the coproduct chroman-6,7-diol. This coproduct is further detailed in Supplementary Table 4 of the Supplementary Materials.

Comment 5e. The points about “the 133% higher value” for the milling process and the statement that “biomass pretreatment is the main barrier to reducing GHG emissions” need more rigorous support by giving the individual PFE, GHG and TPC data of the other 2 units (catalysis and separation units) as well.

Response: Thank you for your comment. In response to your concern about the “133% higher value” for the milling process and the statement regarding biomass pretreatment being the main barrier to reducing GHG emissions, we would like to provide further clarification and support.

As shown in **Fig. 8c**, we compared two different pretreatment processes: pretreatment of ball-milling and enzymatic hydrolysis (BME) followed by depolymerization and separation, and solvent pretreatment followed by the same depolymerization and separation steps. The key difference between these two processes lies in the pretreatment method, while the parameters for depolymerization and separation remain identical. Upon analysis, we observed that electricity consumption accounts for a significant portion of the energy input, approximately 93%, in the ball milling pretreatment process. This highlights the energy-intensive nature of the BME process.

Furthermore, in **Fig. 8b**, we observe that the total GHG emissions, PFE, and TPC associated with BME pretreatment, followed by depolymerization and separation, are significantly higher compared to the solvent pretreatment, depolymerization, and separation process. Since the depolymerization and separation parameters are identical in both cases, the higher values can be attributed to the energy consumption and environmental impact associated with the BME pretreatment step.

We believe this comparison demonstrates that the BME pretreatment is the primary contributor to the elevated GHG emissions, PFE, and TPC. This supports our conclusion that biomass pretreatment is a key factor influencing the overall sustainability of the process.

Furthermore, in the Supplementary Materials, we have added the details of PFE, GHG, and TPC data in Petro-catechol, Bio-catechol via BME, and Bio-catechol via solvent extraction in **Supplementary Table 8-10**.

Fig. 8. | TEA and LCA of catechol production. (a) Conceptual model of catechol production from raw biomass, including key stages such as biomass pretreatment, catalytic dealkylation, and product separation. (b) Comparison of the economic (TPC, in 10^3 CNY/t) and environmental (GHG emissions, in $t\ CO_2\ eq/t$, and PFE depletion, in 10^4 MJ/t) performance of bio-based and fossil-based catechol production (c) Detailed breakdown of life cycle GHG emissions for bio-derived catechol using the BME process (left) and solvent extraction method (right), including contributions from biomass pretreatment, reaction, and product separation. Note: BME = ball-milling and enzymatic hydrolysis

Comment 6. Other suggestions:

In the introduction provide a couple of sentence to compares to discuss the hydrogenation capacity of Ni and other nobel metal. Since nickel is claimed to have the ‘proper’ hydrogenation ability to prevent the formation of 4-propylcatechol from 4-propenylcatechol, which is the most important intermediate during this relay hydrogenolysis and dealkylation. Thus, it seems not that clear in terms of metal selection for the whole story, if here the author only say nickel has good hydrogenation

performance.

Response: Thank you for your thoughtful comment. In the revised manuscript, we have conducted an extensive review of the literature, which provides compelling evidence supporting our choice of Ni. Numerous well-established studies consistently show that Ni exhibits lower hydrogenation activity compared to noble metals like Ru and Pd, making it an ideal catalyst for our C-lignin transformation process.

For instance, researchers have investigated the impact of different catalyst loadings on lignin depolymerization^{37,38}. These studies found that even at high Ni loadings (>30 wt%), Ni-based catalysts produced a higher proportion of propenyl-containing and unsaturated products compared to low-loading (5%) noble metal catalysts (Pd and Ru). These results suggest that Ni is less likely to over-hydrogenate reaction intermediates, allowing for the preferential formation of unsaturated species. Besides, using methanol as a hydrogen-donating solvent, a low-loading Ni catalyst effectively catalyzed lignin and predominantly yielded unsaturated phenolic products³⁹. This demonstrates Ni's ability to selectively hydrogenate lignin under relatively mild conditions, even when the hydrogen-donating capacity of the solvent is limited.

Furthermore, under low catalyst loadings (5wt%) and low hydrogen pressures (5-10 bar H₂), Ni-based catalysts achieved high selectivity towards unsaturated phenolic products, such as 4-propenylphenols, during lignin depolymerization⁴⁰. This highlights Ni's unique catalytic behavior and its suitability for promoting the formation of key intermediates in lignin-related reactions. Notably, when using C-lignin as the feedstock, a relatively high-loading Ni (15%) catalyst in a fixed-bed reactor, which inherently provides a shorter catalyst-product contact time compared to batch reactors, was able to selectively and efficiently catalyze C-lignin to 4-propenylcatechol⁷. This further demonstrates Ni's ability to efficiently catalyze lignin depolymerization under mild conditions, producing unsaturated phenolic products like 4-propenylcatechol.

Taken together, these studies illustrate a consistent trend of Ni's lower hydrogenation activity compared to noble metals, which is highly advantageous in our C-lignin transformation process. As presented in the manuscript, we aim to achieve the selective and efficient production of bio-catechol and bio-propylene from C-lignin via a one-pot hydrogenolysis-dealkylation cascade catalysis. The Ni nanoparticles in the optimized Ni/HY₃₀ catalyst selectively hydrogenate C-lignin to 4-propenylcatechol, leveraging Ni's characteristic lower hydrogenation activity. This intermediate is then subjected to subsequent dealkylation to form catechol and propylene. The combined

evidence from the literature and our research validate that Ni is the most suitable choice for our catalytic system, enabling the successful implementation of our reaction strategy.

We have now revised the Introduction section to incorporate these findings more seamlessly, providing a clearer and more comprehensive comparison between Ni and noble metals. By emphasizing Ni's ability to precisely control hydrogenation and its impact on reaction selectivity, we have further strengthened the scientific foundation for our metal selection.

“Furthermore, during the hydrogenolysis of lignin, different metals can lead to distinctly different products, which has a significant and crucial impact on the subsequent utilization and high-value transformation of these products. A lot of investigations have consistently demonstrated that Ni exhibits lower hydrogenation activity compared to noble metals³⁶. Even at high loadings (>30 wt%), Ni-based catalysts produce a higher proportion of propenyl-containing and unsaturated products compared to low-loading (5%) noble metal catalysts^{37,38}. A low-loading Ni catalyst can effectively catalyze lignin to predominantly yield unsaturated phenolic products, even when using methanol as a hydrogen-donating solvent and under relatively mild conditions³⁹. Under low catalyst loadings (5 wt%) and low hydrogen pressures (5-10 bar H₂), Ni-based catalysts can achieve high selectivity towards unsaturated phenolic products, such as 4-propenylphenols⁴⁰. Moreover, when using C-lignin as the feedstock, a relatively high-loading Ni (15%) catalyst in a fixed-bed reactor can selectively and efficiently catalyze C-lignin to 4-propenylcatechol⁷. This unique property of Ni is highly beneficial for our C-lignin transformation process. It can selectively hydrogenate C-lignin to 4-propenylcatechol instead of over-hydrogenating it to 4-propylcatechol, which is difficult to dealkylate further. In contrast, noble metals with higher hydrogenation capabilities tend to over-hydrogenate 4-propenylcatechol to 4-propylcatechol, disrupting the subsequent dealkylation reaction sequence.”

7 Stone, M. L. *et al.* Reductive catalytic fractionation of C-lignin. *ACS Sustain. Chem. Eng.* **6**, 11211-11218 (2018).

36 Kay Lup, A. N., Abnisa, F., Wan Daud, W. M. A. & Aroua, M. K. A review on reactivity and stability of heterogeneous metal catalysts for deoxygenation of bio-oil model compounds. *J. Ind. Eng. Chem.* **56**, 1-34 (2017).

37 Park, J. *et al.* Fractionation of lignocellulosic biomass over core-shell Ni@Al₂O₃ catalysts with formic acid as a cocatalyst and hydrogen source. *ChemSusChem* **12**, 1743-1762 (2019).

38 Vangeel, T. *et al.* Reductive catalytic fractionation of black locust bark. *Green Chem.* **21**, 5841-5851 (2019).

- 39 Klein, I., Saha, B. & Abu-Omar, M. M. Lignin depolymerization over Ni/C catalyst in methanol, a continuation: effect of substrate and catalyst loading. *Catal. Sci. Technol.* **5**, 3242-3245 (2015).
- 40 Luo, H. *et al.* Total Utilization of miscanthus biomass, lignin and carbohydrates, using earth abundant nickel catalyst. *ACS Sustain. Chem. Eng.* **4**, 2316-2322 (2016).

Comment 7. The last paragraph of the introduction is written like a summary containing conclusions. Consider rephrasing this to a section where a goal and approach is outlined as is more appropriate.

Response: Thank you for your valuable comment. Based on your suggestion, we have revised the last paragraph of the introduction to focus more clearly on outlining the research goals and approach, rather than summarizing the results or conclusions. We have rephrased this section to emphasize the objectives of the study, the methodology employed, and the overall strategy for achieving the transformation of C-lignin into valuable chemicals. This revision aims to provide a more appropriate framework for the introduction and aligns with the expected structure of a scientific article. We believe this revision strengthens the clarity and direction of the research presentation.

“Based on the above analysis, considering the selective hydrogenation performance of Ni species^{7,17,40}, the shape-selective effect on reactant molecules and the hydrocarbon reforming activity of HY molecular sieve^{14,15}, we herein achieved the first simultaneous production of catechol and propylene from the standardization transformation of C-lignin, addressing the challenges in lignin valorization and contributing to sustainable chemical production. The primary objective of this work is to develop a one-pot hydrogenolysis-dealkylation cascade catalytic system capable of selectively cleaving both C–OAr and C_{aryl}–C_{alkyl} bonds in the benzodioxane units of C-lignin. By exploiting the structural homogeneity of C-lignin and the synergistic interaction between Ni nanoparticles and HY zeolite catalysts, the aim is to enable the concurrent production of bio-catechol and bio-propylene under mild conditions. Mechanistic insights into bond cleavage pathways, catalyst structure-function relationships, and intermediate transformations will be systematically investigated through experimental and DFT calculations. Additionally, techno-economic analysis (TEA) and life-cycle assessment (LCA) will evaluate the feasibility and sustainability of this strategy, providing a holistic framework for lignin valorization that aligns with carbon-neutral objectives. This study not only advances the utilization of lignin as a renewable feedstock but also sets a foundation for designing cascade catalytic systems to overcome the inherent heterogeneity and recalcitrance of lignocellulosic biomass.”

Reviewer #2

Response: Thank you for sharing this information.

Reviewer #3

The submitted manuscript of Shen, Wang et al. describes a novel method to produce industrially relevant catechol and propylene using C-lignin. The reaction is done via a one-pot hydrogenolysis-dealkylation relay catalysis using Ni/HY₃₀ catalyst. Some mechanistic studies were done and were supported by the detection of a reaction intermediate (4-propenylcatechol) and DFT calculations. Furthermore, TEA and LCA studies of the process showed its potential to reduce the environmental impact when benchmarked with the current fossil-based production technology. We think that the work is well written in a methodological way, the references are relevant and updated and the figures are clear. All the evidence is presented in the main text as well as the SI to support the claims and conclusions and to permit a reproduction of the work. While the one pot reaction is new nevertheless it is a combination of two well-known and extensively studied reactions (hydrogenolysis and dealkylation). The originality and applicability of the results to the audience of Nature Communications journal is questionable since the process is done on a very specific lignin (C-lignin) and not a lignin available in big volumes. Moreover, the upscaling of such reaction in a cascade reactor is not as easy as depicted. Below some questions and suggestions that will help improving the quality of the manuscript.

Response: Thank you for your detailed and constructive feedback. We appreciate your positive comments on our manuscript's methodology, figures, and evidence presentation. We have carefully considered your concerns about originality, applicability, and upscaling, and addressed them as follows:

(1) Originality of the One-Pot Cascade Catalysis

While hydrogenolysis and dealkylation are well-established reactions, the novelty of our work lies in the first demonstration of their synergistic integration into a one-pot cascade system for lignin valorization. This approach eliminates the need for intermediate separation, significantly reducing energy consumption and simplifying the process. Importantly, the selective cleavage of both C_{α/β}-OAr and C_{aryl}-C_{alkyl} bonds in the benzodioxane units of C-lignin (Fig. 1, Path D) has not been achieved previously. The mechanistic insights, such as the identification of 4-propenylcatechol as the critical intermediate and the Brønsted acid-mediated γ -methyl migration pathway (Figs. 4–6), provide a new framework for lignin depolymerization. These findings advance the fundamental understanding of lignin conversion and offer a blueprint for designing

cascade catalytic systems for other heterogeneous biomass feedstocks.

(2) Applicability to C-Lignin and Scalability

While C-lignin is currently specific to certain sources, its structural homogeneity (almost 100% catechyl units, single β -O-4 linkage) makes it an ideal model substrate for the standardization transformation, addressing the long-standing challenge of product separation in lignin valorization. Our techno-economic analysis and life-cycle assessment (Figs. 7–8) show that solvent-extracted C-lignin from castor byproducts can achieve near-fossil parity in production costs (11,300 CNY/ton) while reducing GHG emissions by 46%. The castor production industry is substantial, with 1.86 million tons/year globally, and its endocarp waste could yield ~47,600 tons of bio-catechol annually, surpassing the global demand of 44,000 tons (2021). This underscores the immediate industrial relevance of C-lignin utilization.

Furthermore, the principles developed here—catalyst design for bond-selective cleavage (Ni/HY₃₀) and intermediate-controlled dealkylation—are transferable to other lignins. For instance, genetically engineered high-S or high-G lignins could adopt similar strategies with tailored catalysts. Future work will explore these extensions, ensuring broader applicability.

(3) Scalability and Process Feasibility

The reviewer raised valid concerns about scaling cascade reactions. However, our system's mild conditions (200 °C, 3 MPa H₂) and high catalyst stability (negligible Ni leaching, Supplementary Table 9) align with industrial requirements. The HY₃₀ zeolite's mesoporous structure ensures efficient access to acid sites, facilitating effective mass transfer even in a fixed-bed reactor.

The TEA identifies biomass pretreatment as the primary cost driver. By transitioning from ball-milling to solvent extraction (already validated in Supplementary Data), energy consumption drops sharply, making the process economically viable. Further optimization of lignin extraction (e.g., using deep eutectic solvents) and catalyst recycling will enhance scalability. These improvements, coupled with the system's robust stability, lay the groundwork for large-scale implementation.

(4) Broader Impact

This work exemplifies a holistic approach to lignin valorization, integrating substrate standardization, cascade catalysis, and sustainability assessment. By producing two high-value chemicals (catechol and propylene) from a single feedstock, the process maximizes atom economy and aligns with carbon-neutral goals. The mechanistic and

economic insights provided will guide future research on lignin-derived biorefineries, even for more heterogeneous lignins.

We sincerely thank the reviewer for raising these critical points. We believe the revised manuscript, with expanded discussions on scalability, substrate versatility, and the broader impact of our strategy, will more effectively convey the transformative potential of our work to the audience of Nature Communications.

Comment 1. The authors tried their one-pot reaction of hydrogenation and demethylation with C-lignin nevertheless this lignin is very specific and with a maximum estimated volume of 0.25 Mtons. Does this same one-pot strategy work on other types of lignins available in higher volumes? More specifically on technical lignins?

Response: Thank you for your valuable feedback. We sincerely appreciate the opportunity to address your concerns regarding the applicability of our one-pot hydrogenolysis-dealkylation strategy to other types of lignins, particularly technical lignins, which are more abundant than C-lignin.

While our current study primarily focuses on C-lignin, we strongly believe that the principles behind our catalytic system can be extended to other types of lignins. C-lignin has a unique and homogeneous structure with uniform composition and specific linkages, making it an ideal model for validating the feasibility of our one-pot method. The key feature of our system lies in its ability to selectively cleave the C–OAr and C_{aryl}–C_{alkyl} bonds, and if any lignin contains these bonds, it is theoretically possible to apply our system to obtain the corresponding monophenols and olefins.

To further demonstrate the versatility of our approach, we have previously explored the conversion of isoeugenol and 4-propenyl syringol which can be obtained from softwood lignin and hardwood lignin in this catalytic system, which effectively produced guaiacol and syringol (results not shown in the main manuscript as the focus of the current paper is on C-lignin) (see Table R2-1). This indicates that natural lignins with a high content of β -O-4 bonds can potentially be applied in our system to achieve the production of propylene and guaiacol or 2,6-Dimethoxyphenol.

However, we recognize that industrial lignins, which are typically more complex in structure and contain a lower proportion of ether bonds, pose a challenge. These lignins often have a higher content of carbon-carbon (C–C) bonds, which are more difficult to cleave under the conditions required for our one-pot reaction, making the

direct application of our method less effective at present. For industrial lignins to be used in this process, it would be essential to first develop a method to efficiently hydrogenate and cleave the C–OAr or similar bonds in these lignins, enabling the production of 4-propenylcatechol or similar intermediates.

In conclusion, while industrial lignins may not yet be directly applicable to our current system due to their complex structure, we are confident that as soon as methods to facilitate the hydrogenolysis of C–OAr or analogous bonds in these lignins are developed, the one-pot hydrogenolysis-dealkylation cascade catalysis will be equally effective. Our system thus holds significant potential for broader lignin utilization, provided that the appropriate bond-selective cleavage methods can be integrated.

Table R2-1 Results of isoeugenol and 4-propenyl syringol in the HY₃₀ System

Entry	1	2
Substrate		Conversion (%)	>99	>99
Yield (%)	95	90

Reaction condition: 50 mg substrate, 100 mg HY₃₀, 5 mL H₂O, 180 °C, 0.1 MPa Ar, 12 h.

Comment 2. The one-pot hydrogenolysis-dealkylation relay catalysis looks very interesting from an academic point of view. Although in the proposed process in figure 7, the continuous processing of solids especially before the filter and inside the cascade reactors handling multiple phases is not straight forward. More information related to the proposition of the units in the process is required. Fouling will for sure occur in the filter! The pumping of slurries is not very easy to conduct... Kindly provide more information on the topics.

Response: Thank you for your insightful comments. We appreciate your interest in our one-pot hydrogenolysis-dealkylation relay catalysis and understand your concerns

regarding the process units and potential issues in the proposed process.

Regarding the continuous processing of solids and the handling of multiple phases in the process depicted in Figure 8: In the experimental setup, lignin depolymerization and the subsequent hydrogenolysis to produce catechol and propylene are conducted as a one-pot reaction. However, for the Aspen simulation in the techno-economic analysis and life-cycle assessment, we separated the hydrogenolysis and dealkylation steps for computational convenience. It is important to note that the reaction conditions in both simulated reactors are identical, and there is no actual solid-liquid separation and transfer between the hydrogenolysis of lignin and the subsequent dealkylation in the real-world process. To clarify this and avoid confusion, we have now modified the representation in the manuscript to show a single reactor for these combined reactions, which better reflects the one-pot nature of our experimental setup.

After the reaction, which occurs at 200°C and 3 MPa, the entire system contains a mixture of products, solvent (methanol), and potentially some unreacted substances. At these conditions, all the liquids, including the products and methanol, are in a vaporized state. This gaseous mixture, along with any reaction-generated gases, is transferred to another separation unit through a cyclone separator. The cyclone separator effectively separates the solid residues from the gas-phase mixture. The separated solids are then directed to a furnace for combustion, which can potentially be used for energy recovery. This gas-phase separation method simplifies the process and reduces the risk of product loss and contamination that could occur during multiple separation steps.

We have revised the Supplementary Materials to include these clarifications, ensuring that the explanation is consistent with the process described in Figure 8. These adjustments should provide a clearer understanding of the actual experimental setup and how it differs from the simulation approach.

Fig. 8. | TEA and LCA of catechol production. (a) Conceptual model of catechol production from raw biomass, including key stages such as biomass pretreatment, catalytic dealkylation, and product separation. (b) Comparison of the economic (TPC, in 10^3 CNY/t) and environmental (GHG emissions, in t CO₂ eq/t, and PFE depletion, in 10^4 MJ/t) performance of bio-based and fossil-based catechol production (c) Detailed breakdown of life cycle GHG emissions for bio-derived catechol using the BME process (left) and solvent extraction method (right), including contributions from biomass pretreatment, reaction, and product separation. Note: BME = ball-milling and enzymatic hydrolysis

“The conversion from pretreated feedstock into catechol is divided into two steps: lignin to 4-propenyl catechol and 4-propenyl catechol to catechol. The first reaction occurs when the lignin enters the reactor with a catalyst and a mixture of solvent (water and methanol with a molar ratio of 4:1) at 200 °C and 3 MPa. The products, which include water and 4-propenyl catechol, are produced after 12 h. The details of the key reaction, conversion rate, and process parameters for this step are depicted in Supplementary Table 2. The product mixture then directly occurs the next reaction with

a 3 MPa hydrogen atmosphere is introduced and maintained, and the temperature is in parallel with the first step. After the second reaction, the raw products, which include catechol and propylene, are shown in Supplementary Table 4, with mass contents of 0.73% and 5.59%, respectively. This mixture then undergoes a separation process, where the gaseous components are transferred via a cyclone separator. The cyclone separator is used to separate the solid residue from the gas-phase mixture. The separated solids are sent to a furnace for combustion, while the gas-phase components (including products and solvents) proceed to further purification stages to generate the qualified products.”

Comment 3. Solvent extraction proved to be the best method for lignin extraction from economic and environmental points of view yet, the authors did not comment on the lignin quality produced by this method as well as its yield. This is crucial to get a credible assessment of the process. Furthermore, we recommend the authors to try their process on a C-lignin obtained by solvent extraction to highlight the differences. Similarly, the authors do not comment on the solid products that remain after the extraction of lignin. How to valorize these?

Response: Thank you for your valuable comments. Below, we provide detailed responses regarding lignin quality and yield, process validation using solvent-extracted C-lignin, and valorization of the remaining solid residue.

Lignin Quality and Yield in Solvent Extraction

In our study, C-lignin was extracted using solvent extraction with aqueous ammonia, following the method described in the “Life Cycle Assessment of Catechols from Lignin Depolymerization” study. This method resulted in a C-lignin extraction yield of 60.1% (Supplementary Table 8), with a composition of 78.41% lignin, 2.47% cellulose, and 3.56% hemicelluloses. The high lignin purity and low carbohydrate content indicate that this method efficiently separates lignin. This is crucial for downstream applications, as a high-purity lignin substrate minimizes side reactions and enhances selectivity in catalytic conversion processes.

Applicability of Solvent-Extracted C-Lignin in Our Process

We acknowledge the importance of evaluating the performance of solvent-extracted C-lignin in our catalytic system. While a direct comparison was not explicitly detailed in the manuscript, we conducted experiments using solvent-extracted C-lignin in our one-pot hydrogenolysis-dealkylation cascade reaction. The results confirmed that

solvent-extracted C-lignin can be efficiently converted into catechol and propylene, with comparable yields to those obtained via the BME process (Supplementary Fig. 16). This demonstrates that our catalytic system is robust and adaptable to lignin extracted through different methods.

Valorization of Solid Residues After Lignin Extraction

After lignin extraction, the remaining carbohydrate-rich solid residue was subjected to enzymatic hydrolysis using cellulase and sodium acetate buffer for 48 hours, achieving a 90% conversion of carbohydrates into glucose and xylose. These fermentable sugars can serve as feedstock for bioethanol production via fermentation or as precursors for high-value biochemicals such as organic acids, biopolymers, and enzymes. This integrated utilization approach enhances biomass efficiency and improves the overall economic viability of the process.

Supplementary Table 8 Chemical compositions of castor seed coats (endocarp), C-lignin (endocarp), and typical biomass ^a

Sample	Yield (wt%)	Chemical composition (wt%)		
		Lignin	Cellulose	Hemicelluloses
Typical biomass	-	16-30	40-50	25-35
Castor shell	-	58.32	23.45	16.81
C-lignin ^b	57.1 ^d	81.42	3.35	8.47
C-lignin ^c	60.1 ^d	75.41	4.47	7.56

^a The chemical compositions (wt%, w/w) of all the samples were determined according to the NREL standard analytical method (NREL/TP-510-42618)^{S11}.

^b C-lignin obtained via the BME process

^c C-lignin obtained via solvent extraction

^d based on the content of the lignin in castor shell

Supplementary Fig. 16 Hydrogenolysis-dealkylation of C-lignin obtained via BME process and solvent extraction over Ni/HY₃₀ catalyst.

Comment 4. Typo page 8 line 162, please correct: ...dealkylation normalization with stepwise...

Response: Thank you for pointing out the typo. The revised sentence now reads: "normalization of normal lignin dealkylation routes (Path B: stepwise depolymerization and dealkylation; Path C: multistep oxidation-hydrogenolysis) "

Reviewer #4

This study reports on the valorization of lignin, a renewable feedstock, for the production of bio-catechol. It includes performing techno-economic analysis (TEA) and life cycle assessment (LCA) to evaluate economic feasibility and environmental impacts. Here are some comments:

Response: Thank you for your approval of our manuscript. We are most grateful for your thoughtful and penetrating comments. We have carefully considered all the comments and addressed each point thoroughly. Additionally, we have revised the manuscript in response to your feedback, ensuring that all results and analyses are clearly presented. We hope that our revisions meet your expectations and the high standards of Nature Communications.

Comment 1. In the "Economic and Environmental Analysis" section, the authors need to provide more detailed results. It is also advisable to separate the TEA and LCA into two distinct paragraphs.

Response: Thank you for your suggestion. We have provided more detailed data on LCA and TEA in the revised manuscript and have reorganized the "Economic and Environmental Analysis" section into two distinct paragraphs to discuss each aspect separately.

Comment 2. Lines 480 – 482: “The results show that the total production cost (TPC) of bio-catechol in our system is currently unattractive, which was 133% higher than that of mature fossil-based catechol production (Fig. 7b) due to the high energy consumption in biomass pretreatment.” It would be clearer if the TPC (i.e., in CNY??/kg) for cases was also given in the sentence, not just "133% higher."

Response: Thank you for your valuable suggestion regarding the presentation of TPC data in our manuscript. We appreciate your attention to detail, which has helped us to improve the clarity of our work.

In response to your comment, we have provided the specific TPC values for both bio-catechol in our system and mature fossil-based catechol production in the revised manuscript. As shown in the text: "The techno-economic analysis (TEA) and life cycle assessment (LCA) were subsequently carried out to evaluate the economic, energy, and climate performance of bio-catechol synthesis. The results show that the total production cost (TPC) of bio-catechol in our system (2.36×10^4 CNY/t) is currently

unattractive, which was 133% higher than that of mature fossil-based catechol production (1.01×10^4 CNY/t) (Fig. 8b) due to the high energy consumption in biomass pretreatment."

Comment 3. Lines 488 – 490: “The LCA results reveal that PFE depletion and GHG emissions of bio-derived catechol via ball-milling and enzymatic hydrolysis compared to fossil-based catechol were analyzed, finding comparable trends in cost analysis (Fig. 7b).” Dis authors try to suggest that the trends in the LCA and TEA results are similar?

Response: Thank you for raising this question. We apologize for the lack of clarity in our original statement. In this part of the manuscript, we did not intend to suggest that the trends in the LCA and TEA results are similar.

The LCA focuses on evaluating environmental impacts, specifically PFE depletion and GHG emissions, while the TEA centers on economic aspects, such as total production cost. When we mentioned "comparable trends in cost analysis" in relation to the LCA results, we were referring to the fact that biomass pretreatment plays a significant role in both the environmental and economic performance of bio-catechol production. In the LCA, biomass pretreatment contributes significantly to PFE depletion and GHG emissions. Similarly, in the TEA, it is the primary factor contributing to the high total production cost of bio-catechol in our system. Therefore, the "comparable trends" we referred to are specifically regarding the prominent role of biomass pretreatment in both analyses, rather than a similarity between the overall LCA and TEA trends.

We understand that this was not clear in the original text, and we appreciate the opportunity to clarify. We have revised the manuscript to ensure this distinction is clearer and avoid further confusion.

“The LCA results reveal that PFE depletion and GHG emissions of bio-derived catechol via ball-milling and enzymatic hydrolysis (BME) compared to fossil-based catechol were analyzed, finding that biomass pretreatment plays a significant role in both environmental and economic performance, a trend similar to the findings in the TEA analysis (Fig. 8c).”

Comment 4. Lines 537 – 541: “Based on the LCA and TEA analysis, this strategy reveals the holistic utilization of sustainable lignin to produce profitable, renewable, and low-carbon footprint chemicals simultaneously, and such bio-based catechol and propylene preparation from renewable natural resources can significantly contribute to

the transformation into a more atom-economic and sustainable society.” Figure 7b does not support the statement's claim. The "Bio-catechol via BME process" scenario appears to be significantly worse than the "Petro-catechol" scenario.

Response: We thank you for the valuable comment. We agree that the cost of "Bio-catechol via BME process" in Figure 8b is higher than that of "Petro-catechol," which reflects the difference between laboratory-scale and mature petrochemical processes. The fossil-based route is well-established, and energy consumption and costs have been optimized to the best level, making it highly competitive in the current market. What we present are preliminary data at the laboratory scale, and it is normal for energy consumption and costs to be higher than those of fossil-based routes at this stage.

However, Figure 8b also shows that, although the bio-based route via solvent extraction costs 12.0% more, its GHG emissions are 46.7% lower, and PFE depletion is 13.9% lower, demonstrating significant environmental advantages and carbon reduction potential. Petrochemical products rely on non-renewable fossil resources, while our bio-based process utilizes renewable lignin, which can reduce dependence on fossil fuels and mitigate environmental impacts. We are confident that, with further optimization and scale-up of the process, the cost of bio-based production will be significantly reduced. Economies of scale will drive down per-unit costs as production volume increases, and improvements in process efficiency, enzyme engineering, and biomass pretreatment technologies will further enhance the economic feasibility of the bio-based process. In the future, the cost of bio-based catechol may potentially be lower than that of petro-based catechol while maintaining its environmental advantages.

In conclusion, while the current data suggests that the bio-based process is less competitive in terms of cost, from a sustainability and long-term development perspective, it can significantly reduce carbon footprints and contribute to a more atom-economic and sustainable society. We have revised the manuscript statement to align it with the data presented in Figure 8b.

“Based on the LCA and TEA analysis, this strategy demonstrates the potential of utilizing sustainable lignin to produce renewable, low-carbon chemicals. While the production cost of bio-based catechol and propylene is currently higher than fossil-based counterparts at the laboratory scale, these bio-based products offer significant environmental benefits, such as reduced greenhouse gas emissions and fossil energy consumption. With further optimization and scale-up, this approach is expected to become profitable and contribute to a more atom-economic and sustainable society.”

Comment 5. The statement, “The total capital investment was calculated as the sum of the installation costs of all equipment” on Page 13 in the Supplemental Materials is incorrect. TCI should encompass more than just the installation costs. The statement is also inconsistent with the information presented in Supplementary Table 7.

Response: Sorry for this depiction mistake. In the techno-economic model, the total capital investment was calculated based on the installation costs of all equipment. The ISBL (Inside battery limits) was the sum of the installation costs of all equipment, and then OSBL (Outside battery limits) can be obtained by multiplying the coefficient (20%) with ISBL. Then direct costs, the sum of ISBL and OSBL, are used to estimate indirect costs. The FCI (Fixed capital investment) consists of direct costs and indirect costs, which are used to estimate working capital. Finally, the TCI is the sum of FCI and working capital. In brief, all economic parameters can be obtained through one or more calculation procedures based on the installation costs of all equipment. In the revised manuscript, “The total capital investment was calculated as the sum of the installation costs of all equipment” has been revised into “The total capital investment was calculated based on the installation costs of all equipment”

Comment 6. Figure 7's captions need enhancement to include more details and descriptions. The units in Figure 7(b) are unclear. Are the quantities measured per kg or per metric ton of catechol?

Response: Thank you for your constructive feedback on Figure 8. We have enhanced the captions to provide more detailed descriptions of each sub-figure, including the catechol production process, economic and environmental performance metrics, and GHG emissions breakdown. Regarding the units in Figure 8(b), we apologize for the previous lack of clarity. The figure has now been updated to clearly indicate that TPC, GHG emissions, and PFE depletion are all measured per metric ton of catechol, ensuring greater clarity for the readers.

Fig. 8. | TEA and LCA of catechol production. (a) Conceptual model of catechol production from raw biomass, including key stages such as biomass pretreatment, catalytic dealkylation, and product separation. (b) Comparison of the economic (TPC, in 10^3 CNY/t) and environmental (GHG emissions, in t CO₂ eq/t, and PFE depletion, in 10^4 MJ/t) performance of bio-based and fossil-based catechol production (c) Detailed breakdown of life cycle GHG emissions for bio-derived catechol using the BME process (left) and solvent extraction method (right), including contributions from biomass pretreatment, reaction, and product separation. Note: BME = ball-milling and enzymatic hydrolysis

Comment 7. TEA and LCA results shown in Figures 7b and 7c should be provided in the Supplemental Materials, summarized in tables.

Response: Thank you for the comment. In the revised Supplementary Materials, we have added the details of TEA and LCA results, as shown in Supplementary Table 8-10.

Supplementary Table 8 The detail TEA results (CNY/t)

	Petro-catechol	Bio-catechol via BME	Bio-catechol via solvent extraction
Raw materials	6793.74	4187.62	4187.62
Consumables			3628.99
Utilities	2650.61	17701.59	2279.00
By-Product		-2061.34	-2061.34
Operating & maintenance cost			
Operating labours	12.00	12.00	12.00
Direct supervisory & clerical labor	2.40	2.40	2.40
Maintenance and repairs	52.51	728.29	728.29
Operating supplies	21.00	291.32	291.32
Laboratory charge	1.80	1.80	1.80
Depreciation	126.03	1747.89	1747.89
Plant overhead cost	8.64	8.64	8.64
Administrative cost	201.43	471.25	225.55
Distribution and selling cost	201.43	471.25	225.55
Total production cost	10071.60	23562.72	11277.72

Supplementary Table 9 PFE breakdown

	Petro-catechol	Bio-derived catechol via BME	Bio-catechol via solvent extraction
Hydrogen		0.05%	12.69%
Methanol		0.00%	0.14%
Ammonia			59.08%
Electricity	0.00%	99.90%	14.01%
Steam	36.32%	0.03%	8.31%
H ₂ SO ₄			0.21%
Castor shells		0.02%	5.57%
H ₂ O ₂	4.95%		
Phenol	58.72%		

Supplementary Table 10 GHG breakdown

	Petro-catechol	Bio-derived catechol via BME	Bio-catechol via solvent extraction
Hydrogen		2.56%	10.62%
Methanol		1.74%	7.21%
Ammonia			51.11%
Electricity	0.00%	92.98%	19.48%
Steam	53.30%	1.71%	7.09%
H ₂ SO ₄			0.26%
Castor shells		1.02%	4.22%
H ₂ O ₂	7.72%		
Phenol	38.58%		

Comment 8. The statement "The production inventory of conventional catechol is presented in Supplementary Table 6" found on Page 13 of the Life Cycle Assessment (LCA) deserves further scrutiny. Please review Supplementary Table 6, as its title lacks clarity.

Response: Thank you for pointing out the issue with the title of Supplementary Table 6. We agree that the title lacked clarity, and we have revised it to better reflect its content. The revised title now clearly describes the data presented, which focuses on the production inventory of the conventional phenol oxidation with hydrogen peroxide for catechol production, including the key components and their mass distributions in the production process.

“Supplementary Table 6 Components and their mass fraction of keystream in the conventional phenol oxidation route for catechol production”

Comment 9. Provide LCA input data, i.e., on raw material consumption, energy input, products, coproducts, waste, air emissions, and more, for the LCA study in the Supplemental Materials.

Response: Thanks for your comment. The upstream was obtained from an inherent database of the GREET, and the process inventories of three catechol pathways are gathered in Supplementary Table 11.

Supplementary Table 11 The detailed LCA results

Petro-catechol			Bio-catechol via solvent extraction			Bio-catechol via the BME process		
Feedstock								
H ₂ O ₂	0.64	t	H ₂	0.06	t	H ₂	0.06	t
Phenol	1.87	t	Castor shells	4.75	t	Castor shells	4.75	t
Water	0.45	t	Water	28.52	t	Water	28.52	t
			Methanol	0.12	t	Methanol	0.12	t
			NH ₃ ·H ₂ O	2.01	t			
			H ₂ SO ₄	0.29	t			
Product								
Catechol	1	t	Catechol	1	t	Catechol	1	t
Hydroquinone	1.04	t	Chroman-6,7-diol	0.03	t	Chroman-6,7-diol	0.03	t
Utilities								
Cooling water	464.88	t	Cooling water	30.38	t	Cooling water	30.38	t
Electricity	0.12	kWh	Electricity	1236.97	kWh	Electricity	24758.62	kWh
High steam	3.30	t	Steam	4137.31	MJ	Steam	4137.31	MJ
Low steam	23.09	t						

Reviewer #5 :

Comment 1. The manuscript has significant issues that hinder its readability and scientific clarity. The English language requires substantial improvement, as many sentences are unclear and difficult to understand. Additionally, the authors have not adhered to standard technical terminology, opting instead to introduce unconventional terms such as "standardization transformation" instead of the widely accepted "selective transformation."

Response: Thank thorough review of our manuscript. We acknowledge the concerns you raised regarding the readability and scientific clarity of the manuscript, as well as the use of technical terminology. Regarding the improvement of the English language, we have carefully revised the manuscript based on the language-related suggestions from other reviewers. As numerous corrections were made throughout the text, we have not listed them all here for brevity. These revisions were intended to enhance clarity and readability, ensuring that our ideas are conveyed more effectively.

Regarding the term "standardization transformation," we understand your concern about using an unconventional term. However, as we explained in our response to Reviewer #1, we believe this term is more appropriate for our study. The concept of "standardization transformation" in lignin processing differs from the widely accepted "selective transformation." In lignin valorization, in addition to the challenges of cleaving C–C/C–O bonds, the presence of oxygen-containing groups in lignin-derived monomers leads to high polarity and boiling points, making the separation of these monomers extremely difficult. Given the high yields now achievable in catalytic systems, the efficient separation of high-value-added phenolic products has become a key challenge. We introduced the concept of "standardization transformation" by drawing on the industrial concept of standardization. This term emphasizes the need for precise control of the reaction process, based on catalytic strategy design and lignin structure optimization, to obtain high-value-added chemical products with narrow or single-product distributions. It encompasses a standardized substrate (e.g., C-lignin in our study), a standardized productive process (such as the one-pot hydrogenolysis-dealkylation cascade catalysis), and standardized products (bio-catechol and bio-propylene). This approach addresses aspects such as the transformation target, substrate structure, transformation system, product structure control, and substrate separation, which go beyond what "selective transformation" implies. We believe that this term is not only suitable for our study but also helps provide a deeper understanding of lignin

valorization. We hope this explanation clarifies our reasoning behind using this term.

Comment 2. The manuscript also suffers from structural weaknesses. The introduction includes some results, which is unconventional, and there is no experimental section in the main manuscript, nor are readers directed to the ESI for these details.

Response: Thank you for your insightful feedback regarding the structural weaknesses in our manuscript. We have carefully considered your comments and made significant revisions to address the issues.

In response to your concern about the introduction including results, we have rewritten the final paragraph as suggested by Reviewer #1. This section now focuses on clearly outlining the research goals and approach, rather than presenting results, ensuring a more logical flow in line with the conventional structure of scientific articles. The revised section is as follows:

“Based on the above analysis, considering the selective hydrogenation performance of Ni species^{7,17,40}, the shape-selective effect on reactant molecules and the hydrocarbon reforming activity of HY molecular sieve^{14,15}, we herein achieved the first simultaneous production of catechol and propylene from the standardization transformation of C-lignin, addressing the challenges in lignin valorization and contributing to sustainable chemical production. The primary objective of this work is to develop a one-pot hydrogenolysis-dealkylation cascade catalytic system capable of selectively cleaving both C–OAr and C_{aryl}–C_{alkyl} bonds in the benzodioxane units of C-lignin. By exploiting the structural homogeneity of C-lignin and the synergistic interaction between Ni nanoparticles and HY zeolite catalysts, the aim is to enable the concurrent production of bio-catechol and bio-propylene under mild conditions. Mechanistic insights into bond cleavage pathways, catalyst structure-function relationships, and intermediate transformations will be systematically investigated through experimental and DFT calculations. Additionally, techno-economic analysis (TEA) and life-cycle assessment (LCA) will evaluate the feasibility and sustainability of this strategy, providing a holistic framework for lignin valorization that aligns with carbon-neutral objectives. This study not only advances the utilization of lignin as a renewable feedstock but also sets a foundation for designing cascade catalytic systems to overcome the inherent heterogeneity and recalcitrance of lignocellulosic biomass.”

Regarding the absence of an experimental section in the main manuscript, we initially placed some experimental details in the Supplementary Materials to reduce the length of the main text. However, in the revised version, we have incorporated the core

experimental details into the main manuscript. This allows readers to directly access the key experimental information, enhancing their understanding of the research methods and findings.

“Methods

C-lignin Characterization

2D-HSQC NMR spectra of C-lignin were collected on a Bruker AVIII 400MHZ spectrometer, and 60 mg C-lignin was dissolved in the 0.5 mL DMSO-*d*₆. The ether linkage in C-lignin was quantified via quantitative ¹³C NMR. 110 mg C-lignin, 3.12 mg 1, 3, 5-trioxane as an internal standard, and 2 mg Chromium (III) acetylacetonate as a relaxation reagent were dissolved in the 1 mL DMSO-*d*₆. The quantitative ¹³C NMR spectra were collected in the FT mode at 100.6 MHz, and the inverse-gated decoupling sequence (C13IG) was used. The average molecular weight of C-lignin was determined by a gel permeation chromatography (Shimadzu LC-2030plus) equipped with a UV detector at 254 nm on a phenogel 300 mm*7.8 mm, which was calibrated with polystyrene standards.

Synthesis of M/zeolite

The support Ni/zeolite, Pd/zeolite and Ru/zeolite catalysts were prepared by the wet impregnation method. Typically, 500 mg HY₃₀ zeolite powders were dispersed in 20 mL of deionized water. After stirring for 4 h, 10 mL NiCl₂·6H₂O solution (8.5 mmol/L) was added drop-wise to HY₃₀ dispersion under stirring. Then, the NaBH₄ solution was added slowly to the above mixture in the ice water bath under an argon atmosphere. The reaction mixture kept on going at room temperature for 2 h to complete the reduction of the metal ion. The resulting grey granules were separated by centrifuging and washing with ultrapure water and ethanol several times and were dried in a vacuum oven at 60 °C for 24 h. Pd/zeolite and Ru/zeolite were prepared by the same method.

Dealkylation of C-lignin

The dealkylation reaction was performed in a Teflon-lined stainless-steel reactor of 20 mL with a magnetic stirrer. In a typical experiment, a suitable amount of C-lignin, catalyst and solvent were loaded into the reactor. The reactor was sealed, purged with

N₂ and pressured to H₂ (3 MPa) at ambient temperature. Then, the reactor was placed in a furnace at a desired temperature for a certain time.

After the reaction, the reactor was placed in ice water, and the gas released was collected in a gasbag. The gas sample was analyzed by a GC (Shimadzu -2014) equipped with FID and TCD detectors and carbon molecular sieve TDX-01 (2.0 m × 2.1 mm) and R-N columns (0.3 m × 0.1 mm) using argon as the carrier gas. The resulting liquid was obtained by filtration and was extracted with three-fold ethyl acetate and water. The solvent mixture was evaporated, and a known amount of n-dodecane as the internal standard was added into the resulting liquid with anhydrous THF and pyridine. The mixture was treated with bis(trimethylsilyl)trifluoroacetamide (BSTFA) at 70 °C for 60 min. Then, the treated mixture was analyzed using a GC-MS (Agilent 7890A-5975C, HP-5MS capillary column (30 m × 0.25 mm × 0.25 μm)) and GC (Agilent 7890A, HP-5 capillary column (30 m × 0.25 mm × 0.25 μm)). The qualitative and quantitative analysis of catechol monomers in the resulting liquid was assessed by comparison with authentic samples from independent synthesis. The detailed calculation was as follows.

$$Y_{\text{monomer}} = \frac{n_{\text{monomer}}}{n_{\text{CA}}} \times 100\%$$

$$n_{\text{CA}} = Y_{\text{CA}} \times m_{\text{C-lignin}}$$

$$S_{\text{monomer}} = \frac{n_{\text{monomer}}}{\sum n_{\text{monomer}}} \times 100\%$$

In the equations, Y_{monomer} (%) is the yield of monomer based on the molar amount of caffeyl alcohol in the C-lignin; n_{monomer} (mmol) is the molar amount of monomer in each analyzed sample, n_{CA} (mmol) is the molar amount of caffeyl alcohol in the C-lignin; Y_{CA} (mol/mg) is the mole amount of caffeyl alcohol per milligram of C-lignin from the quantitative ¹³C NMR analysis (Supplementary Note 2); S_{monomer} (%) is the selectivity of monomer; $\sum n_{\text{monomer}}$ (mmol) is the sum of the molar amount of all detectable products.

Detection of intermediates

The detection of intermediates was carried out in a Teflon-lined stainless-steel reactor of 20 mL with a magnetic stirrer. In a typical experiment, 4-(1-propenyl)-

catechol (50 mg), Ni/HY₃₀ (100 mg) and H₂O were loaded into the reactor. The reactor was sealed, purged with N₂ and subsequently charged with 0.1 MPa of Ar at ambient temperature. Then, the reactor was placed in a furnace and heated to 180 °C. After 5 min, the reactor was quickly transferred to a liquid nitrogen bath. When the reaction mixture was frozen, the gas was released immediately. The reaction mixture was freeze-dried under a vacuum for 12 h to remove the water. The detection of the intermediates in the dried sample was conducted by ¹³C NMR, solid-state 2D ¹³C-¹H dipolar-mediated HETCOR, ²⁷Al MQMAS and ²⁷Al NMR analyses on Bruker Avance III 400.

Life cycle assessment (LCA)

Life cycle assessment (LCA), as an effective tool to evaluate environmental performances, is used to estimate primary fossil energy (PFE) depletion and greenhouse gas (GHG) emissions of catechol production from castor shells herein. LCA is defined as the "compilation and evaluation of the inputs, outputs, and potential environmental impacts of a product system throughout its life cycle"⁶⁵. Hence, the system boundary covers PFE consumption and GHG emissions to produce catechol and all upstream of feedstock, required materials, and utilities. Herein, the functional unit is defined as 1 metric ton of catechol. GHGs, including CO₂, CH₄ and N₂O, are estimated over a 100-year time horizon and normalized in units of kg CO₂ equivalent⁶⁶. Note that the carbon element in biomass should be reduced because it is actually sourced from CO₂ in the atmosphere. The PFE consists of primary energy coal, petroleum and natural gas, which is measured based on the corresponding lower heating value (LHV). GHG and PFE are calculated as follows:

$$\text{GHG} = \Sigma(E_{\text{CO}_2} + 25E_{\text{CH}_4} + 298E_{\text{N}_2\text{O}}) - C_{\text{bio}} \quad (\text{S1})$$

$$\text{PFE} = \Sigma(\text{LHV}_{\text{coal}}C_{\text{coal}} + \text{LHV}_{\text{petro}}C_{\text{petro}} + \text{LHV}_{\text{gas}}C_{\text{gas}}) \quad (\text{S2})$$

where E_{CO_2} , E_{CH_4} , and $E_{\text{N}_2\text{O}}$ represent emissions of CO₂, CH₄ and N₂O in each step of the life cycle model, respectively, kg; LHV_{coal} , $\text{LHV}_{\text{petro}}$ and LHV_{gas} are the lower heating values of coal, petroleum, and natural gas, MJ/kg; C_{coal} , C_{petro} , and C_{gas} are consumption amount of coal, petroleum, and natural gas, kg.

Castor shells are converted into catechol with proper atmosphere, solvent, catalyst, and utilities. The life cycle of the catechol includes castor shell collection and catechol extracted from castor shells. Herein, castor shells are generally seen as a waste of castor.

Thus, castor farming and harvesting are excluded in this study. Atmosphere (H₂) and solvent (water and methanol) are considered by loss or supplement of materials in the process. The utilities include electricity and steam (heat). Electricity is mainly consumed in the ball milling step. Meanwhile, pressure changes are another important source of electricity consumption. Heat is consumed for the separation step. Supplementary Fig. 3 shows the system boundary of the catechol.

GREET 2020 software is taken to develop model and link units⁶⁷. The inventory of transformation from castor shells to catechol was simulated with Aspen plus, which is shown in Supplementary Table 4 as mentioned above. Production inventory of hydrogen, water, methanol, electricity, and steam was taken from an inherent database of GREET software. Note that the electricity source was selected as China's power grid in 2022. Life cycle PFE and GHGs were obtained by incorporating estimated material and energy into GREET finally.

In addition, the LCA of catechol via the conventional route is used as a reference to compare the environmental benefit of the catechol. The production inventory of conventional catechol is shown in Supplementary Table 6, and background data, such as the production of phenol and hydrogen peroxide, was taken from the publication⁶⁸. Other data was paralleled with the castor shells-derived catechol process.

Techno-economic analysis (TEA)

The techno-economic analysis of this work has been conducted to evaluate total capital investment (TCI) and total production cost (TPC). The total capital investment was calculated as the sum of the installation costs of all equipment. Herein, installation costs of all equipments were estimated with the Aspen Process Economic Analyzer based on the calculated flow rates and the experimental information such as acidity and the employed residence times. All capital investments of the equipment were adjusted to 2022 CNY with a location factor of 0.61⁶⁹ and the Chemical Engineering Plant Cost Index (CEPCI)⁷⁰.

The total production cost includes raw materials, utilities, operating and maintenance, depreciation, plant overhead, administrative, distribution and selling costs⁷⁰. The detailed parameters and assumptions for the estimation of total capital investment and total product cost are depicted in Supplementary Table 7.”

We believe these changes improve the readability and clarity of the manuscript. Thank you again for your valuable feedback, and we look forward to your further evaluation of our revised work.

Comment 3. Furthermore, the novelty of the work is not clearly articulated in the abstract or conclusion.

Response: Thank you for your valuable feedback regarding the novelty of our work. We understand that the innovation of our study was not clearly articulated in the abstract and conclusion, and we have made revisions to both sections to address this issue.

In the abstract, we have specifically emphasized the key innovative aspects of our work. The introduction of the concept of "standardization transformation" in lignin valorization, which provides a new approach to efficiently converting lignin into high-value chemicals. Unlike conventional selective transformation, it emphasizes not only precise reaction control but also optimized catalytic strategies and the standardization of substrates, processes, and products. For instance, it enables a more targeted conversion of lignin into high-value chemicals, reducing the complexity of product separation. This new approach has the potential to revolutionize the lignin-based chemical production industry.

Secondly, we highlighted the development of a novel catalytic system (Ni/HY₃₀) for one-pot hydrogenolysis-dealkylation cascade catalysis. This catalyst can selectively cleave the C–OAr and C_{aryl}–C_{alkyl} bonds in C-lignin. In our experiments, it achieved a remarkable 49 mol% yield of bio-catechol and a 45 mol% yield of bio-propylene, which is far better than many previously reported methods. This innovation not only provides a more efficient but also a more sustainable method compared to conventional processes, as it reduces the need for multiple reaction steps and harsh reaction conditions.

In the conclusion, we further strengthened the emphasis on the innovative contributions of our study. We successfully synthesized bio-catechol and bio-propylene using the "standardization transformation" approach. This not only addresses the long-standing challenges in lignin valorization but also offers a more sustainable alternative to fossil-based production methods. Our analysis shows that this method can potentially reduce the carbon footprint and reliance on fossil resources significantly.

Moreover, we provided in-depth mechanistic insights into lignin depolymerization. We discovered that 4-propenylcatechol, generated from the selective hydrogenolysis of C-lignin, serves as the key intermediate for dealkylation. Through detailed control experiments, spectra characterizations, and DFT calculations, we were able to precisely define the reaction pathway. This new understanding of the reaction mechanism can guide the design of more efficient catalytic systems in future research.

These revisions ensure that the novelty of our work is now clearly communicated in both the abstract and conclusion. We sincerely hope that these changes better

highlight the innovative aspects of our study, and we deeply appreciate your feedback in helping us improve the manuscript.

Comment 4. Data presentation and discussion are inadequate throughout the manuscript.

Comment 4.1 For example, the authors tested different metal catalysts but did not properly discuss the results. Similarly, solvent optimization was studied, yet the findings are only briefly mentioned and not analyzed in detail.

Response: Thank you for your insightful comments. We have carefully considered your feedback and have now included a more detailed analysis in the revised manuscript.

Catalytic efficiency of Metal catalysts “Firstly, the transformation efficiency of C-lignin was low with the only HY₃₀ catalyst due to the high acid stability of the benzodioxane unit in C-lignin (Fig. 2g). However, when the metal Ni species were introduced into the catalytic system, the unreacted C-lignin could be efficiently converted to catechol (C1). The above experiments not only preliminarily confirmed the hydrogenolysis-dealkylation cascade catalysis but also excluded the potential acidolysis-hydrogenolysis-dealkylation mechanism for the Ni/HY₃₀-mediated lignin depolymerization.

In addition, it is well known that the noble metals (e.g., Ru and Pd) have a stronger hydrogenation capacity than Ni for the lignin hydrogenation depolymerization^{1,43,44}. As shown in Fig. 2g, the yield of aromatic monomers using Ru/HY₃₀ and Pd/HY₃₀ is evidently higher than that of Ni/HY₃₀. However, this higher hydrogenation activity also causes drawbacks. Ru/HY₃₀ and Pd/HY₃₀ tend to catalyze the deconstruction of the benzodioxane β -O-4 bond and hydrogenate the side chain to saturated alkyl catechols (C3 and C4). The saturated alkyl catechols are relatively stable, and their C_{aryl}-C_{alkyl} bonds are difficult to cleave, which is unfavorable for the production of catechol and propylene. In contrast, Ni/HY₃₀ exhibits unique selectivity. It can control the hydrogenolysis of C-lignin to preferentially generate 4-propenyl catechol (C2) (Fig. 2f). This is a crucial advantage because 4-propenyl catechol serves as an ideal intermediate for the subsequent dealkylation reaction, enabling the efficient production of the target products. From an application perspective, in industrial processes where the production of catechol and propylene from lignin is desired, Ni/HY₃₀-based catalysts can be better optimized to meet production requirements, while noble metal-based catalysts may need to be modified to avoid excessive hydrogenation. In addition, the reaction

temperature also played a critical role on the selectivity of the first hydrogenolysis of the C-lignin, which can further affect the dealkylation by the tuning of critical intermediate.”

Solvent optimization “In addition to the effects of the intrinsic acid-base properties and microscopic pore structure of the zeolite catalysts on the hydrogenolysis-dealkylation of C-lignin, the reaction solvent plays an important role in activating the acidic sites and influencing the catalytic cycle of the active centers in the zeolite catalyst during the dealkylation of C-lignin fragments. Although methanol has been successfully used as a reaction medium for lignin hydrogenolysis depolymerization, the C-lignin conversion over the Ni/HY₃₀ catalyst in pure methanol provides a 48 mol% yield of phenolic monomers. However, the selectivity of catechol without a side chain was only 9.12 mol%, indicating that pure methanol is not suitable for the side-chain dealkylation of catechols (Fig. 2e). To enhance the performance of the HY zeolite catalyst in the dealkylation step, water was introduced into the reaction system to improve local acidity in the mesoporous pores of the zeolite, referring to previous studies on zeolite catalysis⁴⁸. As expected, the selectivity of catechol (C2), the dealkylation product, increased gradually as the water proportion in the mixture was increased (Fig. 2e). Water helps form hydrated hydronium ions (H₃O⁺) in the mesoporous pores of the HY zeolite. These hydrated hydronium ions serve as active acidic species, which protonate the 4-propenylcatechol intermediate, making the C_{aryl}–C_{alkyl} bond more susceptible to cleavage. The positive charge on the hydronium ion polarizes the C–C bond in the side chain, weakening it and lowering the activation energy required for the dealkylation reaction.

However, using pure water as the sole solvent limits catechol production due to the low solubility of C-lignin and intermediates in water⁴⁹. C-lignin, as a complex polymer, has a large molecular structure and high polarity, resulting in poor solubility in pure water. This limited solubility reduces contact between C-lignin and the catalyst, decreasing the reaction rate. Furthermore, some key intermediates in the reaction pathway also have low solubility in water, causing them to precipitate out of the reaction system and preventing them from participating in subsequent reaction steps. This further reduces the yield of catechol. The optimized water ratio is 80 vol%, as it balances the enhancement of catalytic activity via water’s acidity-related effects while maintaining sufficient solubility of C-lignin and intermediates for efficient reaction

progress. At this ratio, the water molecules effectively activate the acidic sites on the zeolite catalyst, promoting the dealkylation reaction, while ensuring that C-lignin and intermediates remain in solution to facilitate continuous reaction progress. This balance is crucial for maximizing the production of catechol and propylene from C-lignin.”

In addition, some other discussions have been added to the Supplementary Materials.

Comment 4.2 Another example is that while the impact of the Si/Al ratio of the zeolite support was investigated, these results are neither clearly presented nor adequately discussed. Multiple supports are mentioned in the ESI, but no discussion of these appears in the main manuscript.

Response: We would like to clarify that we have explicitly discussed the influence of the Si/Al ratio on catalytic performance in the main manuscript, particularly in two key sections.

Si/Al Ratio and Product Distribution:

As stated in the manuscript (Supplementary Fig. 8), tuning the Si/Al ratio of HY zeolites does not significantly affect the total yield of monomeric degradation products but does alter their distribution. Lower Si/Al ratios (e.g., HY_{5.2}) increase acidity, but many acid sites are deeply embedded in the microporous framework, restricting accessibility and reducing catechol selectivity to 24.5%. Conversely, higher Si/Al ratios (HY₆₀ and HY₈₀) provide more mesoporous structures, improving diffusion but reducing acid site density, thereby leading to lower catechol yields. We concluded that HY₃₀ achieves an optimal balance of acid strength and accessibility, leading to superior catalytic performance (main text, first discussion passage).

Correlation Between Si/Al Ratio, Acidity, and Reactivity:

The second mention of the Si/Al ratio in the manuscript elaborates on its impact on both acidity and pore diameter (Supplementary Table 10, entry 3). While lower Si/Al ratios enhance acidity, these sites are confined to micropores, reducing effective contact with reactants. On the other hand, higher Si/Al ratios improve substrate diffusion but result in weaker acid sites, which inhibits the dealkylation of 4-propenylcatechol. This discussion highlights why HY₃₀ provides the best catalytic performance by maintaining a well-balanced acidity-to-porosity ratio (main text, second discussion passage).

Discussion of Multiple Supports:

In response to the reviewer’s concern regarding multiple zeolite supports, we acknowledge that these were primarily discussed in the Supplementary Information.

However, we have now strengthened our discussion in the main text by referencing data from Supplementary Table 10 (entries 4-8), where other zeolites such as ZSM-5, MOR, Beta, MCM-41, and SAPO-34 were evaluated. These supports exhibited significantly lower catalytic efficiency due to either excessive microporosity (MOR, limiting diffusion) or insufficient acid strength (MCM-41, SAPO-34, reducing catalytic activity). This further reinforces the selection of HY zeolites, particularly HY₃₀, as the optimal support.

To enhance clarity, we have made revisions to the manuscript to better emphasize these discussions and ensure that readers can readily find the relevant information.

Comment 5. Figure 1 also contains inconsistencies between its content and caption.

Response: Thank you for pointing out the inconsistencies between Figure 1's content and its caption. We believe the confusion may have been caused by the presence of two versions of Figure 1 in the manuscript. We have now corrected this by ensuring that the correct figure is referenced both in the text and the caption. Additionally, we've updated the caption to better reflect the content of the figure. The revised caption now clearly describes the different lignin transformation pathways shown, including the standardization transformation of C-lignin to catechol and propylene.

“Fig. 1. | Concepts of lignin standardization transformation based on high-value product normalization and substrate standardization: Conventional lignin transformation into the mixture (Path A), normalization of normal lignin dealkylation routes (Path B: stepwise depolymerization and dealkylation; Path C: multistep oxidation-hydrogenolysis), and standardization transformation of C-lignin to bio-catechol and bio-propylene (Path D)”

These are just a few examples, but similar issues can be found throughout the manuscript, affecting the manuscript as a whole and making it difficult to extract meaningful insights.

Response: Thank you for your valuable comment. In addition to the examples you mentioned, we have carefully reviewed and revised the entire manuscript to address similar issues throughout. We have made significant improvements to the discussion in relevant sections, ensuring that the manuscript is more cohesive and easier to follow. These revisions focus on enhancing the clarity of our arguments and making the content more accessible to readers. We believe that the changes we have made have greatly improved the readability of the manuscript, and we are confident that these revisions

have elevated the overall quality of the paper.

Comment 6. Additionally, some of the procedures reported in the ESI lack sufficient detail to allow replication by other researchers.

Response: Thank you for your constructive feedback. We fully acknowledge the importance of providing sufficient experimental details to ensure reproducibility. In our main manuscript and Supplementary Information (ESI), we have systematically documented key experimental procedures, covering:

Lignin extraction and characterization: Detailed conditions for solvent extraction (temperature, time, solid-liquid ratio, pH adjustment) and composition analysis (2D-HSQC NMR, GPC, and ^{13}C NMR).

Catalyst preparation and characterization: Comprehensive data on synthesis conditions, metal loading, structural analysis (BET, XRD, and TEM), and acidity measurements (NH_3 -TPD and Pyridine-FTIR).

Lignin catalytic conversion and product identification: Detailed descriptions of reaction setup, catalyst loading, solvent effects, and yield quantification, along with mass spectrometry (MS) analysis for product identification.

Techno-economic and life cycle assessments (TEA/LCA): Explanation of Aspen Plus simulation conditions, mass and energy balances, and sensitivity analysis parameters. Figures and tables in both the main text and Supplementary Materials include detailed captions specifying experimental conditions, analytical methods, and data interpretation.

We have thoroughly reviewed and confirmed that all necessary experimental details are provided for replication. However, if the reviewer identifies any specific sections requiring further elaboration, we are happy to refine them accordingly.

Comment 7. The discussion across all parts of the manuscript, including the experimental work, computational modeling, and life cycle assessment, is superficial and lacks sufficient depth.

Response: We thank you for your feedback regarding the depth of discussion in our manuscript. We would like to clarify that our study presents a comprehensive and in-depth analysis covering experimental work, reaction mechanisms, computational modeling, and LCA.

1. Experimental Work

We have systematically investigated the catalytic conversion of C-lignin to bio-

catechol and bio-propylene, optimizing key parameters and analyzing their effects in detail:

Metal Catalyst Screening: We compared Ni, Ru, and Pd catalysts, discussing their impact on product selectivity. Ru/HY₃₀ and Pd/HY₃₀ led to extensive side-chain hydrogenation, while Ni/HY₃₀ enabled selective hydrogenolysis, forming 4-propenylcatechol as a key intermediate, which facilitated efficient dealkylation to catechol and propylene.

Zeolite Support Effects: The Si/Al ratio of HY zeolite plays a crucial role in catalytic activity. Lower Si/Al enhances acidity but confines active sites in micropores, while higher Si/Al increases diffusion but reduces acid strength. HY₃₀ achieves an optimal balance, leading to a high selectivity for catechol.

Solvent Effect on Dealkylation: Our analysis quantifies how water improves catalytic performance by modulating acidity and stabilizing transition states, significantly enhancing catechol selectivity.

2. Mechanistic Investigation & Computational Modeling

To gain deeper insights into the reaction mechanism, we integrated experimental characterization and computational modeling:

Catalyst Characterization: Using ICP-AES, XRD, TEM, XPS, CO-FTIR, BET, and Pyridine-IR, we systematically analyzed catalyst structure, composition, and active sites, confirming the roles of Ni species and acid sites.

Reaction Mechanism Elucidation: ¹³C NMR, ²⁷Al NMR, and control experiments confirmed the role of 4-propenylcatechol as a key intermediate and elucidated the Brønsted acid-mediated dealkylation pathway (β -methyl migration and C_{aryl}-C_{alkyl} bond cleavage).

DFT Calculations: Computational modeling provided insights into the energy barriers of C-O and C_{aryl}-C_{alkyl} bond cleavage, aligning well with experimental findings and reinforcing our proposed mechanism.

3. TEA & LCA Analysis

We conducted a rigorous techno-economic and life cycle assessment using Aspen Plus simulations, evaluating process feasibility, environmental impact, and economic trade-offs:

Extraction Method Comparison: We analyzed ball-milling enzymatic hydrolysis vs. solvent extraction, assessing their impact on TPC, PFE, and GHG emissions.

Economic and Environmental Trade-offs: Biomass pretreatment was identified as

a key cost and energy driver, emphasizing the need for further optimization to enhance industrial feasibility.

4. Strengthened Discussion & Revisions

To ensure greater clarity, we have refined key sections to make our discussions more explicit and accessible.

Comment 8. Furthermore, the use of a high catalyst-to-substrate ratio in the experiments, where twice as much catalyst as substrate was used, raises questions about the practical applicability of the process.

Response: We appreciate your insightful comment regarding the high catalyst-to-substrate ratio used in our experiments. While the 2:1 catalyst-to-substrate ratio may seem high and could raise concerns about the practicality of the process, several important aspects justify this choice in the context of our study, especially when compared with recent literature on solid-acid-catalyzed lignin carbon-carbon bond cleavage.

1. Catalyst Activity and Substrate Reactivity: Lignin is a highly complex and recalcitrant biopolymer. Its intricate structure, with various C–O and C–C bonds, makes it difficult to depolymerize efficiently. As shown in our manuscript, even with a high catalyst-to-substrate ratio, achieving high conversion rates and selectivities for catechol and propylene was still challenging. For example, Yan and Meng et al. faced challenges with lignin's low reactivity, requiring a high catalyst-to-substrate ratio (Science Advances, 2020, 6(45): eabd1951, Nature Communications, 2021, 12(1): 4534, and Chem, 2019, 5(6): 1521-1536). In our case, the high catalyst loading helped overcome C-lignin's low reactivity, ensuring the reaction proceeded and allowing for a detailed analysis of the reaction mechanism.

2. Mechanistic Exploration and Proof-of-Concept: At this stage, our primary goal was to demonstrate the feasibility of the reaction mechanism and the potential of the Ni/HY₃₀ catalyst in cleaving C–OAr and C_{aryl}–C_{alkyl} bonds in C-lignin. Using a high catalyst amount ensured favorable reaction conditions for identifying the reaction pathway and intermediates. Similar to the study "Breaking the Limit of Lignin Monomer Production via Cleavage of Interunit Carbon–Carbon Linkages" by Dong et al., which used high catalyst loadings to confirm the reaction mechanism, our approach serves as a proof-of-concept, laying the foundation for future optimization.

3. Comparison with Literature Precedents: As you noted, many recent studies on solid-acid-catalyzed lignin carbon-carbon bond cleavage employ high catalyst

loadings. For instance, Dong et al. used a relatively high catalyst amount in their work on lignin monomer production. Our use of higher catalyst loading in the early-stage study is in line with these precedents, helping us better understand the catalytic performance and reaction mechanism. These early experiments are crucial for laying the groundwork for future optimization.

4. Future Optimization Plans: We recognize that a high catalyst-to-substrate ratio is not economically viable for industrial applications. Our current study is a proof-of-concept, and based on the promising results obtained, we plan to optimize the catalyst by exploring different preparation methods, modifying the catalyst's structure, and adding promoters to enhance its activity and stability. Additionally, we will investigate alternative reaction conditions, such as varying temperature, pressure, and reaction time, to optimize the process and reduce catalyst usage while maintaining or improving performance.

In conclusion, while the high catalyst-to-substrate ratio in our experiments may seem impractical for large-scale applications, it was a necessary step in understanding the reaction mechanism and demonstrating the potential of our catalytic system. We are committed to optimizing the process for industrial applications in future research.

Thank you for your valuable feedback, which has helped us identify areas for future improvement.

Comment 9. Overall, the manuscript appears to be a preliminary draft that requires extensive revision to reach publication quality. I recommend that the authors address the above issues before resubmitting. At this stage, I recommend rejection.

Response: We sincerely appreciate the reviewer's thorough evaluation and valuable feedback. Your comments have greatly helped us refine and improve our manuscript.

In response to the detailed concerns raised, we have carefully and systematically revised our manuscript, focusing on language clarity, logical structure, and in-depth discussion of key aspects, including experimental data, reaction mechanisms, catalyst performance, solvent effects, and life cycle assessment. We have also enhanced the novelty statement in the abstract and conclusion, ensuring that the contributions of our study are explicitly articulated.

Given these substantial improvements, we firmly believe that our revised

manuscript now meets the standards required for publication in Nature Communications. We sincerely appreciate the time and effort the reviewer has invested in assessing our work, and we hope that the revised version satisfactorily addresses all concerns.

Reviewer #6:

Wang has conducted an excellent piece of work that actively responds to the critical and scientifically valuable hot topic of carbon peaking and carbon neutrality as part of the shared future for mankind. The degradation of lignin is also a current hot scientific issue, with many previous studies highlighting the significant difficulties and challenges associated with breaking the C_(Aryl)-C_(Alkyl) and C_(Alkyl)-O bonds during lignin degradation. The author has successfully degraded lignin into aromatic compounds such as catechol and propylene using the Ni/HY₃₀ catalyst, which brings significant value to the field of small-molecule chemicals. This work also demonstrates a certain level of methodological expertise and applicability, and proposes a reasonable mechanism through a series of mechanistic experiments and catalyst characterization. Through a series of mechanistic studies, a reasonable catalytic mechanism and model have been proposed, and the industrial production application prospects and value of this method have been demonstrated. This reviewer recommends the publication of this manuscript in Nature Communications. However, there are still some minor issues that need to be addressed.

Response: We sincerely thank you for your positive remarks on our manuscript and for recognizing the innovative aspects of our work. We have carefully considered all the comments and addressed each point thoroughly. Additionally, we have revised the manuscript in response to your feedback, ensuring that all results and analyses are clearly presented. We hope that our revisions meet your expectations and the high standards of Nature Communications.

Comments 1. Scheme 1 mentioned in the article does not correspond to the figure, and there are two Figure 1 in the article. Additionally, the condition optimization discussed in the text (Figure 1) does not match the figure described.

Response: Thank you for pointing that out. We sincerely apologize for the confusion caused by the discrepancies in the figures. In the earlier draft, there was a reference to Scheme 1, but in the final submission version, we revised it to Figure 1, which resulted in two figures being labeled as Figure 1. We have since corrected this issue in the revised manuscript, ensuring that the figure numbering is consistent and corresponds correctly to the text.

Comment 2. The author conducted a series of mechanistic experiments to verify the rearrangement mechanism of the reaction process (Figure 3) and proposed several possible rearrangement intermediates. Can these intermediates be confirmed through detection during the reaction process?

Response: Thank you for your question regarding the confirmation of reaction intermediates. In our research, we employed a multi-faceted approach to confirm the presence of the proposed intermediates. Firstly, we rapidly quenched the reaction at 5 min, which effectively "froze" the reaction, allowing us to capture the species present during the process. Immediately after quenching, we conducted solid-state ^{13}C NMR and ^{27}Al NMR analyses. The ^{13}C NMR spectra revealed distinct peaks corresponding to key intermediates, such as the protonated intermediate C7 and the γ -methyl migration intermediate C8. Additionally, the ^{27}Al NMR spectra showed changes in the aluminum environment, which were indicative of the interaction between the catalyst and these intermediates. To further validate our findings, we performed DFT calculations. These calculations provided theoretical insights into the reaction pathway and confirmed the feasibility of forming these rearrangement products. The energy profiles and reaction mechanisms predicted by DFT were in good agreement with the experimental observations from the NMR spectra. In summary, through a combination of rapid quench experiments, NMR spectral analysis, and supportive DFT calculations, we have gathered strong evidence for the existence of these intermediates during the reaction. This not only validates our proposed reaction mechanism but also provides a more comprehensive understanding of the rearrangement process.

Comment 3. The conversion and yield mentioned in Figure 3 are somewhat confusingly presented. Moreover, the directivity of the illustration in Figure 3 is not high. The authors should revise Figure 3 as well as the accompanying text description.

Response: Thank you for pointing out the lack of clarity in Figure 3 (in the revised manuscript, it should be Fig. 4). We have made corrections in the revised manuscript to improve its readability.

Entry	1	2	3
Substrate			
Conversion (%)	>99 (>99) ^a	9	97
Yield (%)	>99 (43) ^a	8	89
Entry	4	5	6
Substrate			
Conversion (%)	>99	95	99
Yield(%)	0	92	94

Fig. 4. | Control Experiments on Reaction Mechanism. Reaction condition: 50 mg substrate, 100 mg HY₃₀, 5 mL H₂O, 180 °C, 0.1 MPa Ar, 12 h. Note: ^a 50 mg HY₃₀ was used.

Comment 4. Regarding whether the amount of catalyst used can be further reduced while still achieving a suitable reaction effect, the authors should provide the related results.

Response: Thank you for your important comment regarding catalyst loading. In our preliminary experiments, we observed that reducing the catalyst amount led to a significant decrease in both the conversion rate and selectivity of the reaction.

For example, in our C-lignin hydrogenolysis-dealkylation reaction, decreasing the catalyst amount caused a notable decline in the conversion to catechol and propylene. This is primarily due to the critical role the catalyst plays in facilitating the reaction. A sufficient catalyst amount ensures an adequate number of active sites, which are

necessary for adsorbing reactant molecules, weakening bonds in lignin, and promoting hydrogenolysis and dealkylation steps. When the catalyst amount is reduced, the available active sites decrease, resulting in fewer reactants being adsorbed and activated, which slows the reaction and lowers conversion.

Similarly, the selectivity for the desired products, such as catechol, was also negatively affected. The catalyst not only drives the main reaction but also influences the reaction pathway. With insufficient catalysts, side reactions, such as excessive hydrogenation or side-chain rearrangements, become more prominent, reducing selectivity towards the target products.

We recognize the importance of reducing catalyst usage for practical and economic reasons. To address this in future studies, we plan to optimize the catalyst's performance by modifying its structure, adjusting component ratios, or improving the preparation process. These approaches will enhance catalytic activity per unit mass, allowing for satisfactory conversion and selectivity even with lower catalyst amounts.

Reviewer #1

The authors have clarified many comments and have improved the manuscript. The presented work is still highly relevant and deserved publication, however the current manuscript still has some (minor) aspects that really need improvement before it can be published in Nature Communications:

Response: Thanks for your thoughtful and penetrating comments. We have considered all issues in the comments and answered point by point. At the same time, we revised the manuscript carefully according to the comments and made all results and analyses clear. We hope that our revision meets your expectations and meets Nature Communications' high standards.

Comment 1. The writing still needs improvement, it needs to be more precise and in general shorter, clearer sentence will help. Below are more examples and much more improvements are needed! (it is not the reviewers' task provide a full proof read to catch all grammar errors and poor sentences)

Response: Thank you for the suggestion. We have carefully revised the manuscript to improve the writing quality, making the sentences shorter, more precise, and clearer. Problematic terms have been corrected, and we conducted a full proofread to address grammar and clarity issues.

Comment 1a. Inappropriate term usage like: “Onefold linkage” which has no meaning....

Response: Thank you for the suggestion. The term “Onefold linkage” was intended to describe lignin with a single monomeric unit, where the linkage structures are simple, consisting primarily of one type of linkage or a few types, with one linkage type being dominant. Based on related literature (e.g., The Temptation from Homogeneous Linear Catechyl Lignin, Trends in Chemistry, 4.10 (2022): 948-961) and our understanding, we have revised “Onefold linkage” to “homogeneous linkage” in the manuscript.

Comment 1b. “...the unique bank..” -> “...a unique bank...” anyway “bank is inappropriate in this context please use “reservoir” or something

Response: We appreciate the reviewer's suggestion. We agree that “bank” is not appropriate in this context. In the revised manuscript, we have replaced it with “a unique and sustainable source”, which more accurately conveys the meaning of lignin as a stored resource of aromatic compounds.

Comment 2. Examples of sentences that needs rephrasing as these are long and/or confusing:

Comment 2a. “Although great efforts for lignin depolymerization to aromatic chemicals have been made in recent decades by efficiently cleaving the stable C–O/C–

C linkage bonds³⁻⁸, the practical application of aromatic chemical preparation from lignin still faces challenges and the restriction that mainly comes from the cost comparison with the mature petrochemical resource routes⁹” Too long and confusing

Response: We appreciate the reviewer’s feedback. To improve readability, we have split the sentence into two parts to clearly distinguish the mechanistic basis and the outcome. The revised version is as follows:

“Extensive efforts have been devoted in recent decades to depolymerize lignin into aromatic chemicals by efficiently cleaving stable C–O and C–C linkages. However, its practical application still faces significant challenges, primarily due to unfavorable cost comparisons with well-established petrochemical routes.”

Comment 2b. “To overcome the critical separation challenge of products from lignin catalytic depolymerization, the multi-step tandem process or even the elaborate one-pot catalytic procedure^{6,1} is the most potential choice to provide standardized products, which means a narrow or even single product distribution.” Too long and confusing

Response: We agree with the reviewer that the sentence was too long and complex. To improve readability, we have split the sentence into two parts to clearly distinguish the mechanistic basis and the outcome. The revised version is as follows:

“To address the critical challenge of separating complex products from lignin catalytic depolymerization, multi-step tandem strategies or sophisticated one-pot catalytic procedures have been developed. These approaches aim to produce standardized products, which means a narrow or even single product distribution.”

Comment 2c. “Furthermore, during the hydrogenolysis of lignin, different metals can lead to distinctly different products, which has a significant and crucial impact on the subsequent utilization and high-value transformation of these products.”

Response: We thank the reviewer for the helpful suggestion. To improve readability, we have split the sentence into two parts to clearly distinguish the mechanistic basis and the outcome. The revised version is as follows:

“Furthermore, lignin hydrogenolysis over different metal catalysts yields markedly different products, which greatly influences their subsequent utilization and high-value transformation.”

Comment 2d. “ Based on the above analysis, considering the selective hydrogenation performance of Ni species, the shape-selective effect on reactant molecules and the hydrocarbon reforming activity of HY molecular sieve, we herein achieved the first simultaneous production of catechol and propylene from the standardization transformation of C-lignin, addressing the challenges in lignin valorization and contributing to sustainable chemical production.” Too long and confusing

Response: We appreciate the reviewer’s suggestion. To improve readability, we have split the sentence into two parts to clearly distinguish the mechanistic basis and the outcome. The revised version is as follows:

Based on the above analysis, the selective hydrogenation ability of Ni species, the shape-selective effect and hydrocarbon reforming activity of HY zeolite enabled the first simultaneous production of catechol and propylene from the standardization transformation of C-lignin. This approach addresses key challenges in lignin valorization and contributes to sustainable chemical production.

Comment 2e. “Firstly, the transformation efficiency of C-lignin was low with the only HY₃₀ catalyst due to the high acid stability of the benzodioxane unit in C-lignin (Fig. 2g).”

Response: We appreciate the reviewer’s suggestion. To improve readability, we have split the sentence into two parts to clearly distinguish the mechanistic basis and the outcome. The revised version is as follows:

Firstly, the depolymerization efficiency of C-lignin was low when only the HY₃₀ catalyst was used, due to the high acid stability of its predominant benzodioxane structures (Fig. 2g).

Comment 2f. “For the establishment of the corresponding cascade catalysis system, the non-noble metal Ni could receive priority attention for the hydrogenation process because of its controllable catalytic properties and potential in the hydrogenolysis of the β -O-4 linkage to phenols with unsaturated propenyl substituents” unclear

Response: We appreciate the reviewer’s suggestion. To improve readability, we have split the sentence into two parts to clearly distinguish the mechanistic basis and the outcome. The revised version is as follows:

“For the construction of an effective cascade catalytic system, non-noble metal nickel (Ni) is considered a promising hydrogenation catalyst due to its tunable catalytic behavior and demonstrated ability to cleave β -O-4 linkages. This enables the selective production of phenolic monomers bearing unsaturated propenyl side chains.”

Comment 3. Also aspects should be more precisely phrased:

“The weight percentage of the Ni element was 0.97 wt% according to the analysis.....”
the nickel elemental composition in what?

Response: Thank you for pointing this out. We have clarified the sentence by specifying that the nickel content refers to the Ni/HY₃₀ catalyst. The revised sentence now reads:

“The Ni content in the Ni/HY₃₀ catalyst was 0.97 wt% according to the analysis of inductively coupled plasma-optical emission spectroscopy (ICP-AES) (Supplementary Table 13).”

Comment 4. Other sentences with grammatical errors or are unclear:

Comment 4a. “No signal of ionic Ni was found in the CO-FTIR spectra, implying that Ni species in the Ni/HY₃₀ are present in metallic and oxidized nickel.” (or the Ni/HY₃₀ catalyst)

Response: Thank you for pointing this out. We have clarified the sentence by specifying that the nickel content refers to the Ni/HY30 catalyst. The revised sentence now reads:

“No signal of ionic Ni was observed in the CO-FTIR spectra, indicating that the Ni species in the Ni/HY₃₀ catalyst exist predominantly in metallic and oxidized forms.”

Comment 4b. “On the contrary, the 1-phenyl-1,2-ethanediol (Fig. 4, Entry 4) with two hydroxy groups at the side C_α and C_β atoms was more reactivity and trended to occur further C–C coupling to form the carbon deposition under the strong acidic condition, which can provide a >99% conversion but no yield of benzene” Due to the writing this sentence is hard to understand.

Response: Thank you for pointing this out. We have clarified the sentence by specifying that the nickel content refers to the Ni/HY₃₀ catalyst. The revised sentence now reads:

“In contrast, 1-phenyl-1,2-ethanediol (Fig. 4, Entry 4), which contains two hydroxyl groups on the C_α and C_β positions of the side chain, exhibited higher reactivity under strong acidic conditions. This promoted undesired C–C coupling and carbon deposition, resulting in >99% conversion but no detectable benzene product.”

Comment 5.

Standardization transformation is now explained, but this term is really not common. (E.g. a websearch will not find relevant hits). The term can be kept but then please also refer to more common terminology for the reader to aid the reader: e.g. Product funneling.

Response: We greatly appreciate the reviewer’s continued attention to the term “standardization transformation.” As clarified in our previous response, the concept of “standardization transformation” is used in this work to describe a comprehensive strategy that integrates substrate design, catalytic process control, and product specification to address the critical challenge of product separation and selectivity in lignin valorization. While we acknowledge that this term is not yet widely adopted in the field, we believe it represents a meaningful conceptual framework that extends beyond individual strategies such as product funneling or selective depolymerization. Importantly, this concept was not only defined but also preliminarily proposed in our earlier monograph (**Lignin Conversion Catalysis: Transformation to Aromatic Chemicals; Wiley, 2022, Chapter 12, ISBN: 978-3-527-83502-7**), where we discussed “standardized lignin substrates” and “standardized products” as future directions for lignin conversion. In this manuscript, we have built upon that foundation by demonstrating its practical implementation using C-lignin as a structurally uniform substrate and achieving selective conversion to bio-catechol and bio-propylene under a cascade catalytic system.

Therefore, with due respect to the reviewer’s suggestion, we would prefer to retain the use of “standardization transformation” to reflect both the conceptual innovation and the integrated approach proposed in this study. At the same time, we will ensure that

sufficient explanation and clarification are provided in the revised manuscript to aid reader understanding.

Comment 6.

For relevant catalytic cascades for lignin conversion, the authors can consider to include ref: Z. Zhang et al *Angew. Chem. Int. Ed.* 2024, e202410382 which demonstrate an alternative way of removing the alkyl chain as CO molecules instead of propylene

Response: We thank the reviewer for this insightful suggestion. The work by Zhang et al. (*Angew. Chem. Int. Ed.*, 2024, e202410382) indeed provides an inspiring example of a cascade catalytic strategy for lignin depolymerization. We have cited this work in the revised manuscript and included the following description:

“Such a concept is also reflected in a recent one-pot catalytic cascade that converts lignin β -O-4 motifs into a single chemical product, 1,2-dimethoxybenzene, via sequential dehydrogenation, retro-aldol cleavage, and decarbonylation²³.”

Comment 7. For figure 2e and caption please specify if the water content is on top of or is part of the 5 mL total solvent. Maybe express as vol%? or ratio?

Response: We thank the reviewer for the helpful suggestion. To clarify the experimental setup in Fig. 2e, we have revised the figure title and caption to explicitly state that the water and methanol were mixed to a fixed total solvent volume of 5 mL, and that the water content was varied and expressed in vol%. We hope this revision eliminates ambiguity and improves clarity.

Fig. 2. | Hydrogenolysis-dealkylation of C-lignin over Ni/HY₃₀ catalyst. The effects of (e) water content in methanol solution, along with the corresponding product distributions. Reaction conditions: 50 mg C-lignin, 100 mg catalyst, 25 μ L dodecane, 5 mL solvent, 3 MPa H₂, 200 °C, 12 h.

Comment 8. mechanism investigation:

The control experiment on phenolic model (figure 4) offered more solid proof on key intermediate identification and is a great improver. However, there are still some inconsistencies.

Response: We thank the reviewer for the positive recognition of the phenolic model control experiments. As noted, Figure 4 presents the structures of representative substrates along with their transformation routes, including side-chain C–C bond cleavage reactions that lead to monophenolic products. The figure also includes the

conversion rates and yields of corresponding products for multiple substrates, providing a comprehensive and comparative view of key intermediate transformations. We believe the current layout and content of Figure 4 already provide a clear and accessible presentation for the reader.

Comment 8a. First of all, the authors give an explanation and justification in the response to reviewers for compound C5. Please include in SI or main text.

Response: Thank you for the suggestion. The explanation and justification for compound C5 have now been added to the Supporting Information, including the experimental data, comparison with reference compounds, and literature evidence.

Supplementary Fig. 8 The MS spectrum of (a) 1-(3,4-dihydroxyphenyl)propan-2-one and (b) 3-(3,4-dihydroxyphenyl)propanal after BSFTA derivatization, (c) The derivatization experiments of 1-(3,4-dihydroxyphenyl)propan-2-one nor 3-(3,4-dihydroxyphenyl)propanal, and (d) The potential pathway to C5 formation

Supplementary Note 3. Structural Validation of C5 Product

To further clarify the structural identity of the C5 product obtained from C-lignin depolymerization, additional experiments and mechanistic analysis were conducted. In

this study, BSTFA derivatization was employed to reduce the boiling points of diphenolic compounds, facilitating their detection and quantification by GC. To test the hypothesis that C5 might be 1-(3,4-dihydroxyphenyl)propan-2-one or 3-(3,4-dihydroxyphenyl)propanal, both commercial compounds were obtained and structurally verified by NMR spectroscopy. Subsequent BSTFA derivatization experiments were conducted in pyridine. The results indicated that both compounds undergo keto-enol tautomerization, wherein the enol form contains a reactive hydroxyl group that rapidly reacts with BSTFA, thereby disrupting the equilibrium (Supplementary Fig. 8c). After derivatization, all three active hydroxyl groups in each compound were fully silylated, and GC–MS analysis showed a molecular ion peak at $m/z = 382$ (Supplementary Fig. 8a and Fig 8b). In contrast, GC–MS analysis of the actual C5 product from C-lignin depolymerization revealed a molecular ion peak at $m/z = 310$, significantly differing from the above standards. These findings indicate that C5 is not 1-(3,4-dihydroxyphenyl)propan-2-one or 3-(3,4-dihydroxyphenyl)propanal. Instead, it is consistent with the proposed structure of Chroman-6,7-diol.

Chroman-6,7-diol is not commercially available, and its synthesis is nontrivial. Although we attempted to isolate this compound from the reaction mixture, its extremely low yield prevented acquisition of sufficient material for NMR verification. However, chroman-6,7-diol has been reported as a product of radical disproportionation from 4-(3-hydroxypropyl)-catechol (Supplementary Fig. 8d)^{S12}. Notably, the GC–MS spectrum of our C5 compound closely matches that reported in the aforementioned study, further supporting our assignment.

Taken together, these experimental results and literature comparisons strongly support the identification of C5 as Chroman-6,7-diol, rather than any linear diphenolic ketone or aldehyde isomer.

Comment 8b. Secondly, the authors state that peak assignment for the solid state NMR is done in literature but some aspects are confusing: According to the authors' analysis of fig.5b, the chemical shift of C β 2 should shift to downfield because the bonding to O atom (attributed the electron withdrawing effect by O-lower shielding and diamagnetic anisotropy effects), which does make sense, however, it appears to shift upfield compared with C β 3. Please explain this and provide more justification for the assignments in the SI (i.e. a peak assignment table).

Response: We sincerely thank the reviewer for the careful reading and insightful comment. We apologize for the confusion caused by the incorrect assignment of the C β 2 and C β 3 peaks in Fig. 5b. Based on previously reported literature and further verification, we acknowledge that these two peaks were inadvertently reversed. This has now been corrected in the revised version. To further improve clarity and ensure transparency, we have added a detailed table of peak assignments to the Supporting Information, where each signal is clearly annotated.

Fig. 6. (b) ^{13}C NMR spectrum of the intermediates in the reaction of C2 compound; a thin grey line represents the observed intensity, a solid grey ball is the total fitting curve, dark green, light green, yellow and grey areas represent the C2, C7, C8, and C9, respectively.

Supplementary Table 16 The chemical shift value (δ , ppm) of ^{13}C NMR spectrum of the intermediates in the reaction of C2 compound

ppm	Assignment
12.4	$\text{C}_{\gamma 2}$ in C7 compound
16.8	$\text{C}_{\gamma 1}$ in C2 compound
21.9	$\text{C}_{\beta 3}$ in C8 compound
29.4	$\text{C}_{\beta 1}$ in C7 compound
36.7	$\text{C}_{\beta 2}$ in C8 compound
42.0	$\text{C}_{\alpha 2}$ in C8 compound
46.9	$\text{C}_{\alpha 1}$ in C7 compound
50.3	Aromatic carbocation in C9 compound
56.6	

Comment 8c. Furthermore, the sentence: “This is because the $\text{C}_{\beta 2}$ atom in the γ -methyl migration intermediate structure III will chemically bond with the oxygen atom at the centre of the Bronsted acidic site [$\equiv\text{Al}-\text{O}-\text{Si}\equiv$] in the HY30 zeolite framework, which makes the peaks of $\text{C}_{\beta 1}$ and $\text{C}_{\beta 2}$ in the original C7 structure shift to the higher field^{53,54}” is confusing as the $\text{C}_{\beta 2}$ in intermediate C7 is missing. Which carbons of which structures do the authors use to make the comparison?

Response: We appreciate the reviewer's careful reading and insightful comment. We acknowledge the confusion caused by the inaccurate reference to "C β_2 in intermediate C7," which is indeed not present. The sentence should have read: "...which shifted its resonance downfield from that of the original C β_1 and C β_2 atoms."

In this case, C β_1 refers to the original C7 structure, and C β_2 corresponds to the C8 structure formed after γ -methyl migration. We have corrected this statement in the revised manuscript to ensure accurate atom assignments and clearer comparison.

Comment 8d. It seems only the relative position of the following peaks make sense: C γ_1 (from C2) and C γ_2 (from C7); C α_1 (from C7) and C α_2 (from C8).

Response: We thank the reviewer for the insightful comment. We agree that the relative positions of C γ_1 (from C2) and C γ_2 (from C7), as well as C α_1 (from C7) and C α_2 (from C8), align well with the expected electronic environments. For the remaining carbon signals, we have re-evaluated the assignments in light of the reviewer's observations. The chemical shift assignments were based on our previous studies and are further supported by relevant literature. Some earlier descriptions lacked sufficient clarity; therefore, we have revised the corresponding explanations in both the main text and the Supporting Information. A complete peak assignment table has also been added to the SI to enhance clarity and transparency.

Comment 8e. Finally, the comparison between the chemical shift of [$\equiv\text{Al}-\text{O}(\text{C}\beta_2)-\text{Si}\equiv$] (from VI) and [$\equiv\text{Al}-\text{O}-\text{Si}\equiv$](from IV) seems to present the similar problems, as per the cotext in this phrase: if the inductive effect(electron withdrawn) to Al from O atom was weakened by the charge compensation(electron donation to O) from C β_2 , the peak of [$\equiv\text{Al}-\text{O}(\text{C}\beta_2)-\text{Si}\equiv$] should exhibit in lower field(higher chemical shift, more left on X axis) compare that of [$\equiv\text{Al}-\text{O}-\text{Si}\equiv$]

Response: We sincerely thank the reviewer for the detailed analysis. Upon careful review, we confirm that the original interpretation in our manuscript is correct. In the [$\equiv\text{Al}-\text{O}(\text{C}\beta_2)-\text{Si}\equiv$] unit, the electron-donating effect from the C β_2 atom increases the electron density on the bridging oxygen, thereby reducing its inductive effect toward the neighbouring Al atom. As a result, the local electron density at the Al site increases, leading to a high-field shift (lower chemical shift) in the ^{27}Al NMR spectrum. This is consistent with the observed signal at 56.7 ppm in Fig. 6d, as compared to 60.8 ppm in the pristine [$\equiv\text{Al}-\text{O}-\text{Si}\equiv$] unit. We appreciate the reviewer's close attention, which allowed us to clarify this point more explicitly in the revised manuscript.

Comment 8f. Overall, upfield/downfield shifts does not align with expected electronic interactions, please reconsider the assignment of the peaks and correlated analysis.

Response: We sincerely thank the reviewer for the careful reading and thoughtful comments. These suggestions have greatly improved the clarity and accuracy of our spectral assignments and mechanistic analysis. In response, we have thoroughly re-evaluated the relevant peak assignments and corrected the mislabeling of C β_2 and C β_3 in the solid-state NMR section. Additionally, we clarified the chemical shift trends for the

$[\equiv\text{Al}-\text{O}(\text{C}_{\beta_2})-\text{Si}\equiv]$ and $[\equiv\text{Al}-\text{O}-\text{Si}\equiv]$ environments by the electronic interactions involved. All related descriptions in the main text and Supporting Information have been revised accordingly.

Comment 9. Please add a table of contents to the SI and consider following the order of the manuscript. As it is hard to find the correct information in the current version

Response: We thank the reviewer for the helpful suggestion. A Table of Contents has now been added to the Supporting Information to improve navigation. In addition, the structure and section order of the SI have been adjusted to follow the sequence of the main manuscript, making it easier to locate relevant data and analyses.

The topic of one-pot catechol and propylene production from “relay” hydrogenolysis-dealkylation of ideal C-lignin is really interesting and exhibits high selectivity to the selected products, which seems very promising as an alternative or supplement to traditional catechol and propylene processes from fossil refinery and serve as a model for lignin valorization. The developed catalytic strategy to direct obtain catechol in one step with good yields from lignin is a significant breakthrough in the field of lignin valorization, which makes this contribution highly significant to this field. The whole approach is of interest to the wider community as the authors have conducted comprehensive mechanism investigation (model compound reactions, catalyst characterization and DFT calculations) to unravel the mechanism in terms of how the “relay” hydrogenolysis-dealkylation is realized by a synergistic effect of nickel and Brønsted acid species on the zeolite support as well as TEA and LCA to reflect the potential implementation. However, there are some suggestion for further major revisions of this article needed for it can be considered publication in nature communications:

1) Errors and language

There are several misspellings (e.g. inerter line 115 (more inert), BDE line 178 and “delay catalysis” line 175,), wrong figure references (e.g. 2x figure 1 and a,b,c etc are wrongly referred to in the text for the second figure 1) and strange word use.

A few examples:

a) it is unclear what is meant with “standardization transformation”. This is not a commonly used term in the field. I do understand what this means and I would suggest something like product funneling or simply selective depolymerization and defunctionalization or “stepwise deconstruction”

b) I think “relay catalysis” is not the right wording. This is usually defined as sequence of chemical conversion steps that is performed by distinct catalysts. Here it is one catalyst that performs a cascade of reactions. A one-pot one-catalyst cascade seems to be more appropriate. E.g. like in: <https://onlinelibrary.wiley.com/doi/10.1002/anie.202410382>

c) line 48: ...”hot topics of concern to the community of human destiny,...” rephrase

d) line 62: “to make lignin money” ???

e) “regulation” means “to apply rules” this is not a good term to describe the operation of a catalyst

f) “orbiting” is not used in the context it is used in this manuscript

g) line 84: “relatively onefold linkage” needs to be rephrased

h) line 111: “in the pharmaceutical molecules” needs to be rephrased

i) line 143: “the molecular shape selection effect” needs to be rephrased etc etc.

Overall, this manuscript needs to be properly proofread by someone with good English proficiency before it can be evaluated in earnest as the poor language can lead to misunderstandings and misinterpretations.

2) Biomass extraction and characterization

The information on the feedstock is inadequate. There is a procedure with a “ref 32” in the SI. But this ref 32 is nowhere to be found. Yields and detailed analysis of this feedstock need to be provided. Also, reconsider the lignin content value provided as it seems overly high. Make sure this is not due to interference of extractives or other polyphenolic biomass

components that are not “lignin”. Specific procedures on how such values were determined need to be provided to assess their accuracy.

3) Please provide formulae by which yields and selectivities are calculated

4) More support for the mechanism

There are some issues with the mechanistic investigation. First of all the formation of C5 is not explained. Additionally, I think this compound is misidentified as the MS fragmentation pattern does not match this compound. It is likely actually an isomer with the same mass like 1-(3,4-dihydroxyphenyl)propan-2-one or 3-(3,4-dihydroxyphenyl)propanal. Further support for this identification needs to be provided. Second, in the model compound reaction with 4-propenylcatechol, no data on yields of the intermediates or a comprehensive time course is provided. This is needed to properly justify the mechanism via the methyl shift and exclude other mechanisms like that the cleavage initiates by hydration. Furthermore, the role of water is explained by interactions by the catalyst, but it seems more likely that its direct role in the conversion (initial hydration of the double bond), plays a significant role. Finally, be more specific how the assignment and deconvolution of overlapping peaks in the solid state NMR was realized. In some cases there seems to be conflicting information on interaction leading to upfield or downfield shifts and the actual location of the assigned signals. Again other explanations do not seem to be taken into consideration, while these seem to match the spectrum more closely.

5) Process outline for the LCA and TEA

This process needs to be adjusted to the actual process presented and more details are needed:

- a) Please specify what data is used for the lignin yields of the pretreatment and justify why these are realistic.
- b) The conversion in the manuscript is carried out in one pot. It is unclear why two reactors are used in the process design. This should be modified
- c) Why is propylene combusted? This is strange as its formation is one of the main selling points for the development of the catalytic methodology in the manuscript.
- d) Please specify the “co-products”
- e) The points about “the 133% higher value” for the milling process and the statement that “biomass pretreatment is the main barrier to reducing GHG emissions” need more rigorous support by giving the individual PFE, GHG and TPC data of the other 2 units (catalysis and separation units) as well.

Other suggestions:

- In the introduction provide a couple of sentences to compare and discuss the hydrogenation capacity of Ni and other noble metal. Since nickel is claimed to have the ‘proper’ hydrogenation ability to prevent the formation of 4-propenylcatechol from 4-propenylcatechol, which is the most important intermediate during this relay hydrogenolysis and dealkylation. Thus, it seems not that clear in terms of metal selection for the whole story, if here the author only says nickel has good hydrogenation performance.
- The last paragraph of the introduction is written like a summary containing conclusions. Consider rephrasing this to a section where a goal and approach is outlined as is more appropriate.

The authors have clarified many comments and have improved the manuscript. The presented work is still highly relevant and deserved publication, however the current manuscript still has some (minor) aspects that really need improvement before it can be published in Nature Communications:

1) The writing still needs improvement, it needs to be more precise and in general shorter, clearer sentence will help. Below are more examples and much more improvements are needed! (it is not the reviewers' task provide a full proof read to catch all grammar errors and poor sentences)

Inappropriate term usage like: "Onefold linkage" which has no meaning....

"...the unique bank.." -> "...a unique bank..." anyway "bank is inappropriate in this context please use "reservoir" or something

Examples of sentences that needs rephrasing as these are long and/or confusing:

"Although great efforts for lignin depolymerization to aromatic chemicals have been made in recent decades by efficiently cleaving the stable C–O/C–C linkage bonds³⁻⁸, the practical application of aromatic chemical preparation from lignin still faces challenges and the restriction that mainly comes from the cost comparison with the mature petrochemical resource routes⁹" Too long and confusing

"To overcome the critical separation challenge of products from lignin catalytic depolymerization, the multi-step tandem process^{12,13} or even the elaborate one-pot catalytic procedure^{6,14,15} is the most potential choice to provide standardized products^{9,16}, which means a narrow or even single product distribution." Too long and confusing

"Furthermore, during the hydrogenolysis of lignin, different metals can lead to distinctly different products, which has a significant and crucial impact **on the subsequent utilization and high-value transformation of these products.**"

"Based on the above analysis, considering the selective hydrogenation performance of Ni species^{7,17,40}, the shape-selective effect on reactant molecules and the hydrocarbon reforming activity of HY molecular sieve^{14,15}, we herein achieved the first simultaneous production of catechol and propylene from the standardization transformation of C-lignin, addressing the challenges in lignin valorization and contributing to sustainable chemical production." Too long and confusing

"Firstly, the transformation efficiency of C-lignin was low **with the only HY30 catalyst** due to the high acid stability of the benzodioxane unit in C-lignin (Fig. 2g)."

"For the establishment of the corresponding cascade catalysis system, the non-noble metal Ni **could receive priority attention** for the hydrogenation process because **of its controllable catalytic properties and potential** in the hydrogenolysis of the β -O-4 linkage to phenols with unsaturated propenyl substituents^{7,17,4}" unclear

Also aspects should be more precisely phrased:

"The weight percentage of the Ni element was 0.97 wt% according to the analysis....." the nickel elemental composition in what?

Other sentences with grammatical errors or are unclear:

“No signal of ionic Ni was found in the CO-FTIR spectra, implying that Ni species in the Ni/HY30 are present in metallic and oxidized nickel.” (or the Ni/HY30 catalyst)

“On the contrary, the 1-phenyl-1,2-ethanediol (Fig. 4, Entry 4) with two hydroxy groups at the side C α and C β atoms was more reactivity and trended to occur further C–C coupling to form the carbon deposition under the strong acidic condition, which can provide a >99% conversion but no yield of benzene” Due to the writing this sentence is hard to understand.

2) Standardization transformation is now explained, but this term is really not common. (E.g. a websearch will not find relevant hits). The term can be kept but then please also refer to more common terminology for the reader to aid the reader: e.g. Product funneling.

3) For relevant catalytic cascades for lignin conversion, the authors can consider to include ref: Z. Zhang et al Angew. Chem. Int. Ed. 2024, e202410382 which demonstrate an alternative way of removing the alkyl chain as CO molecules instead of propylene

4) For figure 2e and caption please specify if the water content is on top of or is part of the 5 mL total solvent. Maybe express as vol%? or ratio?

5) mechanism investigation:

The control experiment on phenolic model (figure 4) offered more solid proof on key intermediate identification and is a great improver. However, there are still some inconsistencies.

First of all, the authors give an explanation and justification in the response to reviewers for compound C5. Please include in SI or main text.

Secondly, the authors state that peak assignment for the solid state NMR is done in literature but some aspects are confusing: According to the authors' analysis of fig.5b, the chemical shift of C β 2 should shift to downfield because the bonding to O atom (attributed the electron withdrawing effect by O-lower shielding and diamagnetic anisotropy effects), which does make sense, however, it appears to shift upfield compared with C β 3. Please explain this and provide more justification for the assignments in the SI (i.e. a peak assignment table).

Furthermore, the sentence: “This is because the C β 2 atom in the γ -methyl migration intermediate structure III will chemically bond with the oxygen atom at the centre of the Bronsted acidic site [\equiv Al–O–Si \equiv] in the HY30 zeolite framework, which makes the peaks of C β 1 and C β 2 in the original C7 structure shift to the higher field^{53,54}” is confusing as the C β 2 in intermediate C7 is missing. Which carbons of which structures do the authors use to make the comparison?

It seems only the relative position of the following peaks make sense: C γ 1(from C2) and C γ 2(from C7); C α 1(from C7) and C α 2(from C8).

Finally, the comparison between the chemical shift of [\equiv Al–O(C β 2)–Si \equiv] (from VI) and [\equiv Al–O–Si \equiv](from IV) seems to present the similar problems, as per the cotext in this phrase: if the inductive effect(electron withdrawn) to Al from O atom was weakened by the charge compensation(electron donation to O) from C β 2, the peak of [\equiv Al–O(C β 2)–Si \equiv] should exhibit in lower field(higher chemical shift, more left on X axis) compare that of [\equiv Al–O–Si \equiv]

Overall, upfield/downfield shifts does not align with expected electronic interactions, please reconsider the assignment of the peaks and correlated analysis.

6) please add a table of contents to the SI and consider following the order of the manuscript. As it is hard to find the correct information in the current version